# Robust Meta-learning for Mixed Linear Regression with Small Batches

**Weihao Kong**[*]  **Raghav Somani**[†]  **Sham Kakade**[‡]  **Sewoong Oh**[§]

## Abstract

A common challenge faced in practical supervised learning, such as medical image processing and robotic interactions, is that there are plenty of tasks but each task cannot afford to collect enough labeled examples to be learned in isolation. However, by exploiting the similarities across those tasks, one can hope to overcome such data scarcity. Under a canonical scenario where each task is drawn from a mixture of $k$ linear regressions, we study a fundamental question: can abundant small-data tasks compensate for the lack of big-data tasks? Existing second moment based approaches of [42] show that such a trade-off is efficiently achievable, with the help of medium-sized tasks with $\Omega(k^{1/2})$ examples each. However, this algorithm is brittle in two important scenarios. The predictions can be arbitrarily bad $(i)$ even with only a few outliers in the dataset; or $(ii)$ even if the medium-sized tasks are slightly smaller with $o(k^{1/2})$ examples each. We introduce a spectral approach that is simultaneously robust under both scenarios. To this end, we first design a novel outlier-robust principal component analysis algorithm that achieves an optimal accuracy. This is followed by a sum-of-squares algorithm to exploit the information from higher order moments. Together, this approach is robust against outliers and achieves a graceful statistical trade-off; the lack of $\Omega(k^{1/2})$-size tasks can be compensated for with smaller tasks, which can now be as small as $\mathcal{O}(\log k)$.

## 1 Introduction

Modern machine learning tasks and corresponding training datasets exhibit a long-tailed behavior [73], where a large number of tasks do not have enough training examples to be trained to the desired accuracy. Collecting high-quality labeled data can be time consuming or require expertise. Consequently, in domains such as annotating medical images or processing robotic interactions, there might be a large number of related but distinct tasks, yet each task is associated with only a small batch of training data. However, one can hope to meta-train across those tasks, exploiting their similarities, and collaboratively achieve accuracy far greater than what can be achieved for each task in isolation [29, 58, 41, 52, 68, 61]. This is the goal of meta-learning [62, 67].

Meta-learning is especially challenging under two practically important settings: $(i)$ a few-shot learning scenario where each task is associated with an extremely small dataset; and $(ii)$ an adversarial scenario where a fraction of those datasets are corrupted. We design a novel meta-learning approach that is robust to such data scarcity and adversarial corruption, under a canonical scenario where the tasks are linear regressions in $d$-dimensions and the model parameters are drawn from a discrete distribution of a support size $k$.

---

[*]kweihao@gmail.com. University of Washington

[†]raghavs@cs.washington.edu. University of Washington

[‡]sham@cs.washington.edu. University of Washington & Microsoft Research

[§]sewoong@cs.washington.edu. University of Washington

First, consider a case where we have an *uncorrupted* dataset from a collection of $n$ tasks, each with $t$ training examples. Concretely, the $i$-th task for $i \in \{1, \ldots, n\}$ is associated with a regression parameter $\beta_i \in \{w_1, \ldots, w_k\}$ and a corresponding dataset $\{\mathbf{x}_{i,j} \in \mathbb{R}^d, y_{i,j} \in \mathbb{R}\}_{j=1}^t$ drawn from $y_{i,j} = \beta_i^\top \mathbf{x}_{i,j} + \epsilon_{i,j}$ for some noise $\epsilon_{i,j}$. A formal definition of the generative model is provided in § 1.1. If each task has a large enough training data with $t = \Omega(d)$ examples, it can be accurately learned in isolation. This is illustrated by solid circles in Fig. 1. On the opposite extreme, where each task has only a *single* example (i.e. $t = 1$), significant efforts have been made to make training statistically efficient [14, 77, 63, 78, 47, 16, 64]. However, even the best known result of [16] still requires exponentially many such tasks: $n = \Omega(de^{\sqrt{k}})$ (details in related work in §3). This is illustrated by solid squares in Fig. 1. Perhaps surprisingly, this can be significantly reduced to quasi-polynomial $n = \Omega(k^{\Theta(\log k)})$ sample complexity and quasi-polynomial run-time, with a slightly larger dataset that is only logarithmic in the problem parameters. This result is summarized in the following, with the algorithm and proof presented in §A of the supplementary material.

**Corollary 1.1** (of our results with no corruption, informal)**.** *Given a collection of $n$ tasks each associated with $t = \widetilde{\Omega}(1)$ labeled examples, if the effective sample size $nt = \widetilde{\Omega}(dk^2 + k^{\Theta(\log k)})$, then Algorithm 4 estimates the meta-parameters up to any desired accuracy of $\mathcal{O}(1)$ with high probability in time $\mathrm{poly}(d, k^{(\log k)^2})$, under certain assumptions on the meta-parameters.*

This is a special case of a more general class of algorithms we design, tailored for the following practical scenario; the collection of tasks in hand are heterogeneous, each with varying sizes of datasets (illustrated by the blue bar graphs below in Fig. 1). Inspired by the seminal work of [71], we exploit such heterogeneity by separating the roles of *light* tasks that have smaller datasets and *heavy* tasks that have larger datasets. As we will show, the size of the heavy tasks determines the order of the higher order moments we can reliably exploit. Concretely, we first use the light tasks to *estimate the subspace* spanned by the regression parameters, and then *cluster* heavy tasks by projecting them on the estimated subspace. The first such attempt was taken in [42], where a linkage-based clustering was proposed. However, as this clustering method relies on the second moment statistics, it strictly requires heavy tasks with $\Omega(k^{1/2})$ examples (left panel in Fig. 1). In the absence of such heavy tasks, the abundant light tasks are wasted as no existing algorithm can harness their structural similarities. Such second moment barriers are common in even simpler problems, e.g. [43, 44].

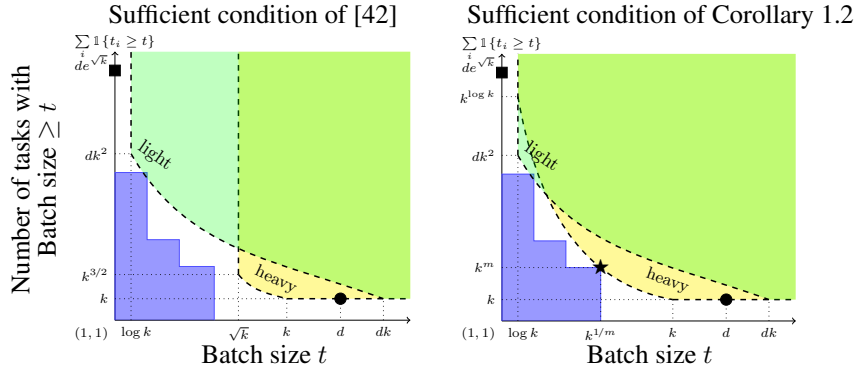

Figure 1: The blue bar graph summarizes the collection of tasks in hand, showing the cumulative count of tasks with more than $t$ examples. Typically, this does not include extremely large data tasks (circle) and extremely large number of small data tasks (square), where classical approaches succeed. When any point in the light (green) region and any point in the heavy (yellow) region are both realized by the blue graph, the corresponding algorithm succeeds. On the left, the collection in blue cannot be learned by any existing methods including [42]. Our approach in Corollary 1.2 significantly extends the heavy region all the way down to $\log k$, leading to a successful meta-learning in this example.

We exploit higher order statistics to break this barrier, using computationally tractable tools from sum-of-squares methods [45]. This gives a class of algorithms parameterized by an integer $m$ (for $m$-th order moment) to be chosen by the analyst tailored to the size of the heavy tasks in hand $t_H = \Omega(k^{1/m})$. This allows for a graceful trade-off between $t_H$ and with the required number of heavy tasks $n_H$. We summarize the result below, with a proof in §A, and illustrate it in Fig. 1

(right). The choice of $m = \Theta(\log k)$ gives the minimum required batch size, as we highlighted in Corollary 1.1.

**Corollary 1.2** (of our results with no corruption, informal)**.** *For any integer $m$, given two collections of tasks, first collection of light tasks with $t_L = \widetilde{\Omega}(1)$, $t_L n_L = \widetilde{\Omega}(dk^2)$, and the second collection of heavy tasks with $t_H = \widetilde{\Omega}(m\, k^{1/m})$, $t_H n_H = \widetilde{\Omega}(k^{\Theta(m)})$, the guarantees of Corollary 1.1 hold.*

Next, consider an adversarial scenario. Outliers are common in meta-learning as diverse sources contribute to the collection. Existing approaches are brittle to a few such outliers. [42] builds upon principal component analysis and linear regression, both of which are known to be brittle to outliers [40, 19]. For example, a single corrupted user can result in an arbitrarily bad subspace estimation in the first step of [42]. This causes the meta-learning algorithm to learn nothing about the true regression parameters, resulting in a completely random prediction in the subsequent step. A fundamental question of interest is, what can be meta-learned from past experience that is only partially trusted? Following robust learning literature [45, 25], we assume a general adversary who can adaptively corrupt any $\alpha$ fraction of the tasks, formally defined in Assumption 2. This parameter $\alpha \in [0, 1]$ captures how powerful an adversary is. Among all adversaries that can corrupt an $\alpha$ fraction of the dataset, we assume the strongest possible one that can adaptively select which samples to corrupt and replace them with arbitrary data points. We make both subspace estimation and clustering steps robust against adversarial corruption. The sum-of-squares approach is inherently robust, when used within an iterative clustering [45]. However, existing robust subspace estimation approaches are suboptimal, requiring $\widetilde{\mathcal{O}}(d^2)$ samples [24]. To this end, we introduce a novel algorithm, and prove its optimality in both accuracy and dependence in the dimension $d$. This resolves an open question posed in [64] on whether it is possible to robustly learn the subspace with $\widetilde{\mathcal{O}}(d)$ samples.

This achieves a similar sample complexity as the uncorrupted case in Corollary 1.2, while tolerating as much corruption as information theoretically possible: $\alpha = \widetilde{\mathcal{O}}(\epsilon/k)$ for an $\epsilon$ accuracy in parameter estimation. Such condition is necessary as otherwise the adversary can focus its attack on one of the mixtures, and incur $\Omega(\epsilon)$ error in estimating the parameter of that component.

**Corollary 1.3** (of Theorem 1, informal)**.** *For any $\epsilon \in (0, 1/k^3)$ and $m \in \mathbb{N}$, given two collections of tasks, the first with $t_L = \widetilde{\Omega}(1)$, $n_L t_L = \widetilde{\Omega}\big(dk\epsilon^{-2}\big)$, and the second with $t_H = \widetilde{\Omega}\big(mk^{1/m}\big)$, $n_H t_H = \widetilde{\Omega}\big(k^{\mathcal{O}(m)}\big)$, if the fraction of corrupted tasks is $\alpha = \widetilde{\mathcal{O}}(\epsilon/k)$, Algorithm 1 achieves up to $\epsilon$ accuracy with high probability in time $\mathrm{poly}\big(d, k^{m^2}, \epsilon^{-1}\big)$, under certain assumptions.*

We provide the algorithm (Algorithm 1) and the analysis (Theorem 1) under the adversarial scenario in the main text. When there is no corruption, the algorithm can be made statistically more efficient with tighter guarantees, which is provided in §A.

## 1.1 Problem formulation and notations

We present the probabilistic perspective on few-shot supervised learning following [31], but focusing on a simple yet canonical case where the tasks are linear regressions. A collection of $n$ tasks are independently drawn according to some prior distribution. The $i$-th task is associated with a *model parameter* $\phi_i = (\beta_i \in \mathbb{R}^d, \sigma_i \in \mathbb{R}_+)$, and a *meta-train dataset* $\{(\mathbf{x}_{i,j}, y_{i,j}) \in \mathbb{R}^d \times \mathbb{R}\}_{j=1}^{t_i}$ of size $t_i$. Each example $(\mathbf{x}_{i,j}, y_{i,j}) \sim \mathbb{P}_{\phi_i}(y|\mathbf{x})\mathbb{P}(\mathbf{x})$ is independently drawn from a linear model, such that

$$y_{i,j} \;=\; \beta_i^\top \mathbf{x}_{i,j} + \epsilon_{i,j} \;, \tag{1}$$

where $\mathbf{x}_{i,j} \sim \mathcal{N}(\mathbf{0}, \mathbf{I}_d)$ and $\epsilon_{i,j} \sim \mathcal{N}(0, \sigma_i^2)$. If $\mathbf{x}_{i,j}$ is from $\mathcal{N}(\mathbf{0}, \boldsymbol{\Sigma})$, we assume to have enough $\mathbf{x}_{i,j}$'s (not necessarily labeled) for whitening, and $\mathbb{P}(\mathbf{x})$ can be made sufficiently close to isotropic.

The goal of meta-learning is to train a model for a new arriving task $\phi_{\text{new}} \sim \mathbb{P}_\theta(\phi)$ from a small size *training dataset* $\mathcal{D} = \{(\mathbf{x}_j^{\text{new}}, y_j^{\text{new}}) \sim \mathbb{P}_{\phi_{\text{new}}}(y|\mathbf{x})\mathbb{P}(\mathbf{x})\}_{j=1}^\tau$ of size $\tau$. This is achieved by exploiting some structural similarities to the meta-train dataset, drawn from the same prior distribution $\mathbb{P}_\theta(\phi)$. To capture such structural similarities, we make a mild assumption that $\mathbb{P}_\theta(\phi)$ is a finite discrete distribution of a support size $k$. This is also known as mixture of linear experts [14]. Concretely, $\mathbb{P}_\theta(\phi)$ is fully defined by a *meta-parameter* $\theta = (\mathbf{W} \in \mathbb{R}^{d \times k}, \mathbf{s} \in \mathbb{R}_+^k, \mathbf{p} \in S^{k-1})$ with $k$ candidate model parameters $\mathbf{W} = [\mathbf{w}_1, \ldots, \mathbf{w}_k]$ and $k$ candidate noise parameters $\mathbf{s} = [s_1, \ldots, s_k]$. The $i$-th task is drawn from $\phi_i \sim \mathbb{P}_\theta(\phi)$, where first a $z_i \sim \text{multinomial}(\mathbf{p})$ selects a component that the task belongs to, and training data is independently drawn from Eq. (1) with $\beta_i = \mathbf{w}_{z_i}$ and $\sigma_i = s_{z_i}$.

Following the definition of [31], the *meta-learning problem* refers to solving the following:

$$\theta^* \in \arg\max_{\theta} \log \mathbb{P}(\theta|\mathcal{D}_{\text{meta-train}}) , \tag{2}$$

which estimates the most likely meta-parameter given *meta-training dataset* defined as $\mathcal{D}_{\text{meta-train}} := \{\{(\mathbf{x}_{i,j}, y_{i,j}) \in \mathbb{R}^d \times \mathbb{R}\}_{j=1}^{t_i}\}_{i=1}^n$. This is a special case of empirical Bayes methods [13]. Our goal is to solve this meta-learning problem *robustly* against an adversarial corruption of $\mathcal{D}_{\text{meta-train}}$ as formally defined in Assumption 2. Once meta-learning is solved, the model parameter of the newly arriving task can be estimated with a Maximum a Posteriori (MAP) or a Bayes optimal estimator:

$$\widehat{\phi}_{\text{MAP}} \in \arg\max_{\phi} \log \mathbb{P}(\phi|\mathcal{D}, \theta^*) , \quad \text{and} \quad \widehat{\phi}_{\text{Bayes}} \in \arg\min_{\phi} \mathbb{E}_{\phi' \sim \mathbb{P}(\phi|\mathcal{D}, \theta^*)}[\ell(\phi, \phi')] , \tag{3}$$

for some choice of a loss $\ell(\cdot)$, which is straightforward. This is subsequently used to predict the label of a new data point $\mathbf{x}$ from $\phi_{\text{new}}$. Concretely, $\widehat{y} \in \arg\max_y \mathbb{P}_{\widehat{\phi}_{\text{MAP/Bayes}}}(y|\mathbf{x})$.

**Notations.** We define $[n] := \{1, \ldots, n\}$, $\forall n \in \mathbb{N}$; $S^{k-1}$ as the standard $k$-dimensional probability simplex; $\|\mathbf{x}\|_p := (\sum_{i=1}^d |x_i|^p)^{1/p}$ as the standard vector $\ell_p$-norm of a vector $\mathbf{x} \in \mathbb{R}^d$ $\forall d \in \mathbb{N}$ $\forall p \geq 1$; $\|\mathbf{A}\|_* := \sum_{i=1}^{\min\{n,m\}} \sigma_i(\mathbf{A})$, $\|\mathbf{A}\|_{\text{F}} := (\sum_{i,j=1}^{m,n} A_{i,j}^2)^{1/2}$ as the standard nuclear norm and Frobenius norm of matrix $\mathbf{A} \in \mathbb{R}^{m \times n}$, where $\sigma_i(\mathbf{A})$ denotes the $i$-th singular value of $\mathbf{A}$ respectively; $\mathcal{N}(\boldsymbol{\mu}, \boldsymbol{\Sigma})$ is the multivariate normal distribution with mean $\boldsymbol{\mu}$ and covariance $\boldsymbol{\Sigma}$; $\mathbb{1}\{\cdot\}$ is the indicator function. We define $\rho_i^2 := s_{z_i}^2 + \|\mathbf{w}_{z_i}\|_2^2$ as the variance of a label $y_{i,j}$ in the $i$-th task, and $\rho^2 := \max_i \rho_i^2$. We define $p_{\min} := \min_{j \in [k]} p_j$, and $\Delta := \min_{i,j \in [k], i \neq j} \|\mathbf{w}_i - \mathbf{w}_j\|_2$ and assume $p_{\min}, \Delta > 0$. We use $\widetilde{\mathcal{O}}$ and $\widetilde{\Omega}$ notations that are extensions of the standard $\mathcal{O}$ and $\Omega$ Bachmann–Landau notations to hide poly-logarithmic factors.

## 1.2 Algorithm and intuitions

Following the recipe of spectral algorithms for clustering [71] and few-shot learning [42], we propose the following approach consisting of three steps. Clustering step requires *heavy tasks*; each task has many labeled examples, but we need a smaller number of such tasks. Subspace estimation and classification steps require *light tasks*; each task has a few labeled examples, but we need a larger number of such tasks. Here, we provide the intuition behind each step and the corresponding requirements. The details are deferred to §B, where we emphasize robustness to corruption of the data. The estimated $\widehat{\theta} = (\widehat{\mathbf{W}}, \widehat{\mathbf{s}}, \widehat{\mathbf{p}})$ is subsequently used in prediction, when a new task arrives.

---

**Algorithm 1**

**Meta-learning**
1. *Subspace estimation:* Compute subspace $\widehat{\mathbf{U}}$ which approximates $\text{span}\{\mathbf{w}_1, \ldots, \mathbf{w}_k\}$.
2. *Clustering:* Project the heavy tasks onto the subspace of $\widehat{\mathbf{U}}$, perform $k$ clustering, and estimate $\widetilde{\mathbf{w}}_\ell$ for each cluster $\ell \in [k]$.
3. *Classification:* Perform likelihood-based classification of the light tasks using $\{\widetilde{\mathbf{w}}_\ell\}_{\ell=1}^k$ estimated from the *Clustering* step; compute refined estimates $\{\widehat{\mathbf{w}}_\ell, \widehat{s}_\ell, \widehat{p}_\ell\}_{\ell=1}^k$ of $\theta$.

**Prediction**
4. *Prediction:* Perform MAP or Bayes optimal prediction using the estimated meta-parameter.

---

**Subspace estimation.** As $\boldsymbol{\Sigma} := \mathbb{E}[y_{i,j}^2 \mathbf{x}_{i,j} \mathbf{x}_{i,j}^\top] = c\mathbf{I}_d + 2\sum_{\ell=1}^k p_\ell \mathbf{w}_\ell \mathbf{w}_\ell^\top$ for some constant $c \geq 0$, the subspace spanned by the regression vectors, $\text{span}\{\mathbf{w}_1, \ldots, \mathbf{w}_k\}$, can be efficiently estimated by Principal Component Analysis (PCA), if we have uncorrupted data. This only requires $\widetilde{\Omega}(d)$ samples. With $\alpha$-fraction of the tasks adversarially corrupted, existing approaches of outlier-robust PCA attempt to simultaneously estimate the principal subspace while *filtering out* the outliers [75]. This removes many uncorrupted data points, and hence can either only tolerate up to $\alpha = \mathcal{O}(1/k^8)$ fraction of corruption (assuming well-separated $\mathbf{w}_\ell$'s). We introduce a new approach in Algorithm 2 that uses a second filter to recover those erroneously removed data points. This improves the tolerance to $\alpha = \mathcal{O}(1/k^4)$ while requiring only $\widetilde{\Omega}(d)$ samples (see Remark B.2). We call this step *robust subspace estimation* (Algorithm 2 in §2.2).

**Clustering.** Once we have the subspace, we project the estimates of $\beta_i$'s to the $k$-dimensional subspace and cluster those points to find the centers. As $k \ll d$ in typical settings, this significantly reduces the sample complexity from $\mathrm{poly}(d)$ to $\mathrm{poly}(k)$. Existing meta-learning algorithm of [42] proposed a linkage based clustering algorithm. This utilizes the bounded property of the second moment only. Hence, strictly requires *heavy tasks* with $t = \Omega(k^{1/2})$. We break this second moment barrier by exploiting the boundedness of higher order moments. The heavy tasks are now allowed to be much smaller, but at the cost of requiring a larger number of such tasks and additional computations.

One challenge is that the (empirical) higher order moments are tensors, and tensor norms are not efficiently computable. Hence boundedness alone does not give an efficient clustering algorithm. We need a stronger condition that the moments are Sum-of-Squares (SOS) bounded, i.e. there *exist* SOS proofs showing that the moments are bounded [45, 35]. This SOS boundedness is now tractable with a convex program, leading to a polynomial time algorithm that is also robust against outliers [45]. One caveat is that existing method in [45] requires data generated from a Poincaré distribution. As shown in Remark H.10, the distribution of our estimate $\widehat{\beta}_i = (1/t) \sum_{j=1}^{t} y_{i,j} \mathbf{x}_{i,j}$ is not Poincaré. Interestingly, as we prove in Lemma H.2, the higher order moments are still SOS bounded. This ensures that we can apply the robust clustering algorithm of [45]. We call this step *robust clustering* (Algorithm 7 in §H).

**Classification and parameter estimation.** Given rough estimates $\widetilde{\mathbf{w}}_\ell$'s as center of those clusters, we grow each cluster by classifying remaining light tasks. Classification only requires $t = \Omega(\log k)$. Once we have sufficiently grown each cluster, we can estimate the parameters to a desired level of accuracy. There are two reasons we need this refinement step. First, in the small corruption regime, where the fraction of corrupted tasks $\alpha$ is much smaller than the desired level of accuracy $\epsilon$, this separation is significantly more sample efficient. The subspace estimation and clustering steps require only $\mathcal{O}(\Delta/\rho)$ accuracy, and the burden of matching the desired $\epsilon$ level of error is left to the final classification step, which is more sample efficient. Next, the classification step ensures an adaptive guarantee. As parameter estimation is done for each cluster separately, a cluster with small noise $s_i$ can be more accurately estimated. This ensures a more accurate prediction for newly arriving tasks. We call this step *classification and robust parameter estimation* (Algorithm 9 in §I).

## 2 Main results

To give a more fine grained analysis, we assume there are two types of light tasks. In meta-learning, subspace estimation uses $\mathcal{D}_{L1}$, clustering uses $\mathcal{D}_H$, and classification uses $\mathcal{D}_{L2}$.

**Assumption 1.** *The heavy dataset $\mathcal{D}_H$ consists of $n_H$ heavy tasks, each with at least $t_H$ samples. The first light dataset $\mathcal{D}_{L1}$ consists of $n_{L1}$ light tasks, each with at least $t_{L1}$ samples. The second light dataset $\mathcal{D}_{L2}$ consists of $n_{L2}$ tasks, each with at least $t_{L2}$ samples. We assume $t_{L1}, t_{L2} < d$.*

The three batches of meta-train datasets are corrupted by an adversary.

**Assumption 2.** *From the datasets $\mathcal{D}_H$, $\mathcal{D}_{L1}$, and $\mathcal{D}_{L2}$, the adversary controls $\alpha_H$, $\alpha_{L1}$, and $\alpha_{L2}$ fractions of the tasks, respectively. The adversary is allowed to inspect all the examples, remove those examples associated with three subsets of tasks (of sizes at most $\alpha_H n_H$, $\alpha_{L1} n_{L1}$, and $\alpha_{L1} n_{L1}$ tasks from $\mathcal{D}_H$, $\mathcal{D}_{L1}$, and $\mathcal{D}_{L2}$), and replace the examples associated with those tasks with arbitrary points. The corrupted meta-train datasets are then presented to the algorithm.*

### 2.1 Meta-learning and prediction

We characterize the achievable accuracy in estimating the meta-parameters $\theta = (\mathbf{W}, \mathbf{s}, \mathbf{p})$.

**Theorem 1.** *For any $\delta \in (0, 1/2)$ and $\epsilon > 0$, given three batches of samples under Assumptions 1 and 2, the meta-learning step of Algorithm 1 achieves the following accuracy for all $i \in [k]$,*

$$\|\widehat{\mathbf{w}}_i - \mathbf{w}_i\|_2 \leq \epsilon s_i \,, \quad \left| \widehat{s}_i^2 - s_i^2 \right| \leq \epsilon s_i^2 / \sqrt{t_{L2}} \,, \quad \text{and} \quad |\widehat{p}_i - p_i| \leq \epsilon \sqrt{t_{L2}/d} \, p_i \,+\, \alpha_{L2} \,,$$

*with probability $1 - \delta$, if the numbers of tasks, samples in each task, and the corruption levels satisfy*

$$n_{L1} = \widetilde{\Omega}\left(\frac{dk^2}{\widetilde{\alpha}t_{L1}} + \frac{k}{\alpha^2}\right), \qquad t_{L1} \geq 1, \qquad \alpha_{L1} = \mathcal{O}(\widetilde{\alpha}),$$

$$n_H = \widetilde{\Omega}\left(\frac{(km)^{\Theta(m)}}{p_{\min}} + \frac{\rho^4}{\Delta^4 p_{\min} t_H}\right), \quad t_H = \Omega\left(\frac{\rho^2}{\Delta^2} \cdot \frac{m}{p_{\min}^{2/m}}\right), \qquad \alpha_H = \widetilde{\mathcal{O}}\left(p_{\min} \min\left\{1, \sqrt{t_H} \cdot \frac{\Delta^2}{\rho^2}\right\}\right),$$

$$n_{L2} = \widetilde{\Omega}\left(\frac{d}{t_{L2} p_{\min} \epsilon^2}\right), \qquad t_{L2} = \Omega\left(\frac{\rho^4}{\Delta^4} \log \frac{kn_{L2}}{\delta}\right), \quad \alpha_{L2} = \mathcal{O}(p_{\min}\epsilon/\log(1/\epsilon)),$$

*where* $\widetilde{\alpha} := \max\{\Delta^2 \sigma_{\min}^2/(\rho^6 k^2), \Delta^6 p_{\min}^2/(k^2\rho^6)\}$, $\sigma_{\min}$ *is the smallest non-zero singular value of* $\sum_{j=1}^{k} p_j \mathbf{w}_j \mathbf{w}_j^{\top}$, *and* $m \in \mathbb{N}$ *is a parameter chosen by the analyst.*

We refer to §1.1 for the setup and notations, and provide key lemmas in §B and a complete proof in §C. We discuss each of the conditions in the following remarks assuming $\Delta = \Omega(\rho)$, for simplicity.

**Remark 2.1** (Separating two types of light tasks)**.** *As* $t_{L1}$ *can be as small as one, the conditions on* $\mathcal{D}_{L2}$ *does not cover the conditions for* $\mathcal{D}_{L1}$. *The conditions on* $\alpha_{L1}$ *and* $n_{L1}$ *can be significantly more strict than what is required for* $\mathcal{D}_{L2}$. *Hence, we separate the analysis for* $\mathcal{D}_{L1}$ *and* $\mathcal{D}_{L2}$.

**Remark 2.2** (Dependency in $\mathcal{D}_{L1}$)**.** *Since we are interested the large* $d$ *small* $t_{L1}$ *setting, the dominant term in* $n_{L1}$ *is* $dk^2/\widetilde{\alpha}t_{L1}$. *The effective sample size* $n_{L1}t_{L1}$ *scaling as* $d$ *is information theoretically necessary. The* $\min\{1/\sigma_{\min}^2, 1/p_{\min}^2\}$ *dependence of* $n_{L1}t_{L1}$ *allows sample efficiency even when* $\sigma_{\min}$ *is arbitrarily small, including zero. This is a significant improvement over the* $\mathrm{poly}(1/\sigma_k)$ *sample complexity of typical spectral methods, e.g. [14, 77], where* $\sigma_k$ *is the k-th singular value of* $\sum_{\ell=1}^{k} p_\ell \mathbf{w}_\ell \mathbf{w}_\ell^{\top}$. *This critically relies on an extension of the gap-free spectral bound of [1, 47]. Our tolerance of* $\alpha_{L1} = \mathcal{O}(p_{\min}^2/k^2)$ *significantly improves upon the state-of-the-art guarantee of* $p_{\min}^4/k^4$ *as detailed in §2.2. Further, we show it is information theoretically optimal. This assumes only bounded fourth moment, which makes our analysis more generally applicable. However, this can be tightened under a stricter conditions of the distribution, as we discuss in §4.*

**Remark 2.3** (Dependency in $\mathcal{D}_H$)**.** *Assuming* $p_{\min} = \Omega(1/k)$, *the dominant term in* $n_H$ *is* $\widetilde{\Omega}((km)^{\Theta(m)}/p_{\min})$, *which is* $\widetilde{\Omega}(k^{\Theta(m)})$ *and the result is trivial when* $m \geq \log(k)$. *This implies a* $n_H = \widetilde{\Omega}(k^{\Theta(m)})$, $t_H = \Omega(m \cdot k^{2/m})$ *trade-off for any integer* $m$, *breaking the* $t_H = \Omega(k^{1/2})$ *barrier of [42]. In fact, for an optimal choice of* $m = \Theta(\log k)$ *to minimize the required examples, it can tolerate as small as* $t_H = \Omega(\log k)$ *examples, at the cost of requiring* $n_H = \widetilde{\Omega}(k^{\Theta(\log k)})$ *such heavy tasks. We conjecture* $t_H = \Omega(\log k)$ *is also necessary for any polynomial sample complexity. For the case of learning mixtures of isotropic Gaussians, [59] shows that super-polynomially many number of samples are information theoretically necessary when the centers are* $o(\sqrt{\log k})$ *apart. This translates to* $t = o(\log k)$ *in our setting. The requirement* $\alpha_H = \mathcal{O}(p_{\min})$ *is optimal. Otherwise, the adversary can remove an entire cluster.*

**Remark 2.4** (Dependency in $\mathcal{D}_{L2}$)**.** *The requirement* $n_{L2} \cdot t_{L2} = \widetilde{\Omega}(d/p_{\min}\epsilon^2)$ *is optimal in* $d, p_{\min}$ *and* $\epsilon$ *due to the lower bound for linear regression. The requirement on* $\alpha_{L2} = \mathcal{O}(p_{\min}\epsilon/\log(1/\epsilon))$ *is also necessary upto a log factor, from lower bound on robust linear regression [26].*

At test time, we use the estimated $\widehat{\theta} = (\widehat{\mathbf{W}}, \widehat{\mathbf{s}}, \widehat{\mathbf{p}})$ to approximate the prior distribution on a new task. On a new arriving task with training data $\mathcal{D} = \{(\mathbf{x}_j^{\mathrm{new}}, y_j^{\mathrm{new}})\}_{j=1}^{\tau}$, we propose the standard MAP or Bayes optimal estimators to make predictions on this new task. The following guarantee is a corollary of Theorem 1 and [42, Theorem 2]. The term $\sum_{i \in [k]} p_i s_i^2$ is due to the noise in the test data $(\mathbf{x}, y)$ and cannot be avoided. We can get arbitrarily close to this fundamental limit with only $\tau = \Omega(\log k)$ samples. This is a minimax optimal sample complexity as shown in [42].

**Corollary 2.5** (Prediction)**.** *Under the hypotheses of Theorem 1, the expected prediction errors of both the MAP and Bayes optimal estimators* $\widehat{\beta}(\mathcal{D})$ *defined in Eq. (3) are bound as* $\mathbb{E}[(\mathbf{x}^{\top}\widehat{\beta}(\mathcal{D}) - y)^2] \leq \delta + (1+\epsilon^2)\sum_{i=1}^{k} p_i s_i^2$, *if* $\tau = \Omega((\rho^4/\Delta^4)\log(k/\delta))$ *and* $\epsilon \leq \min\{\Delta/(10\rho), \Delta^2\sqrt{d}/(50\rho^2)\}$, *where the expectation is over the new task with model parameter* $\phi^{\mathrm{new}} = (\beta^{\mathrm{new}}, \sigma^{\mathrm{new}}) \sim \mathbb{P}_\theta$, *training data* $(\mathbf{x}_j^{\mathrm{new}}, y_j^{\mathrm{new}}) \sim \mathbb{P}_{\phi^{\mathrm{new}}}$, *and test data* $(\mathbf{x}, y) \sim \mathbb{P}_{\phi^{\mathrm{new}}}$.

## 2.2 Novel robust subspace estimation

Our main result relies on making *each step* of Algorithm 1 robust, as detailed in §B. However, as our key innovation is a novel *robust subspace estimation* in the first step, we highlight it in this section.

We aim to estimate the subspace spanned by the true meta-parameters $\{\mathbf{w}_1, \ldots, \mathbf{w}_k\}$. As $\mathbf{\Sigma} := \mathbb{E}[\widehat{\beta}_{i,j}\widehat{\beta}_{i,j}^{\top}] = \{\sum_{\ell=1}^{k} p_\ell(s_\ell^2 + \|\mathbf{w}_\ell\|_2^2)\}\mathbf{I} + 2\sum_{\ell=1}^{k} p_\ell \mathbf{w}_\ell \mathbf{w}_\ell^{\top}$ for $\widehat{\beta}_{i,j}$ in Algorithm 2 line 2, we can

---

**Algorithm 2** Robust subspace estimation

---

1: **Input:** Data $\mathcal{D}_{L1} = \{\{(\mathbf{x}_{i,j}, y_{i,j})\}_{j=1}^{t_{L1}}\}_{i=1}^{n_{L1}}$, $\alpha \in (0, 1/36]$, $\delta \in (0, 0.5)$, $k \in \mathbb{N}$, and $\nu \in \mathbb{R}_+$

2: $\widehat{\beta}_{i,j} \leftarrow y_{i,j} \mathbf{x}_{i,j}$,     for all $i \in [n_{L1}], j \in [t_{L1}]$

3: $S_0 \leftarrow \{\widehat{\beta}_{i,j} \widehat{\beta}_{i,j}^\top\}_{i \in [n_{L1}], j \in [t_{L1}]}$, and $S_{\max} \leftarrow \emptyset$

4: **for** $\ell = 1, \ldots, \log_6 (2/\delta)$ **do**

5:     $t \leftarrow 0$ and $S_{-1} \leftarrow \emptyset$

6:     **while** $t \leq \lceil 9\alpha n \rceil$ and $S_t \neq S_{t-1}$ **do**

7:       $t \leftarrow t + 1$, and $S_t \leftarrow$ Double-Filtering$(S_{t-1}, k, \alpha, \nu)$         [See Algorithm 3]

8:     **if** $|S_{\max}| < |S_t|$ **then** $S_{\max} \leftarrow S_t$

9: **Output:** $\widehat{\mathbf{U}} \leftarrow k\_\text{SVD}\big( \sum_{\widehat{\beta}_{i,j} \in S_{\max}} \widehat{\beta}_{i,j} \widehat{\beta}_{i,j}^\top \big)$

---

use the $k$ empirical principal components; this requires uncorrupted data. To remove the corrupted datapoints, we introduce *double filtering*. We repeat $\log_6(2/\delta)$ times for a high probability result.

---

**Algorithm 3** Double-Filtering

---

1: **Input:** a set of PSD matrices $S = \left\{\mathbf{X}_i \in \mathbb{R}^{d \times d}\right\}_{i \in [n]}$, $k \in \mathbb{N}$, $\alpha \in (0, 1/36]$ and $\nu \in \mathbb{R}_+$

2: $\mathcal{S}_0 \leftarrow [n]$, $\mathbf{U}_0 \leftarrow k\_\text{SVD}\big( \sum_{i \in \mathcal{S}_0} \mathbf{X}_i \big)$, and $z_i \leftarrow \text{Tr}\big[\mathbf{U}_0^\top \mathbf{X}_i \mathbf{U}_0\big]$ for all $i \in \mathcal{S}_0$

3: $\mathcal{S}_G \leftarrow$ First-Filter$(\{z_i\}_{i \in \mathcal{S}_0}, \alpha)$         [Remove the upper and lower $2\alpha$ quantiles]

4: $\mu^{\mathcal{S}_0} \leftarrow (1/n) \sum_{i \in \mathcal{S}_0} z_i$ and $\mu^{\mathcal{S}_G} \leftarrow (1/|\mathcal{S}_G|) \sum_{i \in \mathcal{S}_G} z_i$

5: **if** $\mu^{\mathcal{S}_0} - \mu^{\mathcal{S}_G} \leq 48(\alpha \mu^{\mathcal{S}_G} + \nu\sqrt{k\alpha})$ **then Output:** $S$ [Sample mean not large, no need to filter.]

6: **else**                        [Run a second filter if sample mean is corrupted]

7:     $Z \sim \mathcal{U}[0, 1]$,    $W \leftarrow Z \max\{z_i - \mu^{\mathcal{S}_G}\}_{i \in \mathcal{S}_0 \setminus \mathcal{S}_G}$

8:     $\mathcal{S}_1 \leftarrow \mathcal{S}_G \cup \left\{i \in \mathcal{S}_0 \setminus \mathcal{S}_G \mid z_i - \mu^{\mathcal{S}_G} \leq W\right\}$        [Add some removed points back.]

9: **Output:** $S' = \{\mathbf{X}_i\}_{i \in \mathcal{S}_1}$

---

If the adversarial examples have the outer product $\mathbf{X}_i = \widehat{\beta}_{i',j'} \widehat{\beta}_{i',j'}^\top$'s with small norms, then they are challenging to detect. However, such undetectable corruptions can only perturb the subspace by little. Hence, Algorithm 3 focuses on detecting large corruptions. Ideally, we want to find a subspace by

$$\widehat{\mathbf{U}} \leftarrow \underset{\mathbf{U} \in \mathbb{R}^{d \times k}: \mathbf{U}^\top \mathbf{U} = \mathbf{I}_k}{\arg\max} \quad \underset{\mathcal{S}' \subseteq [n]: |\mathcal{S}'| \geq (1-\alpha)n}{\text{minimize}} \quad \sum_{i \in \mathcal{S}'} \underbrace{\text{Tr}[\mathbf{U}^\top \mathbf{X}_i \mathbf{U}]}_{:= z_i} \,,$$

for $n = n_{L1} t_{L1}$, which is computationally intractable. This relies on the intuition that a good subspace preserves the second moment, even when large (potentially corrupted) points are removed.

We propose a *filtering* approach in Algorithm 3. At each iteration, we alternate between finding a candidate semi-orthogonal matrix $\mathbf{U}_0 \in \mathbb{R}^{d \times k}$ containing the top-$k$ singular vectors using the $k\_\text{SVD}$ routine and then filtering out suspected corrupted data points, which have large trace norms in $\mathbf{U}_0$. Existing filtering approaches (e.g. [75]) use a *single filter* to remove examples with large trace norm (denoted by $z_i$ in Algorithm 3). This suffers from removing too many *uncorrupted* examples. We give a precise comparison in Eq. (6). We instead use two filters to add back some of those mistakenly removed points. The First-Filter partitions the input set into a good set $\mathcal{S}_G$ and a bad set $\mathcal{S}_0 \setminus \mathcal{S}_G$. If the bad set contributed to a significant portion of the projected trace (this can be detected by the shift in the mean of the remaining points $\mu^{\mathcal{S}_G}$), a second filter is applied to the bad set, recovering some of the uncorrupted examples.

This algorithm and our analysis applies more generally to any random vector, and may be of independent interest in other applications requiring robust PCA. Under a mild assumption that $\mathbf{x}_i \sim \mathcal{P}$ has a bounded fourth-moment, we prove the following, with a proof in §D.1.

**Proposition 2.6** (Robust PCA for general PSD matrices). *Let* $S = \{\mathbf{x}_i \sim \mathcal{P}\}_{i=1}^n$ *where* $\Sigma := \mathbb{E}_{\mathbf{x} \sim \mathcal{P}}[\mathbf{x}\mathbf{x}^\top]$ *is the second moment of* $\mathcal{P}$ *supported on* $\mathbb{R}^d$. *Given* $k \in \mathbb{N}$, $\delta \in (0, 0.5)$, *and a corrupted dataset* $S'$ *with* $\alpha \in (0, 1/36]$ *fraction corrupted arbitrarily, if* $\mathcal{P}$ *has a bounded support such that* $\|\mathbf{x}\mathbf{x}^\top - \Sigma\|_2 \leq B$ *for* $\mathbf{x} \sim \mathcal{P}$ *with probability one, and a bounded 4-th moment such that* $\max_{\|\mathbf{A}\|_F \leq 1, \text{rank}(\mathbf{A}) \leq k} \mathbb{E}_{\mathbf{x} \sim \mathcal{P}}\left[\left(\langle \mathbf{A}, \mathbf{x}\mathbf{x}^\top - \Sigma \rangle\right)^2\right] \leq \nu^2$, *and* $n = \Omega((dk^2 +$

$(B/\nu)\sqrt{k\alpha})\log(d/(\delta\alpha))/\alpha)$, *then with probability at least* $1 - \delta$,

$$\text{Tr}[\mathcal{P}_k(\boldsymbol{\Sigma})] - \text{Tr}\left[\widehat{\mathbf{U}}^\top \boldsymbol{\Sigma}\widehat{\mathbf{U}}\right] = \mathcal{O}\left(\alpha\,\text{Tr}[\mathcal{P}_k(\boldsymbol{\Sigma})] + \nu\sqrt{k\alpha}\right), \tag{4}$$

$$\text{and} \quad \left\|\boldsymbol{\Sigma} - \widehat{\mathbf{U}}\widehat{\mathbf{U}}^\top\boldsymbol{\Sigma}\widehat{\mathbf{U}}\widehat{\mathbf{U}}^\top\right\|_* \leq \|\boldsymbol{\Sigma} - \mathcal{P}_k(\boldsymbol{\Sigma})\|_* + \mathcal{O}\left(\alpha\|\mathcal{P}_k(\boldsymbol{\Sigma})\|_* + \nu\sqrt{k\alpha}\right). \tag{5}$$

*where* $\widehat{\mathbf{U}}$ *is the output of Algorithm 2, and* $\mathcal{P}_k(\cdot)$ *is the best rank-$k$ approximation of a matrix in* $\ell_2$.

The first term in the RHS of Eq. (5) is unavoidable, as we are outputting a rank-$k$ subspace. In the setting of Theorem 1 in which we are interested in, the last term of $\nu\sqrt{k\alpha}$ in Equation (4) dominates the second term. We next show that this cannot be improved upon; no algorithm can learn the subspace with an additive error smaller than $\Omega(\nu\sqrt{k\alpha})$ under $\alpha$ fraction of corruption, even with infinite samples. In the following minimax lower bound, since the total variation distance $D_{\text{TV}}(\mathcal{P}, \mathcal{P}') \leq \alpha$, the adversary can corrupted the datapoints from $\mathcal{P}'$ to match the distribution $\mathcal{P}$, by changing just the $\alpha$ fraction. It is impossible to tell if the corrupted samples came from $\mathcal{P}$ or $\mathcal{P}'$, resulting in an $\mathcal{O}(\nu\sqrt{k\alpha})$ error.

**Proposition 2.7** (Information theoretic lower bound). *Let* $\widehat{\mathbf{U}}(\{\mathbf{x}_i\}_{i=1}^n)$ *be any subspace estimator that takes $n$ samples from distribution $\mathcal{P}$ as input, and estimates the $k$ principal components of* $\boldsymbol{\Sigma} := \mathbb{E}_{\mathbf{x}\sim\mathcal{P}'}\left[\mathbf{x}\mathbf{x}^\top\right]$ *from another distribution $\mathcal{P}'$ that is $\alpha$-close in total variation $D_{\text{TV}}$. Then,*

$$\inf_{\widehat{\mathbf{U}}} \max_{\mathcal{P}'\in\Theta_{\nu,B}} \max_{\mathcal{P}:D_{\text{TV}}(\mathcal{P},\mathcal{P}')\leq\alpha} \mathbb{E}_{\{\mathbf{x}_i\}_{i=1}^n\sim\mathcal{P}^n}\left[\left\|\boldsymbol{\Sigma} - \widehat{\mathbf{U}}\widehat{\mathbf{U}}^\top\boldsymbol{\Sigma}\widehat{\mathbf{U}}\widehat{\mathbf{U}}^\top\right\|_* - \|\boldsymbol{\Sigma} - \mathcal{P}_k(\boldsymbol{\Sigma})\|_*\right] = \Omega(\nu\sqrt{k\alpha}),$$

*for any* $k \geq 16, d \geq k^2/\alpha$, *and* $B \geq 2d\nu$, *where* $\Theta_{\nu,B}$ *is a set of all distributions $\mathcal{D}'$ on $\mathbb{R}^d$ such that* $\max_{\|\mathbf{A}\|_\text{F}\leq 1} \mathbb{E}_{\mathbf{x}\sim\mathcal{D}'}\left[\left(\langle\mathbf{A}, \mathbf{x}\mathbf{x}^\top - \boldsymbol{\Sigma}\rangle\right)^2\right] \leq \nu^2$, *and* $\mathbb{P}_{\mathbf{x}\sim\mathcal{D}'}\left[\left\|\mathbf{x}\mathbf{x}^\top - \mathbb{E}\left[\mathbf{x}\mathbf{x}^\top\right]\right\|_2 \leq B\right] = 1$.

**Comparisons with [75].** Outlier-Robust Principal Component Analysis (ORPCA) [75, 28, 76] studies a similar problem under a Gaussian model. For comparison, we can modify the best known ORPCA estimator from [75] to our setting in Proposition 2.6, to get a semi-orthogonal $\widehat{\mathbf{U}}$ achieving

$$\left\|\boldsymbol{\Sigma} - \widehat{\mathbf{U}}\widehat{\mathbf{U}}^\top\boldsymbol{\Sigma}\widehat{\mathbf{U}}\widehat{\mathbf{U}}^\top\right\|_* = \|\boldsymbol{\Sigma} - \mathcal{P}_k(\boldsymbol{\Sigma})\|_* + \mathcal{O}\left(\alpha^{1/2}\|\mathcal{P}_k(\boldsymbol{\Sigma})\|_* + \nu k\alpha^{1/4}\right). \tag{6}$$

We significantly improve in the dominant third term (see Eq. (5)). Simulation results supporting our theoretical prediction are shown in Fig. 2. For the analysis and the experimental setup we refer to §K.

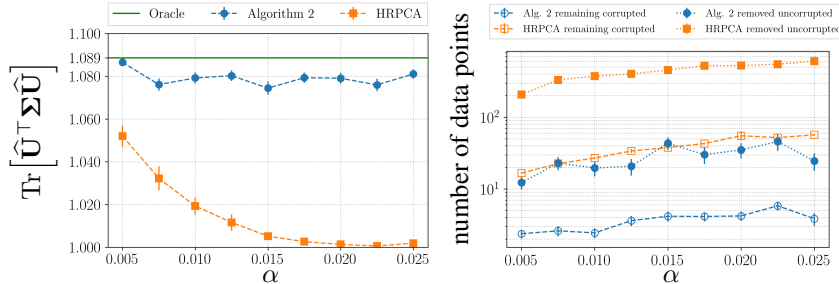

Figure 2: Algorithm 2 performs close to an oracle which knows the corrupted points, improving upon HRPCA of [75], by removing more corrupted points and less uncorrupted ones.

## 3 Related work

**Mixed linear regression.** Previous work on mixed linear regression focus on the setting where each task has only one sample, i.e. $t_i = 1$. As a consequence, all the previous work suffer from either the sample complexity or the running time that scale exponentially in $k$ (specifically at least $\exp(\sqrt{k})$) [78, 47, 16, 64]. In other cases, such blow-up in complexity is hidden in the dependence of the inverse of $k$-th singular value of a moment matrix, which can be arbitrarily large [14, 77, 63].

**Multi-task learning.** [8, 3, 60, 53, 4, 32, 2, 54, 7, 11] address a similar problem as our setting, but focusing on finding a common low-dimensional linear representation, where all tasks can be

accurately solved. Typically, the batch size is fixed and the performance is evaluated on the past tasks used in training. Close to ours are a few concurrent work [27, 69], but their focus is still on recovering the common subspace, and not the meta-parameters.

**Robust regression.** There are several work on robust linear regression and sparse regression problems, [10, 9, 6, 30, 55, 40, 23, 48, 39, 21, 51, 15, 57, 38]. The recent advances in the list-decodable setting [15, 57, 38] can potentially be applied to our mixture setting, but the sample complexity is exponentially large. Recently, [64] studies the robust mixed linear regression problem. In contrast to our setting which allows random noise on the label and adversarial corruption on both covariate $\mathbf{x}$ and label $y$, their setting assumes no noise on label $y$'s, and the adversary is only allowed to corrupt $\alpha$-fraction the label $y$'s. Although their algorithm requires only $\widetilde{\mathcal{O}}(dk)$ samples, the running time is $\widetilde{\mathcal{O}}(k^k nd)$ and also requires a good estimate of the subspace spanned by $\{\mathbf{w}_\ell\}_{\ell=1}^k$.

**Sum-of-squares algorithms.** Our work is inspired by sum-of-squares algorithms that have recently been studied on many learning problems, including linear regression [38, 57], mixture models [35, 46, 37, 56, 22], mean estimation [34, 20], subspace estimation [5]. This provides the key building block of our approach, in breaking the second moment barrier of linkage-based clustering algorithms.

# 4 Conclusion

By exploiting similarities on a collection of related but different tasks, meta-learning predicts a newly arriving task with a far greater accuracy than what can be achieved in isolation. We ask two fundamental questions under a canonical model of $k$-mixed linear regression: $(i)$ can we meta-learn from tasks with only a few training examples each?; and $(ii)$ can we meta-learn from tasks when only part of the data can be trusted? We introduce a novel spectral approach that achieves both simultaneously, significantly improving the required batch size from $\Omega(k^{1/2})$ to $\Omega(\log k)$ while being robust to adversarial corruption. We use a sum-of-squares algorithm to exploit the higher order moments and design a novel robust subspace estimation algorithm that achieves optimal guarantees.

**Closing the gap in robust subspace estimation.** [75, 28, 76, 19] study robust PCA under the Gaussian assumption. For the reasons explained in §2.2, the rate is sub-optimal in $\alpha$ in comparisons to an information theoretic lower bound with a multiplicative factor of $(1 - \Theta(\alpha))$. Applying the proposed Algorithm 2, it is possible to generalize Proposition 2.6 to this Gaussian setting and achieve an optimal upper bound. We leave this as a future research direction, and provide a sketch of how to adapt the proof of our algorithm to the exponential tail setting in §E.

Concretely, our analysis of Algorithm 2 assumes only a bounded 4-th moment of the input vector $\mathbf{z}_i$, of the form $\mathbb{P}\big[\big|(\mathbf{v}^\top \mathbf{z}_i)^2 - \mathbf{v}^\top \Sigma \mathbf{v}\big| \geq t\big] \leq c\, t^{-2}$. Our current proof proceeds by focusing on that $1 - \alpha$ probability mass, which falls in the interval $[-\sqrt{1/\alpha}, \sqrt{1/\alpha}]$. This is tight with only the second moment assumption. More generally, one can consider a family of distributions satisfying $\mathbb{P}\big[\big|(\mathbf{v}^\top \mathbf{z}_i)^2 - \mathbf{v}^\top \Sigma \mathbf{v}\big| \geq \text{variance} \cdot t\big] \leq \exp(-t^\gamma)$. If we have such an exponential concentration, we can instead focus on the subset of examples with second moment $\big|(\mathbf{v}^\top \mathbf{z}_i)^2 - \mathbf{v}^\top \Sigma \mathbf{v}\big|$ falling in the interval $[-\log^{1/\gamma}(1/\alpha), \log^{1/\gamma}(1/\alpha)]$. This bounded distribution has a sub-Gaussian norm $\sqrt{k}\log^{1/\gamma}(1/\alpha)$, and thus we can apply the sub-Gaussian filter (Proposition A.7 of [24]) to learn $\mathbb{E}\big[(\mathbf{v}^\top \mathbf{z}_i)^2\big]$ with error We can obtain an error of $\alpha\sqrt{k}\log^{1/\gamma}(1/\alpha)$. We provide a sketch of how to adapt the proof of our algorithm to the exponential tail setting in §E.

After submission, we became aware of an independent and concurrent result by Jambulapati et al. [36] which studies the robust PCA problem under the assumption that each datapoint $\mathbf{x}$ follows a sub-Gaussian distribution. Their algorithm is very similar to ours, except that it is only applied to estimating the top eigenvector of the covariance matrix, which corresponds to the $k = 1$ special case in our setting. Their sample complexity and recovery guarantee are identical to ours in §E.

**Removing the Gaussianity assumption.** Our approach relies on the special structure of the 4-th moment of $\mathbf{x}_{i,j}$ and the SOS boundedness of higher order moments of $\mathbf{x}_{i,j}$. The approach in [42] is able to get around the 4-th moment requirement, and it is an interesting open problem to make the approach robust to outliers and still preserve the $\widetilde{\mathcal{O}}(d)$ sample complexity. while this class of SOS bounded distributions is fairly broad, as noted in [45], one could hope to establish sum-of-squares bounds for even broader families. For examples, it remains open that whether sum-of-squares certifies moment tensors for all sub-Gaussian distributions.

## Broader Impact

One of the main contribution of this paper is to protect meta-learning approaches against data poisoning attacks. Such robustness encourages participation from data contributors, as they can collaborate without necessarily trusting the other data contributors. This facilitates participation of minor contributors who suffer from data scarcity. This fosters democratization of machine learning by allowing minor contributors to enjoy the benefit of big data through collaboration. Such ecosystem will also encourage data sharing, thus improving transparency.

The adaptive guarantee we provide in Theorem 1 is fair, in the sense that a group that provides low noise data will receive a model with better accuracy. However, one potential risk in fairness is that meta-learning might result in varying accuracy across the groups. This can be problematic as an under-represented group in training data could suffer from inaccurate prediction for that population. This is an active area of research in the fairness community, but there is no strong experimental evidence that this can be mitigated with algorithmic innovations that do not involve collecting more data from the under-represented population.

Another concern in meta-learning with data sharing is privacy. Without proper system to regulate the usage of shared data, sensitive information could be leaked or protected features could be inferred. One silver lining is that robust methods are naturally private, as the trained model is by definition not sensitive to any one particular data point. On the other hand, if the system relies on the participation of various individuals, then either a technological solution needs to be implemented with cryptographic or privacy preserving primitives, or a proper regulation must be enforced.

## Acknowledgments and Disclosure of Funding

Sham Kakade acknowledges funding from the NSF Awards CCF-1637360, CCF-1703574, and CCF-1740551. Sewoong Oh acknowledges funding from Google Faculty Research Award, and NSF awards CCF-1927712, CCF-1705007, IIS-1929955, CNS-2002664, and 2019844.

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
