[Supplementary Material]

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

# Appendix

## A    Proof of Corollary 1.2, Corollary 1.3, Corollary 1.1

---

**Algorithm 4** Meta-learning without adversarial corruptions

---

**Meta-learning**

1. *Subspace estimation:* Compute subspace $\widehat{\mathbf{U}}$ with [42, Algorithm 2] which approximates $\mathrm{span}\{\mathbf{w}_1, \ldots, \mathbf{w}_k\}$.
2. *Clustering:* Project the heavy tasks onto the subspace of $\widehat{\mathbf{U}}$, perform $k$ clustering with Algorithm 7, and estimate $\widetilde{\mathbf{w}}_\ell$. Also estimate $\widetilde{r}_\ell^2$ using Algorithm 8 for each cluster $\ell \in [k]$.
3. *Classification:* Perform likelihood-based classification of the light tasks using $\{\widetilde{\mathbf{w}}_\ell, \widetilde{r}_\ell^2\}_{\ell=1}^k$ estimated from the *Clustering* step; compute refined estimates $\{\widehat{\mathbf{w}}_\ell, \widehat{s}_\ell, \widehat{p}_\ell\}_{\ell=1}^k$ of $\theta$ using [42, Algorithm 4].

**Prediction**

4. *Prediction:* Perform MAP or Bayes optimal prediction using the estimated meta-parameter.

---

We assume that the meta-parameter satisfies $\Delta = \Theta(1)$, $\rho = \Theta(1)$, $p_{\min} = \Theta(1/k)$ for Corollary 1.2, Corollary 1.3, Corollary 1.1.

*Proof of Corollary 1.1.* Recall that for this corollary, we assume $\Delta = \Theta(1)$, $\rho = \Theta(1)$, $p_{\min} = \Theta(1/k)$ and there is no adversarial corruption. Thus we execute Algorithm 4 in this setting. We invoke [42, Lemma 5.1] and get that given $t = \Omega(1)$ and $tn = \widetilde{\Omega}(kn)$, the estimated subspace $\widehat{\mathbf{U}}$ satisfies that for all $i \in [k]$,

$$\left\| \left( \mathbf{I} - \widehat{\mathbf{U}}\widehat{\mathbf{U}}^\top \right) \mathbf{w}_\ell \right\|_2 \leq \Delta/(10) \quad \forall\, \ell \in [k] \,, \tag{7}$$

Then, we can invoke Lemma B.4 with $m = \Theta(\log k)$, $t = \Theta(\log k)$, $n = k^{\Theta(\log k)}$ and get that the estimated parameters $\widetilde{\mathbf{w}}_\ell, \widetilde{r}_\ell^2$ satisfy

$$\|\widetilde{\mathbf{w}}_\ell - \mathbf{w}_\ell\|_2 \leq \frac{\Delta}{10} \,, \qquad \text{and} \qquad \left| \widetilde{r}_\ell^2 - r_\ell^2 \right| \leq r_\ell^2 \frac{\Delta^2}{50\rho^2} \,, \qquad \forall\, \ell \in [k] \,,$$

Finally, given $t = \widetilde{\Omega}(1)$ and $n = \Omega(dk)$, the output of the classification step satisfies

$$\|\widehat{\mathbf{w}}_\ell - \mathbf{w}_\ell\|_2 = \mathcal{O}(1) \,, \quad \left| \widehat{s}_\ell^2 - s_\ell^2 \right| = \mathcal{O}\left( s_\ell^2 \right) \,, \quad \text{and} \quad |\widehat{p}_\ell - p_\ell| = \mathcal{O}(p_\ell) \quad \forall\, \ell \in [k]. \tag{8}$$

To conclude, with $t = \widetilde{\Omega}(1), n = \widetilde{\Omega}(dk^2 + k^{\Theta(\log k)})$, Algorithm 4 can estimate model parameters $\theta$ with arbitrary small constant error. $\qquad\square$

*Proof of Corollary 1.2.* The proof is the same as Corollary 1.2. $\qquad\square$

*Proof of Corollary 1.3.* With the assumptions that $\Delta = \Omega(1)$, $\rho = \Omega(1)$, $p_{\min} = \Omega(1/k)$, $t_{L1} = t_{L2} = \widetilde{\Omega}(1)$, $t_H = \Omega(mk^{1/m})$ Theorem 1 can be simplified to that for all $i \in [k]$, with probability $1 - \delta$

$$\|\widehat{\mathbf{w}}_i - \mathbf{w}_i\|_2 \leq \epsilon s_i \,, \quad \left| \widehat{s}_i^2 - s_i^2 \right| \leq \epsilon s_i^2 \,, \quad \text{and} \quad |\widehat{p}_i - p_i| \leq \epsilon p_i \, + \, \alpha_{L2} \,,$$

as long as,

$$\begin{aligned}
n_{L1} &= \widetilde{\Omega}\left( \tfrac{dk^2}{\widetilde{\alpha}} \right) \,, & \alpha_{L1} &= \mathcal{O}(\widetilde{\alpha}) \,, \\
n_H &= \widetilde{\Omega}\left( (km)^{\Theta(m)} \right) \,, & \alpha_H &= \widetilde{\mathcal{O}}(1/k) \,, \\
n_{L2} &= \widetilde{\Omega}\left( \tfrac{dk}{\epsilon^2} \right) \,, & \alpha_{L2} &= \widetilde{\mathcal{O}}(\epsilon/k) \,,
\end{aligned}$$

where $\widetilde{\alpha} := 1/k^4$. Using the assumption that $\epsilon \leq 1/k^3$. this implies that as long as

$$\begin{aligned}
n_{L1} &= \widetilde{\Omega}\left( \tfrac{dk}{\epsilon^2} \right) \,, & \alpha_{L1} &= \mathcal{O}(\epsilon/k) \,, \\
n_H &= \widetilde{\Omega}\left( (km)^{\Theta(m)} \right) \,, & \alpha_H &= \widetilde{\mathcal{O}}(1/k) \,, \\
n_{L2} &= \widetilde{\Omega}\left( \tfrac{dk}{\epsilon^2} \right) \,, & \alpha_{L2} &= \widetilde{\mathcal{O}}(\epsilon/k) \,,
\end{aligned}$$

Our algorithm can estimate the parameters up to error

$$\|\widehat{\mathbf{w}}_i - \mathbf{w}_i\|_2 \leq \epsilon s_i \,, \quad \left|\widehat{s}_i^2 - s_i^2\right| \leq \epsilon s_i^2 \,, \quad \text{and} \quad |\widehat{p}_i - p_i| \leq \epsilon/k \,,$$

for all $i \in [k]$. $\qquad\qquad\square$

## B  Details of the algorithm and the analyses

We explain and analyze each step in Algorithm 1, which imply our main result, as shown in §C.

### B.1  Robust Subspace estimation

Building upon the tight guarantees of Proposition 2.6, Algorithm 2 achieves the following guarantee, first when $t_{L1} = 1$. The conditions depend on the ground truths meta-parameters. With $\widetilde{\mathcal{O}}(d)$ samples, we can tolerate up to $\mathcal{O}(\epsilon^2 p_{\min}^2/k^2)$ corruption, in an ideal case when $\mathbf{W} \in \mathbb{R}^{d \times k}$ is a semi-orthogonal matrix. For the worst case $\mathbf{W}$, it is $\mathcal{O}(\epsilon^6 p_{\min}^2/k^2)$.

**Lemma B.1** (Learning the subspace). *Under Assumptions 1 and 2, for any target probability* $\delta \in (0, 0.5)$, *and* $\epsilon > 0$, *if* $t_{L1} = 1$,

$$n_{L1} \;=\; \widetilde{\Omega}\left( dk^2 \min\left\{ \frac{\rho^4 k^2}{\epsilon^2 \sigma_{\min}^2}, \frac{k^2}{\epsilon^6 p_{\min}^2} \right\} \right) \,, \quad and \quad \alpha_{L1} \;=\; \mathcal{O}\left( \max\left\{ \frac{\epsilon^2 \sigma_{\min}^2}{\rho^4 k^2}, \frac{\epsilon^6 p_{\min}^2}{k^2} \right\} \right) \,, \quad (9)$$

*then the semi-orthogonal matrix* $\widehat{\mathbf{U}} \in \mathbb{R}^{d \times k}$ *from robust subspace estimation in Algorithm 2 achieves*

$$\left\| \left( \mathbf{I} - \widehat{\mathbf{U}}\widehat{\mathbf{U}}^\top \right) \mathbf{w}_i \right\|_2 \;\leq\; \epsilon\rho \,, \tag{10}$$

*with probability at least* $1 - \delta$, *where* $\sigma_{\min}$ *is the minimum singular value of* $\sum_{j=1}^{k} p_j \mathbf{w}_j \mathbf{w}_j^\top$.

We provide a proof in § D. When $t_{L1} > 1$, we get the following sufficient condition. The dominant first term requires the effective sample size $n_{L1} t_{L1} = \widetilde{\Omega}(d \operatorname{poly}(k))$, which is linear in $d$.

**Remark B.2** (Handling $t_{L1} > 1$). *Lemma B.1 can be naturally generalized to the case where* $1 < t_{L1} < d$, *and the requirement on* $n_{L1}$ *is*

$$n_{L1} \;=\; \widetilde{\Omega}\left( \frac{dk^2}{t_{L1}} \min\left\{ \frac{\rho^4 k^2}{\epsilon^2 \sigma_{\min}^2}, \frac{k^2}{\epsilon^6 p_{\min}^2} \right\} + k \min\left\{ \frac{\rho^8 k^4}{\epsilon^4 \sigma_{\min}^4}, \frac{k^4}{\epsilon^{12} p_{\min}^4} \right\} \right) \,, \tag{11}$$

**Time complexity:** $\mathcal{O}(n_{L1} t_{L1} d)$ for computing $\widehat{\beta}_{i,j}$'s and $\mathcal{O}(n_{L1}^2 t_{L1}^2 dk\alpha \log(1/\delta))$ for the filtering algorithm which uses k_SVD [1]. The running time is from the fact that Double-Filtering (Algorithm 3) executes at most $\mathcal{O}(n_{L1} t_{L1} \alpha)$ times, and the running time of each execution is dominated by k_SVD which takes $\mathcal{O}(n_{L1} t_{L1} dk)$ time.

**Remark B.3** (Gaussianity assumptions). *Although our robust PCA algorithm 2 only requires bounded fourth moment assumption, our robust subspace estimation succeeds with the fact that*

$$\mathbb{E}[y_{i,j}^2 \mathbf{x}_{i,j} \mathbf{x}_{i,j}^\top] = c\mathbf{I} + 2 \sum_{\ell=1}^{k} p_\ell \mathbf{w}_\ell \mathbf{w}_\ell^\top$$

*for some constant* $c \geq 0$. *This depends on the fourth moment property of Gaussian (Fact J.7). [42] adopts a different approach by taking*

$$\mathbb{E}[y_{i,j} y_{i,j'} \mathbf{x}_{i,j} \mathbf{x}_{i,j'}^\top] = \sum_{\ell=1}^{k} p_\ell \mathbf{w}_\ell \mathbf{w}_\ell^\top$$

*for* $j \neq j'$, *which is able to handle general sub-Gaussian* $\mathbf{x}$. *While it is possible to make the approach in [42] robust to outliers with robust mean estimation techniques (Remark K.1), such approaches requires an sub-optimal* $\Omega(d^2)$ *samples complexity. How to make the approach in [42] robust with only linear dependency in* $d$ *remains an open problem, and the key obstacle is that the random matrix* $y_{i,j} y_{i,j'} \mathbf{x}_{i,j} \mathbf{x}_{i,j'}^\top$ *is not PSD.*

## B.2 Robust Clustering

Once we have the subspace, we use the Sum-of-Squares (SOS) algorithm of [45] to cluster the $k$-dimensional points $\widehat{\mathbf{U}}^\top \widehat{\beta}_i$'s where $\widehat{\beta}_i = (1/t_H) \sum_{j \in [t_H]} y_{i,j} \mathbf{x}_{i,j}$. For a value of $m$ as discussed in Remark H.11, we can exploit the $m$-th order moment to filter out corrupted points and recover the clusters. This allows us to gracefully trade off $t_H$ and $n_H$, breaking the barrier of $t = \Omega(k^{1/2})$ in [42]. Further, this approach is robust against adversarial corruption up to a $\mathcal{O}(p_{\min})$ fraction of the data. We explicitly write the algorithm in §G and provide a proof in §H.

Previous work [42] proposed a linkage based clustering algorithm that is able to correctly cluster $\widehat{\beta}_i$'s as long as $t_H = \Omega(\sqrt{k})$ and the second moment is bounded. However, the algorithm fails when $t_H = o(\sqrt{k})$, and it has been noticed in [42] that the failure of such kind of algorithms is inherent since it only relies some boundedness condition on the second moments. It is natural to ask whether it is possible to exploit the boundedness of higher order moments (larger than 2) of the distribution to obtain stronger clustering results. Assuming boundedness of higher order moments is not too restrictive, since typical distributional assumptions, e.g. sub-Gaussianity, often imply boundedness for arbitrarily high order moments.

It turns out in order to *efficiently* exploit the higher order moments assumptions for clustering, one need slight stronger condition than boundedness, that is the moments are sum-of-squares bounded, meaning there *exist* sum-of-squares proofs showing that the moments are bounded [45, 35]. It is also shown in [45] that a Poincaré distribution has sum-of-squares bounded moments, and thus their algorithm can be applied to clustering any Poincaré distributions.

It turns out that in our model, even assuming that $\mathbf{x}_{i,j}$ follows from isotropic Gaussian distribution, the distribution of $\widehat{\beta}_i$ is not Poincaré and thus preventing us from applying the result in [45] directly. Interestingly, as we showed in Lemma H.2, though $\widehat{\beta}_i$ is not Poincaré, the high order moments of $\widehat{\beta}_i$ is still sum-of-squares bounded under Gaussianity assumption of $\mathbf{x}_{i,j}$, and thus we can apply the result of [45] to efficiently cluster $\widehat{\beta}_i$'s, with guarantees formalized in Lemma B.4.

**Lemma B.4** (Clustering and initial parameter estimation). *Under Assumptions 1 and 2, if $\alpha_H$ fraction of the tasks are adversarially corrupted in $\mathcal{D}_H$, and given a semi-orthogonal matrix $\widehat{\mathbf{U}}$ satisfying Eq. (10) with $\epsilon = \mathcal{O}(\Delta/\rho)$, Algorithm 7 with a choice of $m \in \mathbb{N}$, and Algorithm 8 outputs $\left\{ \widetilde{\mathbf{w}}_\ell \in \mathbb{R}^d, \widetilde{r}_\ell^2 \right\}_{j \in [k]}$ satisfying*

$$\|\widetilde{\mathbf{w}}_\ell - \mathbf{w}_\ell\|_2 \leq \frac{\Delta}{10} , \qquad \text{and} \qquad \left| \widetilde{r}_\ell^2 - r_\ell^2 \right| \leq r_j^2 \frac{\Delta^2}{50\rho^2} , \qquad \forall\, \ell \in [k] , \qquad (12)$$

*with probability at least $1 - \delta$, where $\widetilde{r}_\ell^2$ is the robust estimate of $r_\ell^2 := \|\widetilde{\mathbf{w}}_\ell - \mathbf{w}_\ell\|_2^2 + s_\ell^2$, if*

$$\alpha_H < \min\left\{ \frac{p_{\min}}{16}, \frac{C\Delta^2 \sqrt{t_H} p_{\min}}{\rho^2 \log\left(\rho^2/\Delta^2 t_H\right)} \right\}, \quad n_H = \widetilde{\Omega}\left( \frac{(km)^{\Theta(m)}}{p_{\min}} + \frac{\rho^4}{\Delta^4 t_H p_{\min}} \right), \qquad (13)$$

*and $t_H = \Omega(m\rho^2/(p_{\min}^{2/m} \Delta^2))$ for some universal constant $C > 0$.*

Assuming that $p_{\min} = 1/k$, we can therefore get the range of $m$, such that the condition on $t_H$ holds. Which is when

$$\frac{2 \log k}{-W_{-1}\left( -\frac{2c\rho^2 \log k}{t_H \Delta^2} \right)} \leq m \leq \frac{2 \log k}{-W_0\left( -\frac{2c\rho^2 \log k}{t_H \Delta^2} \right)}$$

for some $c > 0$, where $W_0$ and $W_{-1}$ are the Lambert W function, only if $t_H \geq 2ec\rho^2 \Delta^{-2} \log k$.

**Time complexity:** The robust clustering algorithm runs in time $\mathcal{O}\left( (n_H\, k)^{\mathcal{O}(m)} \log(1/\delta) \right)$.

**Remark B.5** (Gaussianity assumption). *The only distributional assumption required for the clustering algorithm is that $y_{i,j} \mathbf{x}_{i,j}$ is SOS bounded. It is noted in [46] that this is a general assumption, and the algorithm can be applied much more broadly than Gaussian.*

## B.3 Classification and robust estimation

Once the $\mathbf{w}_\ell$'s are estimated from the robust clustering step, we can efficiently classify any light task with $t_{L2} = \Omega(\log(kn_{L2}))$. After classification, we use a robust linear regression method [26] on

each group, which can tolerate up to $\mathcal{O}(p_{\min})$ fraction of corruption. It is critical that we separate the role of heavy and light tasks as initial estimation (robust clustering) and refinement (classification and robust estimation). This allows us to have abundant $n_{L2}p_{\min} = \widetilde{\Omega}(d/t_{L2})$ light tasks compensate for scarce $n_H p_{\min} = \Omega(\text{poly}(k))$ heavy tasks. We provide the algorithm and a proof in § I.

**Lemma B.6** (Refined parameter estimation via classification)**.** *Under Assumptions 1 and 2 with $\alpha_{L2}$ fraction of the tasks are adversarially corrupted in $\mathcal{D}_{L2}$, and given estimated parameters $\widetilde{\mathbf{w}}_i, \widetilde{r}_i$ satisfying Eq. (12), with probability $1 - \delta$, for any accuracy $\epsilon > 0$, if $t_{L2} = \Omega(\rho^4 \log(kn_{L2}/\delta)/\Delta^4)$,*

$$\alpha_{L2} = \mathcal{O}(p_{\min}\epsilon \log^{-1}(1/\epsilon)), \quad and \quad n_{L2} = \widetilde{\Omega}(dt_{L2}^{-1}p_{\min}^{-1}\epsilon^{-2}), \tag{14}$$

*then Algorithm 9 outputs estimated parameters $\{\widehat{\mathbf{w}}_i\}_{i=1}^k, \{\widehat{s}_i\}_{i=1}^k, \{\widehat{p}_i\}_{i=1}^k$ such that for all $i \in [k]$,*

$$\|\widehat{\mathbf{w}}_i - \mathbf{w}_i\|_2 \le \epsilon s_i, \quad |\widehat{s}_i^2 - s_i^2| \le \epsilon s_i^2/\sqrt{t_{L2}}, \quad and \quad |\widehat{p}_i - p_i| \le \epsilon p_i \sqrt{t_{L2}/d} + \alpha_{L2}. \tag{15}$$

**Time complexity:** $\mathcal{O}(n_{L2}^2 t_{L2}^2 d)$. The running time of Algorithm 9 is dominated by the robust linear regression procedure ( [26, Algorithm 2]), which takes at most $n_{L2}t_{L2}\alpha$ iterations and $\mathcal{O}(n_{L2}t_{L2}d)$ time for SVD per iteration.

## C  Proof of meta-learning in Theorem 1

Applying Lemma B.1 with $\epsilon = \Delta/(10\rho)$, we get a semi-orthogonal matrix $\widehat{\mathbf{U}}$ satisfying

$$\left\|\left(\mathbf{I} - \widehat{\mathbf{U}}\widehat{\mathbf{U}}^\top\right)\mathbf{w}_i\right\|_2 \le \Delta/10, \tag{16}$$

with $\widetilde{\alpha} := \max\{\Delta^2\sigma_{\min}^2/(\rho^6 k^2), \Delta^6 p_{\min}^2/(k^2\rho^6)\}$ if

$$n_{L1} = \widetilde{\Omega}\left(\frac{dk^2}{\widetilde{\alpha}t_{L1}} + \frac{k}{\widetilde{\alpha}^2}\right), \quad and \quad \alpha_{L1} = \mathcal{O}(\widetilde{\alpha}). \tag{17}$$

Since we have sufficiently accurate estimate of the $k$-dimensional subspace spanned by the columns of $\mathbf{W}$, we can cluster the tasks more efficiently in this lower dimensional space using robust a clustering algorithm. We use a SOS algorithm in Algorithm 7 with a choice of $m \in \mathbb{N}$ such that $t_H = \Omega\left(m\rho^2/(p_{\min}^{2/m}\Delta^2)\right)$, to get

$$\|\widetilde{\mathbf{w}}_j - \mathbf{w}_j\|_2 \le \Delta/10, \quad and \quad |\widetilde{r}_j^2 - r_j^2| \le r_j^2\Delta^2/(50\rho^2), \tag{18}$$

for all $j \in [k]$, using Lemma B.4, which requires

$$n_H = \widetilde{\Omega}\left(\frac{(km)^{\Theta(m)}}{p_{\min}} + \frac{\rho^4}{\Delta^4 p_{\min}t_H}\right), \quad and \ \alpha_H = \widetilde{\mathcal{O}}\left(p_{\min} \cdot \min\left\{1, \frac{\Delta^2\sqrt{t_H}}{\rho^2}\right\}\right). \tag{19}$$

It follows from Lemma B.6 that for any desired accuracy $\epsilon \ge (\alpha_{L2}/p_{\min}) \log(p_{\min}/\alpha_{L2})$ if

$$n_{L2} = \widetilde{\Omega}\left(\frac{d}{t_{L2}p_{\min}\epsilon^2}\right), \quad and \quad t_{L2} = \Omega(\log(kn_{L2}/\delta)/\Delta^4) \tag{20}$$

the output of our algorithm achieves

$$\|\widehat{\mathbf{w}}_i - \mathbf{w}_i\|_2 \le \epsilon s_i, \tag{21}$$

$$|\widehat{s}_i^2 - s_i^2| \le \frac{\epsilon}{\sqrt{t_{L2}}}s_i^2, \quad and \tag{22}$$

$$|\widehat{p}_i - p_i| \le \epsilon\sqrt{t_{L2}/d}\, p_i + \alpha_{L2}. \tag{23}$$

for all $i \in [k]$, as long as $\alpha_{L2} = \mathcal{O}(p_{\min}\epsilon/\log\frac{1}{\epsilon})$.

# D Proof of robust subspace estimation analysis in Lemma B.1

We first prove for the simple setting where $t_{L1} = 1$ and resolving a discrepancy in the independence when $t_{L1} > 1$ at the end of the section. Further, for notational convenience, we use $i$ in place of $(i, j)$, and $n$ for $n_{L1}t_{L1}$.

First we compute the expectation of $\widehat{\beta}_i \widehat{\beta}_i^\top$. Using Lemma K.2, we have

$$\mathbf{M} := \mathbb{E}\left[\widehat{\beta}_i \widehat{\beta}_i^\top\right] = 2 \sum_{j=1}^{k} p_j \mathbf{w}_j \mathbf{w}_j^\top + \left(\sum_{j=1}^{k} p_j \left(\|\mathbf{w}_j\|^2 + s_j^2\right)\right) \mathbf{I}_d,$$

and we define $\bar{\rho}^2 = \left(\sum_{j=1}^{k} p_j(\|\mathbf{w}_j\|^2 + s_j^2)\right)$.

Since our goal is to recover the top $k$ eigenspace of $\mathbf{M}$, we would like to apply Proposition 2.6 to $\mathbf{x}_i = \widehat{\beta}_i$, however $\widehat{\beta}_i$ does not satisfy the spectral norm bound requirement on $\|\mathbf{x}_i \mathbf{x}_i^\top - \mathbf{\Sigma}\|_2$. The following proposition shows that we can resolve this issue through conditioning on the event that $\widehat{\beta}_i$ is bounded.

**Proposition D.1.** *For any $0 < \delta \le 1/2$, define event*

$$\mathcal{E} := \left\{\left\|\widehat{\beta}_i\right\|_2 \le \rho\sqrt{d}\log(nd/\delta), \ \forall i \in [n]\right\}.$$

*Then conditioned on event $\mathcal{E}$, the distribution of $\widehat{\beta}_i$ satisfies the prerequisite of Proposition 2.6 with*

1. $\left\|\widehat{\beta}_i \widehat{\beta}_i^\top - \mathbb{E}\left[\widehat{\beta}_i \widehat{\beta}_i^\top \big| \mathcal{E}\right]\right\|_2 \le \underbrace{5\rho^2 d \log^2(nd/\delta)}_{=B}$ .

2. $\mathbb{E}\left[\text{Tr}\left[\mathbf{A}\left(\widehat{\beta}_i \widehat{\beta}_i^\top - \mathbb{E}[\widehat{\beta}_i \widehat{\beta}_i^\top \mid \mathcal{E}]\right)\right]^2 \Big| \mathcal{E}\right] \le \underbrace{\mathcal{O}\left(k\rho^4\right)}_{=\nu^2(k)}$ .

*The mean shift is bounded under $\mathcal{E}$ as:*

$$\left\|\mathbb{E}\left[\widehat{\beta}_i \widehat{\beta}_i^\top\right] - \mathbb{E}\left[\widehat{\beta}_i \widehat{\beta}_i^\top \big| \mathcal{E}\right]\right\|_2 \le \mathcal{O}\left(\rho^2 \sqrt{\delta}\right), \tag{24}$$

The proof is deferred to the end of the section.

Recall that $\mathbf{M} := \mathbb{E}\left[\widehat{\beta}_i \widehat{\beta}_i^\top\right]$, and let us define $\mathbf{M}' := \mathbb{E}\left[\widehat{\beta}_i \widehat{\beta}_i^\top \big| \mathcal{E}\right]$ be the mean conditioned on $\mathcal{E}$. With Proposition D.1, we can apply Proposition 2.6 to obtain a nuclear norm guarantee on our our subspace estimation algorithm

**Proposition D.2.** *Given that*

$$n = \widetilde{\Omega}(dk^2/\alpha),$$

*with probability $1 - 2\delta$, then Algorithm 2 returns a rank-$k$ semi-orthogonal matrix $\widehat{\mathbf{U}} \in \mathbb{R}^{d \times k}$ satisfying*

$$\left\|\widehat{\mathbf{U}}\widehat{\mathbf{U}}^\top \left(\sum_{j=1}^{k} p_j \mathbf{w}_j \mathbf{w}_j^\top\right) \widehat{\mathbf{U}}\widehat{\mathbf{U}}^\top - \sum_{j=1}^{k} p_j \mathbf{w}_j \mathbf{w}_j^\top\right\|_* = \mathcal{O}\left(\rho^2 k\sqrt{\alpha}\right)$$

*for $\delta \in (0, 0.5)$.*

*Proof of Proposition D.2.* We apply Proposition 2.6 to $\mathbf{x}_i = \widehat{\beta}_i$ conditioned on event $\mathcal{E}$ and get

$$\text{Tr}\left[\widehat{\mathbf{U}}\widehat{\mathbf{U}}^\top \mathbf{M}' \widehat{\mathbf{U}}\widehat{\mathbf{U}}^\top\right] \ge (1 - \mathcal{O}(\alpha)) \text{Tr}[\mathcal{P}_k(\mathbf{M}')] - \mathcal{O}\left(\rho^2 k\sqrt{\alpha}\right) \tag{25}$$

with probability $1 - 2\delta$, when

$$n = \widetilde{\Omega}\left(\left(dk^2 + \frac{\rho^2 d}{\sqrt{k}\rho^2} \cdot \sqrt{k\alpha}\right)/\alpha\right) = \widetilde{\Omega}(dk^2/\alpha).$$

WLOG, for the remaining analysis we will assume $\delta \leq 1/nd$. The nuclear norm term in the proposition statement can be bounded as

$$\left\|\widehat{\mathbf{U}}\widehat{\mathbf{U}}^\top(\mathbf{M} - \bar{\rho}^2\mathbf{I})\widehat{\mathbf{U}}\widehat{\mathbf{U}}^\top - (\mathbf{M} - \bar{\rho}^2\mathbf{I})\right\|_*$$

$$= \mathrm{Tr}\left[\mathbf{M} - \bar{\rho}^2\mathbf{I}\right] - \mathrm{Tr}\left[\widehat{\mathbf{U}}\widehat{\mathbf{U}}^\top(\mathbf{M} - \bar{\rho}^2\mathbf{I})\widehat{\mathbf{U}}\widehat{\mathbf{U}}^\top\right]$$

$$= \mathrm{Tr}\left[\mathbf{M} - \bar{\rho}^2\mathbf{I}\right] - \mathrm{Tr}\left[\widehat{\mathbf{U}}^\top(\mathbf{M}' - \bar{\rho}^2\mathbf{I})\widehat{\mathbf{U}}\right] - \mathrm{Tr}\left[\widehat{\mathbf{U}}^\top(\mathbf{M} - \mathbf{M}')\widehat{\mathbf{U}}\right]$$

$$\leq \mathrm{Tr}\left[\mathbf{M} - \bar{\rho}^2\mathbf{I}\right] - \mathrm{Tr}\left[\widehat{\mathbf{U}}^\top(\mathbf{M}' - \bar{\rho}^2\mathbf{I})\widehat{\mathbf{U}}\right] + \rho^2 k\sqrt{\delta} \quad \text{(Using Equation (24))}$$

$$= \mathrm{Tr}\left[\mathbf{M} - \bar{\rho}^2\mathbf{I}\right] - \mathrm{Tr}\left[\widehat{\mathbf{U}}^\top\mathbf{M}'\widehat{\mathbf{U}}\right] + k\bar{\rho}^2 + \rho^2 k\sqrt{\delta}$$

$$\leq \mathrm{Tr}\left[\mathbf{M} - \bar{\rho}^2\mathbf{I}\right] - (1 - \mathcal{O}(\alpha))\mathrm{Tr}[\mathcal{P}_k(\mathbf{M}')] + \rho^2 k\sqrt{\alpha} + k\bar{\rho}^2 + \rho^2 k\sqrt{\delta} \qquad (26)$$

$$\text{(Using Equation (25))}.$$

We need the following bound on $\mathrm{Tr}[\mathcal{P}_k(\mathbf{M}')]$ before proceeding:

$$\mathrm{Tr}[\mathcal{P}_k(\mathbf{M}')] = \mathrm{Tr}\left[\mathcal{P}_k(\mathbf{M}' - \bar{\rho}^2\mathbf{I})\right] + k\bar{\rho}^2$$

$$\geq \mathrm{Tr}\left[\mathcal{P}_k(\mathbf{M} - \bar{\rho}^2\mathbf{I})\right] - k\rho^2\sqrt{\delta} + k\bar{\rho}^2 \qquad (27)$$

$$= \mathrm{Tr}\left[\mathbf{M} - \bar{\rho}^2\mathbf{I}\right] - k\rho^2\sqrt{\delta} + k\bar{\rho}^2,$$

where Equation (27) holds by the following matrix perturbation bound:

$$\left|\mathrm{Tr}\left[\mathcal{P}_k(\mathbf{M} - \bar{\rho}^2\mathbf{I})\right] - \mathrm{Tr}\left[\mathcal{P}_k(\mathbf{M}' - \bar{\rho}^2\mathbf{I})\right]\right| \leq \sum_{i=1}^k \left|\lambda_i(\mathbf{M}' - \bar{\rho}^2\mathbf{I}) - \lambda_i(\mathbf{M} - \bar{\rho}^2\mathbf{I})\right|$$

$$\leq k\rho^2\sqrt{\delta} \qquad \text{(Using Equation (24))}.$$

Plugging Equation (27) back in Equation (26), we have Equation (26) bounded by

$$\leq \mathcal{O}\left(\alpha\,\mathrm{Tr}\left[\mathbf{M} - \bar{\rho}^2\mathbf{I}\right] + \alpha k\bar{\rho}^2 + \rho^2 k\sqrt{\alpha} + \rho^2 k\sqrt{\delta}\right)$$

$$\leq \mathcal{O}\left(\alpha\,\mathrm{Tr}\left[\mathbf{M} - \bar{\rho}^2\mathbf{I}\right] + \rho^2 k\sqrt{\alpha} + \rho^2 k\sqrt{\delta}\right) \quad \text{(Using } \delta \leq 1/nd \leq \alpha\text{)}$$

$$\leq \mathcal{O}\left(\alpha\,\mathrm{Tr}\left[\mathbf{M} - \bar{\rho}^2\mathbf{I}\right] + \rho^2 k\sqrt{\alpha}\right)$$

Thus, we have obtained that

$$\left\|\widehat{\mathbf{U}}\widehat{\mathbf{U}}^\top\left(\sum_{j=1}^k p_j\mathbf{w}_j\mathbf{w}_j^\top\right)\widehat{\mathbf{U}}\widehat{\mathbf{U}}^\top - \sum_{j=1}^k p_j\mathbf{w}_j\mathbf{w}_j^\top\right\|_*$$

$$= \mathcal{O}\left(\alpha\,\mathrm{Tr}\left[\sum_{j=1}^k p_j\mathbf{w}_j\mathbf{w}_j^\top\right] + \rho^2 k\sqrt{\alpha}\right)$$

$$= \mathcal{O}\left(\rho^2 k\sqrt{\alpha}\right). \qquad \qquad \square$$

The following lemma connects this nuclear norm bound to a subspace bound that we want.

**Lemma D.3** (Gap-free spectral bound). *Given $k$ vectors $\mathbf{x}_1, \mathbf{x}_2, \cdots, \mathbf{x}_k \in \mathbb{R}^d$, we define $\mathbf{X}_i = \mathbf{x}_i\mathbf{x}_i^\top$ for each $i \in [k]$. For any $\gamma \geq 0$, $\sigma \in \mathbb{R}_+$, and any rank-$k$ PSD matrix $\widehat{\mathbf{M}} \in \mathbb{R}^{d \times d}$ such that*

$$\left\|\widehat{\mathbf{M}} - \mathcal{P}_k\left(\sigma^2\mathbf{I} + \sum_{i=1}^k \mathbf{X}_i\right)\right\|_* \leq \gamma, \qquad (28)$$

*where $\mathcal{P}_k(\cdot)$ is a best rank-k approximation of a matrix, we have*

$$\sum_{i\in[k]}\left\|\mathbf{x}_i^\top\left(\mathbf{I}-\widehat{\mathbf{U}}\widehat{\mathbf{U}}^\top\right)\right\|_2^2 \leq \min\left\{\gamma^2\sigma_{\max}/\sigma_{\min}^2\,,\,2\gamma^{2/3}\sigma_{\max}^{1/3}k^{2/3}\right\}, \qquad (29)$$

*where $\sigma_{\min}$ is the smallest non-zero singular value of $\sum_{i\in[k]}\mathbf{X}_i$, and $\sigma_{\max}=\left\|\sum_{i\in[k]}\mathbf{X}_i\right\|_2$, and $\widehat{\mathbf{U}}\in\mathbb{R}^{d\times k}$ is the matrix consisting of the top-k singular vectors of $\widehat{\mathbf{M}}$. Further, for all $i\in[k]$, we have*

$$\left\|\mathbf{x}_i^\top\left(\mathbf{I}-\widehat{\mathbf{U}}\widehat{\mathbf{U}}^\top\right)\right\|_2^2 \leq \min\left\{\gamma^2\|\mathbf{x}_i\|_2^2/\sigma_{\min}^2\,,\,2\gamma^{2/3}\|\mathbf{x}_i\|_2^{2/3}\right\}. \qquad (30)$$

We provide a proof in Section D.6. Using this lemma with $\sigma=0$,

$$\widehat{\mathbf{M}}=\widehat{\mathbf{U}}\widehat{\mathbf{U}}^\top\left(\sum_{j=1}^k p_j\mathbf{w}_j\mathbf{w}_j^\top\right)\widehat{\mathbf{U}}\widehat{\mathbf{U}}^\top,$$

$\mathbf{X}_i=p_i\mathbf{w}_i\mathbf{w}_i^\top$ for all $i\in[k]$, and the nuclear norm bound in Proposition D.2, we get

$$p_i\left\|\mathbf{w}_i^\top\left(\mathbf{I}-\widehat{\mathbf{U}}\widehat{\mathbf{U}}^\top\right)\right\|_2^2 \leq \min\left\{\rho^2\frac{\gamma^2}{\sigma_{\min}^2}p_i, 2\rho^{2/3}\gamma^{2/3}p_i^{1/3}\right\}$$

$$\implies \left\|\mathbf{w}_i^\top\left(\mathbf{I}-\widehat{\mathbf{U}}\widehat{\mathbf{U}}^\top\right)\right\|_2^2 \leq \min\left\{\rho^2\frac{\gamma^2}{\sigma_{\min}^2}, 2\frac{\rho^{2/3}\gamma^{2/3}}{p_i^{2/3}}\right\}$$

$$\lesssim \min\left\{\rho^6 k^2\alpha/\sigma_{\min}^2, \rho^2 k^{2/3}\alpha^{1/3}/p_i^{2/3}\right\}. \qquad (31)$$

Since we are aiming for $\left\|\mathbf{w}_i^\top\left(\mathbf{I}-\widehat{\mathbf{U}}\widehat{\mathbf{U}}^\top\right)\right\|_2 = \epsilon\rho$ error, we need

$$\alpha=\mathcal{O}\left(\max\left\{\frac{\epsilon^2\sigma_{\min}^2}{\rho^4 k^2}, \frac{\epsilon^6 p_{\min}^2}{k^2}\right\}\right)$$

and the sample complexity is

$$n = \widetilde{\Omega}\left(dk^2\min\left\{\frac{\rho^4 k^2}{\epsilon^2\sigma_{\min}^2}, \frac{k^2}{\epsilon^6 p_{\min}^2}\right\}\right).$$

In the analysis above, we assume that each example $\beta_i$ is independently drawn. While this is true when $t_{L_1}=1$, it is no longer the case when $t_{L_1}>1$ where we have to break up the examples from each task into $t_{L_1}$ different estimators. Recall that $\widehat{\mathbf{p}}$ is the vector of the empirical fractions of the examples that correspond to each linear regressor, and $\mathbf{p}$ is the population version of it. For given a pair of parameters $n_{L_1}, t_{L_1}$ let us define

$$\widehat{\mathbf{p}}\sim\frac{1}{n_{L_1}t_{L_1}}\text{multinomial}(n_{L_1},\mathbf{p})\cdot t_{L_1},$$

and independently

$$\widehat{\mathbf{p}}^*\sim\frac{1}{n_{L_1}t_{L_1}}\text{multinomial}(n_{L_1}\cdot t_{L_1},\mathbf{p}).$$

Notice that $\widehat{\mathbf{p}}$ corresponds to the setting where there are $n_{L_1}$ tasks with each task having $t_{L_1}$ examples, and $\widehat{\mathbf{p}}^*$ corresponds to the setting where there are $n_{L_1}\cdot t_{L_1}$ tasks, each with 1 example. By Proposition J.5, we know that when

$$n_{L_1}\geq\frac{2k\log(2/\delta)}{\alpha^2},$$

$\|\widehat{\mathbf{p}}-\widehat{\mathbf{p}}^*\|_1\leq\alpha$ with probability $1-\delta$. Hence if we denote the set $G=\{\beta_i\}_{i=1}^{n_{L_1}\cdot t_{L_1}}$ to be the set of the data coming from the model where each task has $t_{L_1}$ examples. There exists a distribution of set $L, E$ such that $G=(G'\setminus L')\cup E'$ with $|L|=|E|$, $G'$ has data from $n_{L_1}\cdot t_{L_1}$ independent tasks with

1 example per task, and $|L| = |E| \leq \alpha$ with probability $1 - \delta$. Thus we have obtain a reduction from $t_{L_1}$ examples per task setting to the 1 example per task setting, in which case our algorithm receives a dataset with less than $2\alpha$ fraction of corruption with probability $\delta$. Since our previous proof applies to this setting, this concludes the proof of Lemma B.1 and the final sample complexity is

$$n_{L1} = \widetilde{\Omega}\Big( \frac{dk^2}{t_{L1}} \min\Big\{ \frac{\rho^4 k^2}{\epsilon^2 \sigma_{\min}^2}, \frac{k^2}{\epsilon^6 p_{\min}^2} \Big\} + k \min\Big\{ \frac{\rho^8 k^4}{\epsilon^4 \sigma_{\min}^4}, \frac{k^4}{\epsilon^{12} p_{\min}^4} \Big\} \Big) , \tag{32}$$

We are left to show $B = \mathcal{O}(\rho^2 d \log^2(nd/\delta))$, $\nu(k) = \mathcal{O}(\rho^2 k)$, and the mean shift bound in Eq. (24).

*Proof of Proposition D.1.* We first show $B = \mathcal{O}((\rho^2 d) \log^2(nd/\delta))$. From [42, Proposition A.1], we have $\|\widehat{\beta}_i\|_2^2 \leq (\rho^2 d) \log^2(nd/\delta)$ (i.e. event $\mathcal{E}$ happens) with probability at least $1 - \delta$. Using this, we have

$$\begin{aligned}
\Big\| \widehat{\beta}_i \widehat{\beta}_i^\top - \mathbf{M} \Big\|_2 &\leq \Big\| \widehat{\beta}_i \Big\|_2^2 + \|\mathbf{M}\|_2 \\
&\leq \rho^2 d \log^2(nd/\delta) + 3\rho^2 \\
&\leq 4\rho^2 d \log^2(nd/\delta) ,
\end{aligned} \tag{33}$$

for all $i \in [n]$ with probability at least $1 - \delta$.

Second, we bound the mean shift conditioned on event $\mathcal{E}$

$$\Big\| \mathbb{E}\Big[ \widehat{\beta}_i \widehat{\beta}_i \Big| \mathcal{E} \Big] - \mathbf{M} \Big\|_2 = \max_{\|\mathbf{v}\|_2 = 1} |\mathbb{E}[z_\mathbf{v}|\mathcal{E}]|, \tag{34}$$

where $z_\mathbf{v} := \left( \mathbf{v}^\top \widehat{\beta}_i \right)^2 - \mathbf{v}^\top \mathbf{M}\mathbf{v}$. The random variable $z_\mathbf{v}$ is centered with variance

$$\begin{aligned}
\mathbb{E}\left[ \left( \left( \mathbf{v}^\top \widehat{\beta}_i \right)^2 - \mathbf{v}^\top \mathbf{M}\mathbf{v} \right)^2 \right] &\leq \mathbb{E}\left[ \left( \mathbf{v}^\top \widehat{\beta}_i \right)^4 \right] - \left( \mathbf{v}^\top \mathbf{M}\mathbf{v} \right)^2 \\
&= \mathcal{O}(\rho^4) \quad (\mathbf{v}^\top \widehat{\beta}_i \text{ is sub-exponential r.v.}) ,
\end{aligned} \tag{35}$$

Recall that $\mathbf{M}' := \mathbb{E}\Big[ \widehat{\beta}_i \widehat{\beta}_i \Big| \mathcal{E} \Big]$, then using J.1, we have

$$\begin{aligned}
\|\mathbf{M}' - \mathbf{M}\|_2 &= \max_{\|\mathbf{v}\|_2 = 1} |\mathbb{E}[z_\mathbf{v}|\mathcal{E}]| \\
&\leq \max_{\|\mathbf{v}\|_2 = 1} \frac{\mathbb{E}[z_\mathbf{v}] + \sqrt{(1 - \mathbb{P}(\mathcal{E})) \cdot \mathrm{Var}(z_\mathbf{v})}}{\mathbb{P}[\mathcal{E}]} \\
&\leq \mathcal{O}\left( \rho^2 \frac{\sqrt{\delta}}{\mathbb{P}[\mathcal{E}]} \right) \\
&\leq \mathcal{O}\left( \rho^2 \sqrt{\delta} \right).
\end{aligned} \tag{36}$$

Finally, we show $\nu^2(k) = \mathcal{O}(k\rho^4)$. For any symmetric real matrix $\mathbf{A}$ with $\mathrm{rank}(\mathbf{A}) = k$, and $\|\mathbf{A}\|_F \leq 1$,

$$\begin{aligned}
\mathbb{E}\left[ \mathrm{Tr}\Big[ \mathbf{A}\Big( \widehat{\beta}_i \widehat{\beta}_i^\top - \mathbf{M}' \Big) \Big]^2 \Big| \mathcal{E} \right] &= \mathbb{E}\left[ \mathrm{Tr}\Big[ \mathbf{A}\Big( \widehat{\beta}_i \widehat{\beta}_i^\top - \mathbf{M} + (\mathbf{M} - \mathbf{M}') \Big) \Big]^2 \Big| \mathcal{E} \right] \\
&= \mathbb{E}\left[ \mathrm{Tr}\Big[ \mathbf{A}\Big( \widehat{\beta}_i \widehat{\beta}_i^\top - \mathbf{M} \Big) \Big]^2 \Big| \mathcal{E} \right] + \mathrm{Tr}[\mathbf{A}(\mathbf{M} - \mathbf{M}')]^2 \\
&\leq \mathbb{E}\left[ \mathrm{Tr}\Big[ \mathbf{A}\Big( \widehat{\beta}_i \widehat{\beta}_i^\top - \mathbf{M} \Big) \Big]^2 \Big| \mathcal{E} \right] + \mathcal{O}(\rho^4)
\end{aligned} \tag{37}$$

where the last inequality is obtained using Equation (36). Considering the first term in Equation (37), we get

$$
\mathbb{E}\left[\operatorname{Tr}\left[\mathbf{A}\left(\widehat{\beta}_i\widehat{\beta}_i^\top - \mathbf{M}\right)\right]^2 \Big| \mathcal{E}\right]
$$

$$
= \mathbb{E}\left[\left(\sum_{j=1}^k \lambda_j \mathbf{v}_j \mathbf{v}_j^\top \left(\widehat{\beta}_i\widehat{\beta}_i^\top - \mathbf{M}\right)\right)^2 \Bigg| \mathcal{E}\right]
$$

$$
= \mathbb{E}\left[\sum_{j,j'=1}^k \lambda_j \lambda_{j'}\left(\left(\widehat{\beta}_i^\top \mathbf{v}_j\right)^2\left(\widehat{\beta}_i^\top \mathbf{v}_{j'}\right)^2 - \mathbf{v}_j^\top \mathbf{M}\mathbf{v}_j \mathbf{v}_{j'}^\top \mathbf{M}\mathbf{v}_{j'}\right) \Bigg| \mathcal{E}\right]
$$

$$
\leq \sum_{j,j'=1}^k \lambda_j \lambda_{j'}\left(\sqrt{\mathbb{E}\left[\left(\widehat{\beta}_i^\top \mathbf{v}_j\right)^4 \Big| \mathcal{E}\right]\mathbb{E}\left[\left(\widehat{\beta}_i^\top \mathbf{v}_{j'}\right)^4 \Big| \mathcal{E}\right]} - \mathbf{v}_j^\top \mathbf{M}\mathbf{v}_j \mathbf{v}_{j'}^\top \mathbf{M}\mathbf{v}_{j'}\right) \quad \text{(Cauchy-Schwarz)}
$$

$$
\leq \mathcal{O}\left(\left(\sum_{j=1}^k \lambda_j\right)^2 \rho^4\right) \qquad \text{(4-th moment bound)}
$$

$$
\leq \mathcal{O}\left(k\rho^4\right). \tag{38}
$$

Using Equation (38) in Equation (37), we get

$$
\mathbb{E}\left[\operatorname{Tr}\left[\mathbf{A}\left(\widehat{\beta}_i\widehat{\beta}_i^\top - \mathbf{M}'\right)\right]^2 \Big| \mathcal{E}\right] \leq \underbrace{\mathcal{O}\left(k\rho^4\right)}_{=\nu^2(k)}. \qquad \qquad \square
$$

## D.1 Proof of Proposition 2.6

The following main technical lemma guarantees that for any distribution $\mathbf{X} \sim \mathcal{P}$ with a bounded support and a bounded second moment, the filtering algorithm we introduce in Algorithm 2 robustly finds an accurate estimate of the principal subspace.

**Lemma D.4** (Main Lemma for Algorithm 2). *Let $\mathcal{P}$ be a distribution over $d \times d$ PSD matrices with the property that,*

$$
\mathbb{E}_{\mathbf{X}\sim\mathcal{P}}[\mathbf{X}] = \mathbf{M}, \quad \|\mathbf{X} - \mathbf{M}\|_2 \leq B, \quad \text{and} \quad \max_{\|\mathbf{A}\|_\mathrm{F}\leq 1, \operatorname{rank}(\mathbf{A})\leq k} \mathbb{E}_{\mathbf{X}\sim\mathcal{P}}\left[\operatorname{Tr}[\mathbf{A}(\mathbf{X} - \mathbf{M})]^2\right] \leq \nu(k)^2.
$$

*Let a set of $n$ random matrices $G = \left\{\mathbf{X}_i \in \mathbb{R}^{d \times d}\right\}_{i \in [n]}$ where each $\mathbf{X}_i$ is independently drawn from $\mathcal{P}$, and the at most $\alpha$ fraction is corrupted by an adversary such that the input dataset $S = (G \backslash L) \cup E$ with $|E| = |L| \leq \alpha n$, $L \subset G$. There exists a numerical constant $c > 0$ such that for any $0 < \alpha < c$, if $n = \Omega((dk^2 + (B/\nu)\sqrt{k\alpha})\log(d/(\delta\alpha))/\alpha)$, Algorithm 2 outputs a dataset $S' \subseteq S$ satisfying the following for $\widehat{\mathbf{M}} = \frac{1}{|S'|}\sum_{\mathbf{X}_i \in S'}\mathbf{X}_i$:*

*1. for the top-$k$ singular vectors $\widehat{\mathbf{U}} \in \mathbb{R}^{d \times k}$ of $\widehat{\mathbf{M}}$,*
$$
\operatorname{Tr}\left[\widehat{\mathbf{U}}^\top\left(\widehat{\mathbf{M}} - \mathbf{M}\right)\widehat{\mathbf{U}}\right] \leq 48\alpha \operatorname{Tr}\left[\widehat{\mathbf{U}}^\top \mathbf{M}\widehat{\mathbf{U}}\right] + 102\nu\sqrt{k\alpha}.
$$

*2. for all rank-$k$ semi-orthogonal matrices $\mathbf{V} \in \mathbb{R}^{d \times k}$, we have*
$$
\operatorname{Tr}\left[\mathbf{V}^\top\left(\widehat{\mathbf{M}} - \mathbf{M}\right)\mathbf{V}\right] \geq -10\alpha \operatorname{Tr}\left[\mathbf{V}^\top \mathbf{M}\mathbf{V}\right] - 8\nu\sqrt{k\alpha}.
$$

We provide a proof in Section D.2.

The proof of Proposition 2.6 is straightforward given Lemma D.4. For the first claim, note that,

$$
\operatorname{Tr}\left[\widehat{\mathbf{U}}^\top \mathbf{\Sigma}\widehat{\mathbf{U}}\right] \geq \operatorname{Tr}\left[\widehat{\mathbf{U}}^\top \widehat{\mathbf{\Sigma}}\widehat{\mathbf{U}}\right] - 48\alpha \operatorname{Tr}[\mathcal{P}_k(\mathbf{\Sigma})] - 102\nu\sqrt{k\alpha} \qquad \text{(Using Lemma D.4, part 1)}
$$

$$
\geq \operatorname{Tr}\left[\mathbf{U}^\top \widehat{\mathbf{\Sigma}}\mathbf{U}\right] - 48\alpha \operatorname{Tr}[\mathcal{P}_k(\mathbf{\Sigma})] - 102\nu\sqrt{k\alpha} \qquad \text{(Property of SVD)}
$$

$$
\geq \operatorname{Tr}[\mathcal{P}_k(\mathbf{\Sigma})] - 58\alpha \operatorname{Tr}[\mathcal{P}_k(\mathbf{\Sigma})] - 110\nu\sqrt{k\alpha} \qquad \text{(Using Lemma D.4, part 2).}
$$

For the second claim, since $\boldsymbol{\Sigma} \succeq \widehat{\mathbf{U}}\widehat{\mathbf{U}}^\top \boldsymbol{\Sigma} \widehat{\mathbf{U}}\widehat{\mathbf{U}}^\top$, we have

$$
\begin{aligned}
\left\| \boldsymbol{\Sigma} - \widehat{\mathbf{U}}\widehat{\mathbf{U}}^\top \boldsymbol{\Sigma} \widehat{\mathbf{U}}\widehat{\mathbf{U}}^\top \right\|_* &= \mathrm{Tr}\left[ \boldsymbol{\Sigma} - \widehat{\mathbf{U}}\widehat{\mathbf{U}}^\top \boldsymbol{\Sigma} \widehat{\mathbf{U}}\widehat{\mathbf{U}}^\top \right] \\
&\leq \mathrm{Tr}[\boldsymbol{\Sigma}] - (1 - 58\alpha)\,\mathrm{Tr}[\mathcal{P}_k(\boldsymbol{\Sigma})] + 110\nu\sqrt{k\alpha} \quad \text{(From the first claim).} \\
&= \mathrm{Tr}[\boldsymbol{\Sigma} - \mathcal{P}_k(\boldsymbol{\Sigma})] + 58\alpha\,\mathrm{Tr}[\mathcal{P}_k(\boldsymbol{\Sigma})] + 110\nu\sqrt{k\alpha} \\
&= \|\boldsymbol{\Sigma} - \mathcal{P}_k(\boldsymbol{\Sigma})\|_* + 58\alpha\|\mathcal{P}_k(\boldsymbol{\Sigma})\|_* + 110\nu\sqrt{k\alpha}.
\end{aligned}
$$

Similarly, we also get

$$
\left\| \mathcal{P}_k(\boldsymbol{\Sigma}) - \widehat{\mathbf{U}}\widehat{\mathbf{U}}^\top \boldsymbol{\Sigma} \widehat{\mathbf{U}}\widehat{\mathbf{U}}^\top \right\|_* \leq 58\alpha\|\mathcal{P}_k(\boldsymbol{\Sigma})\|_* + 110\nu\sqrt{k\alpha} \tag{39}
$$

## D.2   Proof of the main analysis of Algorithm 2 in Lemma D.4

The proof of Lemma D.4 is divided into two parts, for each statement of the lemma. Both parts are proven under the following good event. We provide a proof of this lemma in Section D.4.

**Lemma D.5.** *Under the hypotheses of Lemma D.4, when $n = \Omega((dk^2 + \frac{B}{\nu}\sqrt{k\epsilon})\log(d/(\epsilon\widetilde{\delta}))/\epsilon)$ with probability $1-\widetilde{\delta}$, the following events happen for all semi-orthogonal matrices $\mathbf{V} \in \mathbb{R}^{d\times k} s.t. \mathbf{V}^\top\mathbf{V} = \mathbf{I}_k$,*

1. *There exists $G_{\mathbf{V}} \subset G$ such that*

   (a) $|G_{\mathbf{V}}| \geq (1 - \epsilon)n$,

   (b) $\left| \frac{1}{|G_{\mathbf{V}}|} \mathrm{Tr}\left[ \sum_{\mathbf{X}_i \in G_{\mathbf{V}}} \left( \mathbf{V}^\top(\mathbf{X}_i - \mathbf{M})\mathbf{V} \right) \right] \right| \leq 1.01\nu\sqrt{k\epsilon}$, and

   (c) $\frac{1}{|G_{\mathbf{V}}|} \sum_{\mathbf{X}_i \in G_{\mathbf{V}}} \mathrm{Tr}\left[ \mathbf{V}^\top(\mathbf{X}_i - \mathbf{M})\mathbf{V} \right]^2 \leq 6.01k\nu^2$,

2. $\left| \dfrac{1}{n} \displaystyle\sum_{\mathbf{X}_i \in G} \mathrm{Tr}\left[ \mathbf{V}^\top(\mathbf{X}_i - \mathbf{M})\mathbf{V} \right] \right| \leq \nu\sqrt{k\epsilon}$,

3. *All subset $T \subset G$ such that $|T| \leq \epsilon n$ satisfies*

$$
\sum_{\mathbf{X}_i \in T} \mathrm{Tr}\left[ \mathbf{V}^\top(\mathbf{X}_i - \mathbf{M})\mathbf{V} \right] \leq 7n\nu\sqrt{k\epsilon} + n\epsilon\,\mathrm{Tr}\left[ \mathbf{V}^\top\mathbf{M}\mathbf{V} \right].
$$

We provide the proof in Section D.4.

### D.2.1   Part 1 of Lemma D.4

**Proposition D.6** (Correctness of Algorithm 3). *For a set $G$ of $n$ uncorrupted matrices defined as in the hypotheses of Lemma D.5 and for some $\epsilon \geq \alpha > 0$, suppose the that set $S$ input to Algorithm 3 satisfies the following: $S = (G \setminus L) \cup E$ with $L \subset G$, $|E| \leq \alpha|G|$, $|L| \leq 9\alpha|G|$, $|S| \leq |G|$, and $E \cap G = \emptyset$. If the events in Lemma D.5 hold, then a single call of Algorithm 3 outputs a set $S' \subseteq S$ achieving one of the following two guarantees.*

1. *If Algorithm 3 returns $S' = S$ (unchanged), then*

$$
\mathrm{Tr}\left[ \widehat{\mathbf{U}}^\top \left( \frac{1}{|S'|} \sum_{\mathbf{X}_i \in S'} \mathbf{X}_i - \mathbf{M} \right) \widehat{\mathbf{U}} \right] \leq 48\alpha\,\mathrm{Tr}\left[ \widehat{\mathbf{U}}^\top\mathbf{M}\widehat{\mathbf{U}} \right] + 102\nu\sqrt{k\alpha} \tag{40}
$$

2. *If Algorithm 3 returns $S' \subset S$, then there exist two sets $L' \supseteq L$ and $E' \subseteq E$ such that $S' = (G \setminus L') \cup E'$ and $\mathbb{E}[2|L'| + |E'|] \leq 2|L| + |E|$.*

*where $\widehat{\mathbf{U}}$ is the top-k singular matrix of $\frac{1}{|S'|}\sum_{\mathbf{X}_i \in S'} \mathbf{X}_i$.*

We provide the proof in Section D.3. We are left to show that when $n$ is sufficiently large, then Algorithm 2 terminates before removing too many points, with probability at least $1 - \delta$. To get the

logarithmic dependence on $\delta$, we divide the meta-training dataset $\mathcal{D}_{L1}$ into $\log(1/\delta)$ partitions of an equal size, and apply the same routine of Algorithm 2.

We show that each run of Algorithm 2 succeeds with a strictly positive probability, and hence one of them is guaranteed to succeed with probability at least $1 - \delta$. Further, we have a simple way of choosing the successful run; we select the one that outputs the largest set $S'$. Precisely, we call a routine successful if $|S'| \geq (1 - 8\alpha)|G|$.

First, we need to show that the conditions of Proposition D.6 hold, throughout the iterations. However, the condition that $|L| \leq 9\alpha|G|$ might be violated by chance, as we only have guarantee in expectation. We thus bound the probability that there exists a sub-routine of 1-D filtering that results in $|L'| \geq 8\alpha|G|$. The proof is similar to the one in [24]. Let $L_i$ denote the removed set of uncorrupted points in $S_i = (G \backslash L_i) \cup E_i$. Notice that this event implies $|L_T| \geq 8\alpha|G|$ as $L_i$'s are a monotonically increasing sequence of sets. From Markov's inequality, we have $\mathbb{P}(|L_T| \geq 8\alpha|G|) \leq \mathbb{E}[|L_T|]/(8\alpha|G|) \leq 1/6$, where in the last inequality we used the fact that $\mathbb{E}[|L_T|] \leq (1/2)\,\mathbb{E}[2|L_T| + |E_T|] \leq (1/2)(2|L_0| + |E_0|) \leq (3/2)\alpha|G|$. Hence, with probability at least $5/6$ one run succeeds (taking a union bound with Lemma D.5 for the good events with a choice of $\widetilde{\delta} = 1/6$). Out of $\log_6(2/\delta)$ runs, one succeeds with probability at least $1 - \delta/2$.

### D.2.2   Part 2 of Lemma D.4

There exists a set $T \subset G$ such that $S' \supseteq T$, and $|T| \geq (1-9\alpha)|G|$. Since $S'$ contains a good fraction of points from $G$, then for any semi-orthogonal matrix $\mathbf{V} \in \mathbb{R}^{d \times k}$, we have

$$
\begin{aligned}
&\sum_{\mathbf{X}_i \in S'} \mathrm{Tr}\big[\mathbf{V}^\top \mathbf{X}_i \mathbf{V}\big] \\
&\geq \sum_{\mathbf{X}_i \in G} \mathrm{Tr}\big[\mathbf{V}^\top \mathbf{X}_i \mathbf{V}\big] - \sum_{\mathbf{X}_i \in G \backslash T} \mathrm{Tr}\big[\mathbf{V}^\top \mathbf{X}_i \mathbf{V}\big] \\
&\geq \sum_{\mathbf{X}_i \in G} \mathrm{Tr}\big[\mathbf{V}^\top \mathbf{X}_i \mathbf{V}\big] - \left( \sum_{\mathbf{X}_i \in G \backslash T} \mathrm{Tr}\big[\mathbf{V}^\top \mathbf{M} \mathbf{V}\big] + n\alpha\,\mathrm{Tr}\big[\mathbf{V}^\top \mathbf{M} \mathbf{V}\big] + 7n\nu\sqrt{k\alpha} \right) \\
&\hspace{8cm} \text{(Using Lemma D.5, part 3)} \\
&\geq \sum_{\mathbf{X}_i \in G} \mathrm{Tr}\big[\mathbf{V}^\top \mathbf{M} \mathbf{V}\big] - n\nu\sqrt{k\alpha} - \left( \sum_{\mathbf{X}_i \in G \backslash T} \mathrm{Tr}\big[\mathbf{V}^\top \mathbf{M} \mathbf{V}\big] + n\alpha\,\mathrm{Tr}\big[\mathbf{V}^\top \mathbf{M} \mathbf{V}\big] + 7n\nu\sqrt{k\alpha} \right) \\
&\hspace{8cm} \text{(Using Lemma D.5, part 2)} \\
&= \sum_{\mathbf{X}_i \in T} \mathrm{Tr}\big[\mathbf{V}^\top \mathbf{M} \mathbf{V}\big] - 8n\nu\sqrt{k\alpha} - n\alpha\,\mathrm{Tr}\big[\mathbf{V}^\top \mathbf{M} \mathbf{V}\big] \\
&\geq n(1 - 10\alpha)\,\mathrm{Tr}\big[\mathbf{V}^\top \mathbf{M} \mathbf{V}\big] - 8n\nu\sqrt{k\alpha} \qquad (\because |T| \geq (1-9\alpha)n)
\end{aligned}
$$

$$
\begin{aligned}
&\implies n\,\mathrm{Tr}\Big[\mathbf{V}^\top \widehat{\mathbf{M}} \mathbf{V}\Big] \geq |S'|\,\mathrm{Tr}\Big[\mathbf{V}^\top \widehat{\mathbf{M}} \mathbf{V}\Big] \geq n(1 - 10\alpha)\,\mathrm{Tr}\big[\mathbf{V}^\top \mathbf{M} \mathbf{V}\big] - 8n\nu\sqrt{k\alpha} \\
&\text{or, } \mathrm{Tr}\Big[\mathbf{V}^\top \big(\widehat{\mathbf{M}} - \mathbf{M}\big) \mathbf{V}\Big] \geq -10\alpha\,\mathrm{Tr}\big[\mathbf{V}^\top \mathbf{M} \mathbf{V}\big] - 8\nu\sqrt{k\alpha}
\end{aligned}
$$

when the good events of Lemma D.5 hold, which happens if $n = \Omega\Big( \frac{1}{\epsilon}\big(k + \frac{B}{\nu}\sqrt{k\epsilon}\big) \log \frac{d}{\delta} \Big)$ with probability at least $1 - \delta$.

### D.3 Proof of Proposition D.6

1. To show the first part of Proposition D.6, notice that by Lemma D.5, part 1, there exists a subset $G_{\widehat{\mathbf{U}}} \subset G$ such that

$$\left|G_{\widehat{\mathbf{U}}}\right| \geq (1-\alpha)n$$

$$\left|\frac{1}{\left|G_{\widehat{\mathbf{U}}}\right|} \sum_{\mathbf{X}_i \in G_{\widehat{\mathbf{U}}}} \text{Tr}\left[\widehat{\mathbf{U}}^\top (\mathbf{X}_i - \mathbf{M})\widehat{\mathbf{U}}\right]\right| \leq 1.01\nu\sqrt{k\alpha}$$

$$\frac{1}{\left|G_{\widehat{\mathbf{U}}}\right|} \sum_{\mathbf{X}_i \in G_{\widehat{\mathbf{U}}}} \text{Tr}\left[\widehat{\mathbf{U}}^\top (\mathbf{X}_i - \mathbf{M})\widehat{\mathbf{U}}\right]^2 \leq 6.01k\nu^2.$$

For the input to the First-Filter algorithm, $S_0 = (G \setminus L) \cup E$, we can reclassify the sample in $G \setminus G_{\widehat{\mathbf{U}}}$ to be the error and have $S_0 = (G_{\widehat{\mathbf{U}}} \setminus L) \cup E'$ where $|L| \leq 9\alpha n$ and $|E'| \leq |E| + \alpha n \leq 2\alpha n$. Hence the output of the First-Filter algorithm satisfies

$$\left|\frac{1}{|S_G|} \sum_{\mathbf{X}_i \in S_G} \text{Tr}\left[\widehat{\mathbf{U}}^\top \mathbf{X}_i \widehat{\mathbf{U}}\right] - \text{Tr}\left[\widehat{\mathbf{U}}^\top \mathbf{M}\widehat{\mathbf{U}}\right]\right| \leq 54\nu\sqrt{k\alpha}, \tag{41}$$

using Proposition J.4, and from **if** condition of Algorithm 3, we have

$$\frac{1}{n}\sum_{i=1}^{n} \text{Tr}\left[\widehat{\mathbf{U}}^\top \mathbf{X}_i \widehat{\mathbf{U}}\right] - \frac{1}{|S_G|} \sum_{\mathbf{X}_i \in S_G} \text{Tr}\left[\widehat{\mathbf{U}}^\top \mathbf{X}_i \widehat{\mathbf{U}}\right] \leq 48\left(\nu\sqrt{k\alpha} + \alpha \text{Tr}\left[\widehat{\mathbf{U}}^\top \mathbf{M}\widehat{\mathbf{U}}\right]\right). \tag{42}$$

Combining the above two inequalities, we get

$$\frac{1}{n}\sum_{i=1}^{n} \text{Tr}\left[\widehat{\mathbf{U}}^\top \mathbf{X}_i \widehat{\mathbf{U}}\right] - \text{Tr}\left[\widehat{\mathbf{U}}^\top \mathbf{M}\widehat{\mathbf{U}}\right]$$

$$= \frac{1}{n}\sum_{i=1}^{n} \text{Tr}\left[\widehat{\mathbf{U}}^\top \mathbf{X}_i \widehat{\mathbf{U}}\right] - \frac{1}{|S_G|} \sum_{\mathbf{X}_i \in S_G} \text{Tr}\left[\widehat{\mathbf{U}}^\top \mathbf{X}_i \widehat{\mathbf{U}}\right]$$

$$+ \frac{1}{|S_G|} \sum_{\mathbf{X}_i \in S_G} \text{Tr}\left[\widehat{\mathbf{U}}^\top \mathbf{X}_i \widehat{\mathbf{U}}\right] - \text{Tr}\left[\widehat{\mathbf{U}}^\top \mathbf{M}\widehat{\mathbf{U}}\right]$$

$$\leq \frac{1}{n}\sum_{i=1}^{n} \text{Tr}\left[\widehat{\mathbf{U}}^\top \mathbf{X}_i \widehat{\mathbf{U}}\right] - \frac{1}{|S_G|} \sum_{\mathbf{X}_i \in S_G} \text{Tr}\left[\widehat{\mathbf{U}}^\top \mathbf{X}_i \widehat{\mathbf{U}}\right]$$

$$+ \left|\frac{1}{|S_G|} \sum_{\mathbf{X}_i \in S_G} \text{Tr}\left[\widehat{\mathbf{U}}^\top \mathbf{X}_i \widehat{\mathbf{U}}\right] - \text{Tr}\left[\widehat{\mathbf{U}}^\top \mathbf{M}\widehat{\mathbf{U}}\right]\right|$$

$$\leq 102\nu\sqrt{k\alpha} + 48\alpha \text{Tr}\left[\widehat{\mathbf{U}}^\top \mathbf{M}\widehat{\mathbf{U}}\right]. \tag{43}$$

2. We use $x_i$ to denote $\text{Tr}\left[\mathbf{U}^\top \mathbf{X}_i \mathbf{U}\right]$ and $\mu^P$ to denote $\text{Tr}\left[\mathbf{U}^\top \mathbf{M} \mathbf{U}\right]$. Our goal is to show that our Algorithm removes much less good points, from the set $G \cap (S \setminus S_G)$. Notice that,

$$
\begin{aligned}
\sum_{\substack{x_i \in S \setminus S_G, \\ x_i \geq \mu^{S_G}}} \left(x_i - \mu^{S_G}\right) &= \sum_{x_i \in S \setminus S_G} \left(x_i - \mu^{S_G}\right) - \sum_{\substack{x_i \in S \setminus S_G, \\ x_i < \mu^{S_G}}} \left(x_i - \mu^{S_G}\right) \\
&\geq \sum_{x_i \in S \setminus S_G} \left(x_i - \mu^{S_G}\right) \\
&= \sum_{x_i \in S} \left(x_i - \mu^{S_G}\right) - \sum_{x_i \in S_G} \left(x_i - \mu^{S_G}\right) \\
&= \sum_{x_i \in S} \left(x_i - \mu^{S_G}\right) \\
&\geq 48n\left(\alpha\mu^{S_G} + \nu\sqrt{k\alpha}\right) \qquad \text{(from the if clause in Algorithm 3)} \\
&\geq 48n\left(\alpha\mu^P + (1 - 18\alpha)\nu\sqrt{k\alpha}\right) \\
&\geq 48n\left(\alpha\mu^P + \frac{1}{2}\nu\sqrt{k\alpha}\right)
\end{aligned}
\tag{44}
$$

where the last inequality is using $\alpha \leq 1/36$, and the second last from the guarantee from the First-Filter algorithm. (by the if clause in the algorithm).

On the other hand, note that,

$$
\left|\left\{x_i \in G \cap (S \setminus S_G) \,\middle|\, x_i \geq \mu^{S_G}\right\}\right| \leq |(S \setminus S_G)| \leq n\alpha.
\tag{45}
$$

Therefore,

$$
\begin{aligned}
\sum_{\substack{x_i \in G \cap (S \setminus S_G), \\ x_i \geq \mu^{S_G}}} \left(x_i - \mu^{S_G}\right) &\leq \sum_{\substack{x_i \in G \cap (S \setminus S_G), \\ x_i \geq \mu^{S_G}}} \left(x_i - \mu^P\right) + |S \setminus S_G|\left|\mu^P - \mu^{S_G}\right| \\
&\leq \sum_{\substack{x_i \in G \cap (S \setminus S_G), \\ x_i \geq \mu^{S_G}}} \left(x_i - \mu^P\right) + n\alpha\left|\mu^P - \mu^{S_G}\right| \\
&\leq n\left(7\nu\sqrt{k\alpha} + \alpha\mu^P + 18\nu\alpha\sqrt{k\alpha}\right) \\
&\qquad \text{(Lemma D.5, part 3 and Proposition J.4)} \\
&\leq n\left((7 + 18\alpha)\nu\sqrt{k\alpha} + \alpha\mu^P\right) \\
&\leq n\left(8\nu\sqrt{k\alpha} + \alpha\mu^P\right) \qquad (\because \alpha \leq 1/36 \leq 1/18.) \\
&\leq n\left(8\nu\sqrt{k\alpha} + 16\alpha\mu^P\right) \\
&= 16n\left(\alpha\mu^P + \frac{1}{2}\nu\sqrt{k\alpha}\right)
\end{aligned}
\tag{47}
$$

where the last inequality follows by using Lemma D.5, part 3, in conjunction with Equation (45), and using the guarantee provided in Proposition J.4. Hence we have shown that

$$
\sum_{\substack{x_i \in S \setminus S_G, \\ x_i \geq \mu^{S_G}}} \left(x_i - \mu^{S_G}\right) \geq 3 \sum_{\substack{x_i \in G \cap (S \setminus S_G), \\ x_i \geq \mu^{S_G}}} \left(x_i - \mu^{S_G}\right).
\tag{48}
$$

Applying Fact D.7, we get

$$
\mathbb{E}[|S \setminus S'|] = \frac{\sum\limits_{\substack{i \in S \setminus S_G, \\ x_i \geq \mu^{S_G}}} \left(x_i - \mu^{S_G}\right)}{\max\{x_i - \mu^{S_G}\}_{i \in S \setminus S_G}}
\tag{49}
$$

$$
\text{and, } \mathbb{E}[|G \cap (S \setminus S')|] = \frac{\sum\limits_{\substack{i \in G \cap (S \setminus S_G), \\ x_i \geq \mu^{S_G}}} \left(x_i - \mu^{S_G}\right)}{\max\{x_i - \mu^{S_G}\}_{i \in S \setminus S_G}}.
\tag{50}
$$

This combined with Equation (48) gives

$$
\begin{aligned}
3 \, \mathbb{E}[|G \cap (S \setminus S')|] &\leq \mathbb{E}[|S \setminus S'|] \\
&= \mathbb{E}[|(L' \setminus L) \cup (E \setminus E')|] \\
&\leq \mathbb{E}[|L' \setminus L|] + \mathbb{E}[|E \setminus E'|].
\end{aligned}
\tag{51}
$$

Since $|L' \setminus L| = |G \cap (S \setminus S')|$, we finally have

$$
\mathbb{E}[|G \cap (S \setminus S')|] \leq \frac{1}{2} \mathbb{E}[|E \setminus E'|],
$$
$$
\implies \mathbb{E}[2|L'| + |E'|] \leq 2|L| + |E|.
\tag{52}
$$

Hence, we can run the filter on $\{\mathbf{X}_i \in S \setminus S_G\}$, and guarantee that on every application of Algorithm 3, we remove at-least 2 times more corrupted points in expectation than good points.

This completes the proof.

**Fact D.7** (Filter based on mean). *Assuming that $a \leq x_1 \leq \ldots \leq x_n \leq b$, $t \sim \mathcal{U}[a, b]$, then*

$$
\mathbb{E}\left[\sum_{i=1}^{n} \mathbb{1}\{x_i > t\}\right] = \sum_{i=1}^{n} (x_i - a)/(b - a)
$$

### D.4 Proof of Lemma D.5

#### D.4.1 Proof of Lemma D.5, part 1

The basic idea of the proof is the following. First we construct a $\epsilon$-net on semi-orthogonal matrices $\mathbf{V} \in \mathbb{R}^{d \times k}$, and argue that for each semi-orthogonal matrices $\mathbf{V}$ on the net, the set $\left\{\mathbf{X}_i \in G \,\middle|\, \text{Tr}\left[\mathbf{V}^\top (\mathbf{X}_i - \mathbf{M})\mathbf{V}\right] \leq \Theta(\nu\sqrt{k/\epsilon})\right\}$ satisfies the three conditions in the lemma. Then, for each matrix $\mathbf{V}'$ that is not on the set, we argue there exists a $G_\mathbf{V}$ with $\mathbf{V}$ on the $\epsilon$-net which satisfies the three conditions under $V'$ in the lemma. We show the three conditions for a matrix on the net as follows.

**Lemma D.8.** *Let $G = \{x_i \sim \mathcal{P}\}_{i=1}^{n}$ where $\mu^P$ is the mean, and $\sigma_P^2$ is the variance of a real distribution $\mathcal{P}$. For any real number $0 < \epsilon \leq 1/18$, define set $T := \left\{x_i \in G \,\middle|\, |x_i - \mu^P| \leq 3\sigma_P/\sqrt{\epsilon}\right\}$, and $\mu^T := \frac{1}{|T|} \sum_{x_i \in T} x_i$. Then with probability at least $1 - 3\exp(-\Theta(n\epsilon))$,*

1. $|T| \geq (1 - \epsilon)n$,

2. $\left|\mu^T - \mu^P\right| \leq \sigma_P \sqrt{\epsilon}$, and

3. $\frac{1}{|T|}\sum_{x_i \in T} \left(x_i - \mu^P\right)^2 \leq 2\sigma_P^2$.

We provide a proof in Section D.5.

**Proposition D.9** (Covering Number for Low-Rank Matrices, Lemma 3.1 in [12]). *Let $S_r := \{\mathbf{X} \in \mathbb{R}^{n_1 \times n_2} : \text{rank}(\mathbf{X}) \leq r, \|\mathbf{X}\|_F = 1\}$. Then there exists an $\epsilon$-net $\bar{S}_{r,\epsilon} \subset S_r$ with respect to Frobenius norm obeying*

$$
\left|\bar{S}_{r,\epsilon}\right| \leq (9/\epsilon)^{(n_1 + n_2 + 1)r}.
$$

With a union bound over all the elements of the net, we show that for each matrix $\mathbf{V}$ in the net, set $\left\{ \mathbf{X}_i \in G \,\middle|\, \mathrm{Tr}\big[\mathbf{V}^\top(\mathbf{X}_i - \mathbf{M})\mathbf{V}\big] \leq \Theta(\nu\sqrt{k/\epsilon}) \right\}$ satisfies the three conditions in the lemma. The following technical proposition will helps us connect an arbitrary semi-orthogonal matrix $\mathbf{V}'$ to the net.

**Proposition D.10.** *With probability $1 - \delta$, all subset $T \subset G$ such that $|T| \geq (1 - \epsilon)n$ satisfies $\frac{1}{|T|}\sum_{\mathbf{X}_i \in T}\|\mathbf{X}_i - \mathbf{M}\|_{\mathrm{F}}^2 \leq \nu^2 d^2/\delta(1 - \epsilon)$*

*Proof.* For each $i, j \in [d]$, define matrix $\mathbf{E}^{i,j} \in \mathbb{R}^{d \times d}$ such that
$$
E^{i,j}_{i',j'} = \begin{cases} 1 & \text{if } i' = i \text{ and } j' = j, \\ 0 & \text{otherwise.} \end{cases}
$$
Then,
$$
\mathop{\mathbb{E}}_{\mathbf{X} \sim \mathcal{P}}\Big[\|\mathbf{X} - \mathbf{M}\|_{\mathrm{F}}^2\Big] = \sum_{i,j \in [d]} \mathop{\mathbb{E}}_{\mathbf{X} \sim \mathcal{P}}\Big[\mathrm{Tr}[\mathbf{E}_{i,j}(\mathbf{X} - \mathbf{M})]^2\Big]
$$
$$
\leq \nu^2 d^2,
$$
where the last inequality follows from the assumption in Lemma D.5. By Markov's inequality,
$$
\mathbb{P}\left[\frac{1}{n}\sum_{i=1}^n \|\mathbf{X}_i - \mathbf{M}\|_{\mathrm{F}}^2 \geq \nu^2 d^2/\delta\right] \leq \frac{\delta}{\nu^2 d^2}\,\mathbb{E}\left[\frac{1}{n}\sum_{i=1}^n \|\mathbf{X}_i - \mathbf{M}\|_{\mathrm{F}}^2\right]
$$
$$
\leq \delta. \tag{53}
$$
Then for any subset $T \subset G$, and $|T| \geq (1 - \epsilon)n$,
$$
\frac{1}{|T|}\sum_{\mathbf{X}_i \in T}\|\mathbf{X}_i - \mathbf{M}\|_{\mathrm{F}}^2 \leq \frac{1}{|T|}\sum_{\mathbf{X}_i \in G}\|\mathbf{X}_i - \mathbf{M}\|_{\mathrm{F}}^2 \leq \nu^2 d^2/\delta(1 - \epsilon). \qquad \square
$$

We show Lemma D.5, part 1 as follows. By Proposition D.9, there exists an $\delta'$-net of size $(9\sqrt{k}/\delta')^{(k+d+1)k}$ over the set $S_k = \{\mathbf{V} \in \mathbb{R}^{d \times k} : \mathrm{rank}(\mathbf{V}) \leq k, \|\mathbf{V}\|_{\mathrm{F}} = \sqrt{k}\}$. Since the set of rank $k$ projection matrices $P_k := \{\mathbf{V} \in \mathbb{R}^{d \times k}, \mathbf{V}^\top\mathbf{V} = \mathbf{I}_k\} \subset S_k$, there exists a $\delta'$-net $\bar{P}_k$ of size $\left(18\sqrt{k}/\delta'\right)^{(k+d+1)k} = \exp(\Theta(dk\log(k/\delta')))$ of the set $P_k$. Fixing a projection matrix $\mathbf{V}$, the distribution of $\mathbf{V}^\top\mathbf{X}_i\mathbf{V}$ satisfies $\mathbb{E}\big[\mathrm{Tr}\big[\mathbf{V}^\top\mathbf{X}_i\mathbf{V}\big]\big] = \mathrm{Tr}\big[\mathbf{V}^\top\mathbf{M}\mathbf{V}\big]$ and
$$
\mathbb{E}\Big[\mathrm{Tr}\big[\mathbf{V}^\top(\mathbf{X}_i - \mathbf{M})\mathbf{V}\big]^2\Big] = k^2\,\mathbb{E}\left[\mathrm{Tr}\left[\frac{1}{k}\mathbf{V}\mathbf{V}^\top(\mathbf{X}_i - \mathbf{M})\right]^2\right] \leq k\nu^2.
$$

Thus, we can apply Lemma D.8 with a union bound to show that, given that
$$
n = \Omega(dk^2\log(k/\delta\delta')/\epsilon),
$$
with probability $1 - \delta$, for each projection matrix $\mathbf{V} \in \bar{P}_k$, there exist a subset of random matrices $G_{\mathbf{V}} \subset G$ such that $G_{\mathbf{V}} := \left\{ X_i \in G \,\middle|\, \big|\mathrm{Tr}\big[\mathbf{V}^\top(\mathbf{X}_i - \mathbf{M})\mathbf{V}\big]\big| \leq 3\nu\sqrt{k/\epsilon} \right\}$, and it satisfies

1. $|G_{\mathbf{V}}| \geq (1 - \epsilon)n$,
2. $\left|\frac{1}{|G_{\mathbf{V}}|}\sum_{\mathbf{X}_i \in G_{\mathbf{V}}} \mathrm{Tr}\big[\mathbf{V}^\top(\mathbf{X}_i - \mathbf{M})\mathbf{V}\big]\right| \leq \nu\sqrt{k\epsilon}$, and
3. $\frac{1}{|G_{\mathbf{V}}|}\sum_{\mathbf{X}_i \in G_{\mathbf{V}}} \mathrm{Tr}\big[\mathbf{V}^\top(\mathbf{X}_i - \mathbf{M})\mathbf{V}\big]^2 \leq 2k\nu^2$.

From now on, let us set
$$
\delta' = \sqrt{\epsilon\delta}/100d.
$$
The remaining argument conditions on that the event in Proposition D.10, which happens with probability $1 - \delta$, namely, for all $G \subset S$ such that $|G| \geq (1 - \epsilon)n$,
$$
\frac{1}{|G|}\sum_{\mathbf{X}_i \in G}\|\mathbf{X}_i - \mathbf{M}\|_{\mathrm{F}}^2 \leq \nu^2 d^2/\delta(1 - \epsilon). \tag{54}
$$

The following proposition deal with the projections that are not in $\bar{P}_k$.

**Proposition D.11.** *For any arbitrary projection matrix $\mathbf{V}'$, which may not be an element of $\bar{P}_k$, define projection matrix $\mathbf{V} \in \bar{P}_k$ such that $\|\mathbf{V} - \mathbf{V}'\|_{\mathrm{F}} \leq \delta$. Then $G_{\mathbf{V}}$ is still an $\epsilon$-good set for projection $\mathbf{V}'$, namely*

1. $|G_{\mathbf{V}}| \geq (1 - \epsilon)n$,

2. $\left| \frac{1}{|G_{\mathbf{V}}|} \sum_{\mathbf{X}_i \in G_{\mathbf{V}}} \mathrm{Tr}\left[ \mathbf{V'}^{\top} (\mathbf{X}_i - \mathbf{M})\mathbf{V}' \right] \right| \leq 1.01\nu\sqrt{k\epsilon}$, *and*

3. $\frac{1}{|G_{\mathbf{V}}|} \sum_{\mathbf{X}_i \in G_{\mathbf{V}}} \mathrm{Tr}\left[ \mathbf{V'}^{\top} (\mathbf{X}_i - \mathbf{M})\mathbf{V}' \right]^2 \leq 6.01 k\nu^2$.

*Proof.*

1. $|G_{\mathbf{V}}| \geq (1 - \epsilon)n$ holds trivially.

2. Observe that

$$
\left| \frac{1}{|G_{\mathbf{V}}|} \sum_{\mathbf{X}_i \in G_{\mathbf{V}}} \mathrm{Tr}\left[ \mathbf{V'}^{\top} (\mathbf{X}_i - \mathbf{M})\mathbf{V}' \right] \right|
$$

$$
\leq \frac{1}{|G_{\mathbf{V}}|} \left( \mathrm{Tr}\left[ \sum_{\mathbf{X}_i \in G_{\mathbf{V}}} \left( \mathbf{V'}^{\top} (\mathbf{X}_i - \mathbf{M})(\mathbf{V}' - \mathbf{V}) \right) \right] \right.
$$

$$
\left. + \mathrm{Tr}\left[ \sum_{\mathbf{X}_i \in G_{\mathbf{V}}} \left( (\mathbf{V}' - \mathbf{V})^{\top} (\mathbf{X}_i - \mathbf{M})\mathbf{V} \right) \right] + \mathrm{Tr}\left[ \sum_{\mathbf{X}_i \in G_{\mathbf{V}}} \left( \mathbf{V}^{\top} (\mathbf{X}_i - \mathbf{M})\mathbf{V} \right) \right] \right)
$$

$$
\leq \frac{2}{|G_{\mathbf{V}}|} \sum_{\mathbf{X}_i \in G_{\mathbf{V}}} \|\mathbf{X}_i - \mathbf{M}\|_{\mathrm{F}} \delta' + \nu\sqrt{k\epsilon}
$$

$$
\leq 2\delta' \sqrt{\frac{1}{|G_{\mathbf{V}}|} \sum_{\mathbf{X}_i \in G_{\mathbf{V}}} \|\mathbf{X}_i - \mathbf{M}\|_{\mathrm{F}}^2} + \nu\sqrt{k\epsilon}
$$

$$
\leq 1.01\nu\sqrt{k\epsilon}
$$

3. Notice that

$$
\frac{1}{|G_{\mathbf{V}}|} \sum_{\mathbf{X}_i \in G_{\mathbf{V}}} \mathrm{Tr}\left[ \left( \mathbf{V'}^{\top} (\mathbf{X}_i - \mathbf{M})\mathbf{V}' \right) \right]^2
$$

$$
= \frac{1}{|G_{\mathbf{V}}|} \sum_{\mathbf{X}_i \in G_{\mathbf{V}}} \left( \mathrm{Tr}\left[ (\mathbf{V'}^{\top} (\mathbf{X}_i - \mathbf{M})(\mathbf{V}' - \mathbf{V})) \right] + \mathrm{Tr}\left[ \left( (\mathbf{V}' - \mathbf{V})^{\top} (\mathbf{X}_i - \mathbf{M})\mathbf{V} \right) \right] \right.
$$

$$
\left. + \mathrm{Tr}\left[ \left( \mathbf{V}^{\top} (\mathbf{X}_i - \mathbf{M})\mathbf{V} \right) \right] \right)^2
$$

$$
\leq \frac{3}{|G_{\mathbf{V}}|} \sum_{\mathbf{X}_i \in G_{\mathbf{V}}} \left( \mathrm{Tr}\left[ \left( \mathbf{V'}^{\top} (\mathbf{X}_i - \mathbf{M})(\mathbf{V}' - \mathbf{V}) \right) \right]^2 \right.
$$

$$
\left. + \mathrm{Tr}\left[ \left( (\mathbf{V}' - \mathbf{V})^{\top} (\mathbf{X}_i - \mathbf{M})\mathbf{V} \right) \right]^2 + \mathrm{Tr}\left[ \left( \mathbf{V}^{\top} (\mathbf{X}_i - \mathbf{M})\mathbf{V} \right) \right]^2 \right)
$$

$$
\tag{55}
$$

Notice that by Cauchy-Schwarz,

$$
\mathrm{Tr}\left[ \left( \mathbf{V'}^{\top} (\mathbf{X}_i - \mathbf{M})(\mathbf{V}' - \mathbf{V}) \right) \right]^2 \leq \left\| (\mathbf{V}' - \mathbf{V})^{\top} \mathbf{V}' \right\|_{\mathrm{F}}^2 \|\mathbf{X}_i - \mathbf{M}\|_{\mathrm{F}}^2 \leq \delta'^2 \|\mathbf{X}_i - \mathbf{M}\|_{\mathrm{F}}^2,
$$

and likewise

$$\mathrm{Tr}\left[\left((\mathbf{V}' - \mathbf{V})^\top(\mathbf{X}_i - \mathbf{M})\mathbf{V}\right)\right]^2 \leq \delta'^2\|\mathbf{X}_i - \mathbf{M}\|_{\mathrm{F}}^2.$$

By Equation (54), Equation (55) is bounded by

$$6\nu^2 k + \nu^2\epsilon/10000 \leq 6.01k\nu^2. \qquad \square$$

This completes all the proofs.

### D.4.2 Proof of Lemma D.5, part 2

*Proof of Lemma D.5, part 2.* Since $\|\mathbf{X} - \mathbf{M}\|_2 \leq B$, and $\max_{\|\mathbf{A}\|_* \leq 1} \mathbb{E}\left[(\mathrm{Tr}[\mathbf{A}(\mathbf{X}_i - \mathbf{M})])^2\right] \leq \nu^2$ for all $i \in G$, therefore, from Bernstein inequality, we have

$$\mathbb{P}\left[\left\|\frac{1}{n}\sum_{i \in G}\mathbf{X}_i - \mathbf{M}\right\|_2 \geq \nu\sqrt{\epsilon/k}\right] \leq 2d\exp\left[\frac{-n^2\nu^2\epsilon/(2k)}{nd\nu^2 + Bn\nu\sqrt{\epsilon/k}/3}\right] \leq \delta \qquad (56)$$

Define matrix $\mathbf{A}^{(1,\mathbf{v})}, \mathbf{A}^{(2,\mathbf{v})}, \ldots, \mathbf{A}^{(d,\mathbf{v})}$ such that $\mathbf{A}_{i,*}^{(i,\mathbf{v})} = \mathbf{v}$, and the other rows of $\mathbf{A}^{(i,\mathbf{v})}$ are 0.

$$\left\|\mathbb{E}\left[\sum_{i=1}^n(\mathbf{X}_i - \mathbf{M})^2\right]\right\|_2 = \max_{\|\mathbf{v}\|_2=1}\sum_{i=1}^n \mathbb{E}\left[\mathbf{v}^\top(\mathbf{X}_i - \mathbf{M})(\mathbf{X}_i - \mathbf{M})\mathbf{v}\right]$$

$$= \max_{\|\mathbf{v}\|_2=1}\sum_{i=1}^n\sum_{j=1}^d \mathbb{E}\left[\mathrm{Tr}\left[\mathbf{A}^{(j,\mathbf{v})}(\mathbf{X}_i - \mathbf{M})\right]^2\right]$$

$$\leq nd\nu^2$$

if $n \geq \frac{2}{\epsilon}\left(dk + \frac{B}{3\nu}\sqrt{\epsilon k}\right)\log\frac{2d}{\delta}$. Therefore for any semi-orthogonal matrix $\mathbf{V} \in \mathbb{R}^{d \times k}$,

$$\left|\frac{1}{n}\sum_{\mathbf{X}_i \in G}\mathrm{Tr}\left[\mathbf{V}^\top(\mathbf{X}_i - \mathbf{M})\mathbf{V}\right]\right| \leq \left|\frac{1}{n}\sum_{\mathbf{X}_i \in G}\mathrm{Tr}[\mathcal{P}_k(\mathbf{X}_i - \mathbf{M})]\right|$$

$$\leq k\left\|\frac{1}{n}\sum_{i \in G}\mathbf{X}_i - \mathbf{M}\right\|_2$$

$$\leq \nu\sqrt{k\epsilon}, \qquad (57)$$

if $n = \Omega\left(\left(\frac{1}{\epsilon}\left(dk + \frac{B}{\nu}\sqrt{k\epsilon}\right)\right)\log\frac{d}{\delta}\right).$ $\qquad \square$

### D.4.3 Proof of Lemma D.5, part 3

Here we show that for any semi-orthogonal matrix $\mathbf{V} \in \mathbb{R}^{d \times k}$ on the net $\bar{P}_k$, and any subset $T$, $\sum_{\mathbf{X}_i \in T}\mathrm{Tr}\left[\mathbf{V}^\top(\mathbf{X}_i - \mathbf{M})\mathbf{V}\right] \leq n\nu\sqrt{k/\epsilon}$ condition on the event defined in Lemma D.5, part 1 happens.

Denote

$$G_L = \left\{\mathbf{X}_i \in G \,\middle|\, \mathrm{Tr}\left[\mathbf{V}^\top(\mathbf{X}_i - \mathbf{M})\mathbf{V}\right] < -3\nu\sqrt{k/\epsilon}\right\},$$

$$G_M = \left\{\mathbf{X}_i \in G \,\middle|\, -3\nu\sqrt{k/\epsilon} \leq \mathrm{Tr}\left[\mathbf{V}^\top(\mathbf{X}_i - \mathbf{M})\mathbf{V}\right] \leq 3\nu\sqrt{k/\epsilon}\right\},$$

$$G_H = \left\{\mathbf{X}_i \in G \,\middle|\, \mathrm{Tr}\left[\mathbf{V}^\top(\mathbf{X}_i - \mathbf{M})\mathbf{V}\right] > 3\nu\sqrt{k/\epsilon}\right\}.$$

Given a subset $T$, with $|T| \leq \epsilon n$ let $T_L = \left\{\mathbf{X}_i \in T \,\middle|\, \mathrm{Tr}\left[\mathbf{V}^\top(\mathbf{X}_i - \mathbf{M})\mathbf{V}\right] < 3\nu\sqrt{k/\epsilon}\right\}$ and $T_H = T \setminus T_L$. By definition, $\sum_{\mathbf{X}_i \in T_L}\mathrm{Tr}\left[\mathbf{V}^\top(\mathbf{X}_i - \mathbf{M})\mathbf{V}\right] \leq 3\nu\sqrt{k/\epsilon}|T_L|.$

$$\sum_{\mathbf{X}_i \in T_H} \mathrm{Tr}\left[\mathbf{V}^\top (\mathbf{X}_i - \mathbf{M})\mathbf{V}\right]$$

$$\leq \sum_{\mathbf{X}_i \in G_H} \mathrm{Tr}\left[\mathbf{V}^\top (\mathbf{X}_i - \mathbf{M})\mathbf{V}\right] \qquad (\because T_H \subseteq G_H)$$

$$\leq \sum_{\mathbf{X}_i \in G_H} \mathrm{Tr}\left[\mathbf{V}^\top (\mathbf{X}_i - \mathbf{M})\mathbf{V}\right] + \sum_{\mathbf{X}_i \in G_L} \mathrm{Tr}\left[\mathbf{V}^\top (\mathbf{X}_i - \mathbf{M})\mathbf{V}\right] + |G_L| \, \mathrm{Tr}\left[\mathbf{V}^\top \mathbf{M}\mathbf{V}\right]$$

$$= \sum_{\mathbf{X}_i \in G} \mathrm{Tr}\left[\mathbf{V}^\top (\mathbf{X}_i - \mathbf{M})\mathbf{V}\right] - \sum_{\mathbf{X}_i \in G_M} \mathrm{Tr}\left[\mathbf{V}^\top (\mathbf{X}_i - \mathbf{M})\mathbf{V}\right] + |G_L| \, \mathrm{Tr}\left[\mathbf{V}^\top \mathbf{M}\mathbf{V}\right]$$

$$\leq \sum_{\mathbf{X}_i \in G} \mathrm{Tr}\left[\mathbf{V}^\top (\mathbf{X}_i - \mathbf{M})\mathbf{V}\right] - \sum_{\mathbf{X}_i \in G_M} \mathrm{Tr}\left[\mathbf{V}^\top (\mathbf{X}_i - \mathbf{M})\mathbf{V}\right] + n\epsilon \, \mathrm{Tr}\left[\mathbf{V}^\top \mathbf{M}\mathbf{V}\right]$$

(Using Lemma D.5, part 1(a))

$$\leq 3n\nu\sqrt{k\epsilon} + n\epsilon \, \mathrm{Tr}\left[\mathbf{V}^\top \mathbf{M}\mathbf{V}\right],$$

where the last inequality holds since $\sum_{\mathbf{X}_i \in G} \mathrm{Tr}\left[\mathbf{V}^\top (\mathbf{X}_i - \mathbf{M})\mathbf{V}\right] \leq n\nu\sqrt{k\epsilon}$ by Lemma D.5, part 2, and $\sum_{\mathbf{X}_i \in G_M} \mathrm{Tr}\left[\mathbf{V}^\top (\mathbf{X}_i - \mathbf{M})\mathbf{V}\right] \leq 2n\nu\sqrt{k\epsilon}$ by Lemma D.5, part 1(b) with $G_M = G_\mathbf{V}$ when $\mathbf{V}$ is the net $\bar{P}_k$. Combining the bound on $T_L$ and $T_H$ yields that $\sum_{\mathbf{X}_i \in T} \mathrm{Tr}\left[\mathbf{V}^\top (\mathbf{X}_i - \mathbf{M})\mathbf{V}\right] \leq 6n\nu\sqrt{k\epsilon} + n\epsilon \, \mathrm{Tr}\left[\mathbf{V}^\top \mathbf{M}\mathbf{V}\right]$.

For a projection matrix $\mathbf{V}'$ not on the $\delta'$-net $\bar{P}_k$,

$$\sum_{\mathbf{X}_i \in T} \mathrm{Tr}\left[\mathbf{V}'^\top (\mathbf{X}_i - \mathbf{M})\mathbf{V}'\right]$$

$$= \left( \mathrm{Tr}\left[\sum_{\mathbf{X}_i \in T} \left(\mathbf{V}'^\top (\mathbf{X}_i - \mathbf{M})(\mathbf{V}' - \mathbf{V})\right)\right] + \mathrm{Tr}\left[\sum_{\mathbf{X}_i \in T} \left((\mathbf{V}' - \mathbf{V})^\top (\mathbf{X}_i - \mathbf{M})\mathbf{V}\right)\right] \right.$$

$$\left. + \mathrm{Tr}\left[\sum_{\mathbf{X}_i \in T} \left(\mathbf{V}^\top (\mathbf{X}_i - \mathbf{M})\mathbf{V}\right)\right] \right)$$

$$\leq 2 \sum_{\mathbf{X}_i \in T} \|\mathbf{X}_i - \mathbf{M}\|_F \delta' + 6n\sqrt{k\epsilon} + n\epsilon \, \mathrm{Tr}\left[\mathbf{V}^\top \mathbf{M}\mathbf{V}\right]$$

$$\leq 2\delta'\sqrt{|T|}\sqrt{\sum_{\mathbf{X}_i \in T} \|\mathbf{X}_i - \mathbf{M}\|_F^2} + 6n\sqrt{k\epsilon} + n\epsilon \, \mathrm{Tr}\left[\mathbf{V}^\top \mathbf{M}\mathbf{V}\right]$$

$$\leq \epsilon\nu n + 6\nu n\sqrt{k\epsilon} + n\epsilon \, \mathrm{Tr}\left[\mathbf{V}^\top \mathbf{M}\mathbf{V}\right]$$

$$\leq 7\nu n\sqrt{k\epsilon} + n\epsilon \, \mathrm{Tr}\left[\mathbf{V}^\top \mathbf{M}\mathbf{V}\right] \tag{58}$$

## D.5 Proof of Lemma D.8

1. Using Markov's inequality we get

$$\mathbb{P}_{x\sim\mathcal{P}}\left[\left|x - \mu^P\right| \geq z\right] \leq \frac{\mathbb{E}_{x\sim\mathcal{P}}\left[(x - \mu^P)^2\right]}{z^2}$$

$$\leq \frac{\sigma_P^2}{z^2}$$

$$\implies \mathbb{P}_{x\sim\mathcal{P}}\left[\left|x - \mu^P\right| \geq \sigma_P/\sqrt{\epsilon}\right] \leq \epsilon.$$

Let us define the indicator random variable $Z_i := \mathbb{1}\left\{\left|x_i - \mu^P\right| \geq \sigma_P/\sqrt{\epsilon}\right\}$, and let $p := \mathbb{E}[Z_i]$. Then from the Chernoff bound we get,

$$\mathbb{P}_{G \sim \mathcal{P}^n}\left[\frac{1}{n}\sum_{i=1}^{n} Z_i \geq (1+z)p\right] \leq \exp\left[-\frac{z^2 pn}{3}\right].$$

Set $(1+z)p = 2\epsilon$, we get

$$\mathbb{P}_{G \sim \mathcal{P}^n}\left[\frac{1}{n}\sum_{i=1}^{n} Z_i \geq 2\epsilon\right] \leq \exp\left[-\frac{(2\epsilon/p - 1)^2 pn}{3}\right]$$

$$\leq \exp\left[-\frac{\epsilon n}{3}\right],$$

where the last inequality holds since $p \leq \epsilon$. This implies

$$\mathbb{P}[|T| \leq (1 - 2\epsilon)n] \leq \exp[-\epsilon n/3]$$

2. Define event $\mathcal{E} := \left\{x : \left|x - \mu^P\right| \leq \sigma_P/\sqrt{\epsilon}\right\}$. In order to show that $\mu^T = \frac{1}{|T|}\sum_{x_i \in T} x_i$ concentrates to $\mu^P$, we will (1) apply Bernstein inequality to argue that $\mu^T$ concentrate around $\mu^{P'}$ which is the mean of $\mathcal{P}'$, the distribution of $x$ conditioned on the event $\mathcal{E}$, and (2) argue that the mean $\mu^{P'}$ is close to $\mu^P$, which, by triangle inequality, concludes the proof.

First, we prove a bound on $\left|\mu^{P'} - \mu^P\right|$, thus finishing part (2) of the proof.

**Proposition D.12.**

$$\left|\mu^{P'} - \mu^P\right| \leq 2\sqrt{\epsilon}\sigma_P.$$

*Proof.* Notice that

$$\left|\mu^{P'} - \mu^P\right| = \left|\mathbb{E}_{X \sim \mathcal{P}}[X \mid \mathcal{E}] - \mu^P\right|$$

$$= \left|\mathbb{E}_{X \sim \mathcal{P}}[X - \mu^P \mid \mathcal{E}]\right|$$

$$= \frac{1}{\mathbb{P}[\mathcal{E}]}\left|\mathbb{E}_{X \sim \mathcal{P}}\left[(X - \mu^P)\mathbb{1}\left\{\left|X - \mu^P\right| \leq \sigma_P/\sqrt{\epsilon}\right\}\right]\right|$$

$$= \frac{1}{\mathbb{P}[\mathcal{E}]}\left|\mathbb{E}_{X \sim \mathcal{P}}\left[(X - \mu^P)\mathbb{1}\left\{\left|X - \mu^P\right| \geq \sigma_P/\sqrt{\epsilon}\right\}\right]\right|$$

$$\leq \frac{1}{\mathbb{P}[\mathcal{E}]}\sqrt{\mathbb{E}_{X \sim \mathcal{P}}\left[(X - \mu^P)^2\right]\mathbb{E}_{X \sim \mathcal{P}}\left[\mathbb{1}\left\{\left|X - \mu^P\right| \geq \sigma_P/\sqrt{\epsilon}\right\}\right]} \quad (59)$$

$$\leq \frac{\sqrt{1 - \mathbb{P}[\mathcal{E}]}}{\mathbb{P}[\mathcal{E}]}\sigma_P \quad (60)$$

$$\leq 2\sqrt{\epsilon}\sigma_P,$$

where Equation (59) holds by Cauchy-Schwarz, and Equation (60) holds since

$$\mathbb{E}_{X \sim \mathcal{P}}\left[(X - \mu^P)^2\right] \leq \sigma_P^2. \qquad \square$$

To show part (1), let us first show the following simple fact due to Bernstein inequality

**Proposition D.13.** *Given $n$ iid samples $X_1, \ldots, X_n$ from distribution $\mathcal{P}'$, it holds that*

$$\mathbb{P}_{X_i \sim \mathcal{P}'}\left[\left|\frac{1}{n}\sum_{i=1}^{n} X_i - \mu^{P'}\right| \geq \sigma_P\sqrt{\epsilon}\right] \leq 2\exp(-n\epsilon/13)$$

*Proof.* First we bound the variance of $\mathcal{P}'$,

$$\mathop{\mathbb{E}}_{X\sim\mathcal{P}'}\left[\left(X-\mu^{P'}\right)^2\right] = \frac{1}{\mathbb{P}[\mathcal{E}]}\mathop{\mathbb{E}}_{X\sim\mathcal{P}}\left[\left(X-\mu^{P'}\right)^2\mathbb{1}\left\{\left|X-\mu^P\right|\le\sigma_P/\sqrt{\epsilon}\right\}\right]$$

$$\le \frac{1}{\mathbb{P}[\mathcal{E}]}\mathop{\mathbb{E}}_{X\sim\mathcal{P}}\left[\left(X-\mu^{P'}\right)^2\right]$$

$$= \frac{1}{\mathbb{P}[\mathcal{E}]}\mathop{\mathbb{E}}_{X\sim\mathcal{P}}\left[\left(X-\mu^P\right)^2+\left(\mu^P-\mu^{P'}\right)^2\right]$$

$$\le \frac{1}{\mathbb{P}[\mathcal{E}]}\left(\sigma_P^2+4\epsilon\sigma_P^2\right)$$

$$\le 6\sigma_P^2.$$

Hence, we can apply Bernstein inequality (Proposition J.2) and get

$$\mathbb{P}_{X_i\sim\mathcal{P}'}\left[\left|\frac{1}{n}\sum_{i=1}^n X_i-\mu^{P'}\right|\ge\sigma_P\sqrt{\epsilon}\right]\le 2\exp\left[\frac{-n^2\sigma_P^2\epsilon/2}{6n\sigma_P^2+n\sigma_P^2/3}\right]$$

$$\le 2\exp(-n\epsilon/13). \qquad \square$$

Notice that condition on the size of set $T$, each $X_i\in T$ follows from distribution $\mathcal{P}'$ independently. Hence, we have that condition on the size of $T$, with probability at least $1-2\exp(-|T|\epsilon/13)$, it holds that $\left|\frac{1}{|T|}\sum_{X_i\in T}X_i-\mu^{P'}\right|\le\sigma_P\sqrt{\epsilon}$. Since we have shown that $|T|\ge n(1-2\epsilon)$ with probability $1-\exp(-n\epsilon/3)$, by a union bound, we conclude that with probability at-least $1-3\exp(-n\epsilon/13)$,

$$\left|\frac{1}{|T|}\sum_{X_i\in T}X_i-\mu^{P'}\right|\le\sigma_P\sqrt{\epsilon}.$$

Combining Proposition D.12 with triangle inequality yields that with probability $1-3\exp(-n\epsilon/13)$,

$$\left|\mu^G-\mu^P\right|=\left|\frac{1}{|T|}\sum_{X_i\in T}X_i-\mu^P\right|\le 3\sigma_P\sqrt{\epsilon}.$$

3. Let us define the function $y(x):=\left(x-\mu^P\right)\cdot\mathbb{1}\left\{\left|x-\mu^P\right|\le\sigma_P/\sqrt{\epsilon}\right\}$. This implies $|y(X)|\le\sigma_P/\sqrt{\epsilon}$. Then,

$$\mathop{\mathbb{E}}_{X\sim\mathcal{U}(G)}\left[y(X)^2\right]\le\left|\mathop{\mathbb{E}}_{X\sim\mathcal{U}(G)}\left[y(X)^2\right]-\mathop{\mathbb{E}}_{X\sim\mathcal{P}}\left[y(X)^2\right]\right|+\mathop{\mathbb{E}}_{X\sim\mathcal{P}}\left[y(X)^2\right] \qquad (61)$$

Looking at the above two terms individually, with the second term

$$\mathop{\mathbb{E}}_{x\sim\mathcal{P}}\left[y(x)^2\right]=\mathop{\mathbb{E}}_{x\sim\mathcal{P}}\left[\left(x-\mu^P\right)^2\cdot\mathbb{1}\left\{\left|x-\mu^P\right|\le\sigma_P/\sqrt{\epsilon}\right\}\right]$$

$$\le\mathop{\mathbb{E}}_{X\sim\mathcal{P}}\left[\left(x-\mu^P\right)^2\right]$$

$$\le\sigma_P^2. \qquad (62)$$

For the first term we apply [72, Lemma 5.44.] on the random variable $y(x)$ for $x\sim\mathcal{P}$, and get that with probability $1-\exp(-C\epsilon n)$ for a constant $C>0$,

$$\left|\mathop{\mathbb{E}}_{X\sim\mathcal{U}(G)}\left[y(x)^2\right]-\mathop{\mathbb{E}}_{x\sim\mathcal{P}}\left[y(x)^2\right]\right|\le\frac{\sigma_P^2}{2}. \qquad (63)$$

Applying the fact that $|T|\ge(1-2\epsilon)n$ holds with probability $1-\exp(-\epsilon n/3)$, and plugging Equation (62) and (63) in Equation (61), we get that with probability $1-\exp(-\Theta(\epsilon n))$

$$\frac{1}{|T|}\sum_{X_i\in T}\left(X_i-\mu^P\right)^2=\frac{n}{|T|}\mathop{\mathbb{E}}_{X\sim\mathcal{U}(G)}\left[y(X)^2\right]\le 2\sigma_P^2. \qquad (64)$$

Taking a union bound of the probability of the three conditions and replacing $\epsilon$ by $\epsilon/9$ yield the statement of the lemma.

## D.6 Proof of Lemma D.3

We claim that

$$\left\|\left(\mathbf{I} - \widehat{\mathbf{U}}\widehat{\mathbf{U}}^\top\right)\mathbf{V}_j\right\|_* \leq \gamma/\sigma_j , \tag{65}$$

where $\mathbf{V}_j = [\mathbf{v}_1 \ldots \mathbf{v}_j]$ is the matrix consisting of the $j$ singular vectors of $\mathbf{M} = \sigma^2\mathbf{I} + \sum_{i'\in[k]}\mathbf{X}_{i'}$ corresponding to the top $j$ singular values, and $\sigma_j$ is the $j$-th singular value. This follows from the proof of the gap-free Wedin's theorem in [1, Lemma B.3], which proves a similar bound on the spectral norm. Concretely, let $\widehat{\mathbf{U}}_\perp \in \mathbb{R}^{d\times(d-k)}$ denote an orthogonal matrix spanning the null space of $\widehat{\mathbf{U}}^\top$. We can write the singular value decomposition as

$$\widehat{\mathbf{M}} = \widehat{\mathbf{U}}\widehat{\mathbf{D}}\widehat{\mathbf{U}}^\top , \quad \mathbf{B} = \mathbf{V}_j\mathbf{D}\mathbf{V}_j^\top + \mathbf{V}_j'\mathbf{D}'\mathbf{V}_j'^\top , \tag{66}$$

where $\mathbf{V}_j'$ spans the subspace orthogonal to $\mathbf{V}_j$, and $\mathbf{B} = \sum_{i=1}^k \mathbf{X}_i + \sigma^2\mathbf{U}\mathbf{U}^\top$. Let $\mathbf{R} = \widehat{\mathbf{M}} - \mathbf{B}$, and we get

$$
\begin{aligned}
\widehat{\mathbf{U}}_\perp^\top\widehat{\mathbf{M}}\mathbf{V}_j &= \widehat{\mathbf{U}}_\perp^\top\mathbf{R}\mathbf{V}_j + \widehat{\mathbf{U}}_\perp^\top\mathbf{B}\mathbf{V}_j \\
&= \widehat{\mathbf{U}}_\perp^\top\mathbf{R}\mathbf{V}_j + \widehat{\mathbf{U}}_\perp^\top\mathbf{V}_j\mathbf{D} .
\end{aligned}
$$

Since $\widehat{\mathbf{U}}_\perp^\top\widehat{\mathbf{M}} = \mathbf{0}$, taking nuclear norm and applying the triangular inequality,

$$
\begin{aligned}
\left\|\widehat{\mathbf{U}}_\perp^\top\mathbf{V}_j\right\|_* &= \left\|\widehat{\mathbf{U}}_\perp^\top\mathbf{R}\mathbf{V}_j\mathbf{D}^{-1}\right\|_* \\
&\leq \gamma/\sigma_j .
\end{aligned}
$$

To get the first term on the upper bound (29), we follow the analysis of [47, Lemma 5]. Notice that $\mathbf{x}_i$ lie on the subspace spanned by $\mathbf{V}_j$ where $j$ is the rank of $\sum_{i'\in[k]}\mathbf{X}_{i'}$. It follows from $\left\|\left(\mathbf{I} - \widehat{\mathbf{U}}\widehat{\mathbf{U}}^\top\right)\mathbf{V}_j\right\|_{\mathrm{F}} \leq \left\|\left(\mathbf{I} - \widehat{\mathbf{U}}\widehat{\mathbf{U}}^\top\right)\mathbf{V}_j\right\|_* \leq \gamma/\sigma_j$ with a choice of $j = k$ that

$$
\begin{aligned}
\sum_{i\in[k]}\left\|\left(\mathbf{I} - \widehat{\mathbf{U}}\widehat{\mathbf{U}}^\top\right)\mathbf{V}_j\mathbf{V}_j^\top\mathbf{x}_i\right\|_2^2 &= \mathrm{Tr}\left[\left(\mathbf{I} - \widehat{\mathbf{U}}\widehat{\mathbf{U}}^\top\right)\mathbf{V}_j\mathbf{V}_j^\top\left(\sum_{i\in[k]}\mathbf{x}_i\mathbf{x}_i^\top\right)\mathbf{V}_j\mathbf{V}_j^\top\left(\mathbf{I} - \widehat{\mathbf{U}}\widehat{\mathbf{U}}^\top\right)\right] \\
&\leq \left\|\sum_{i\in[k]}\mathbf{x}_i\mathbf{x}_i^\top\right\|_2 \left\|\mathbf{V}_j\mathbf{V}_j^\top\left(\mathbf{I} - \widehat{\mathbf{U}}\widehat{\mathbf{U}}^\top\right)\left(\mathbf{I} - \widehat{\mathbf{U}}\widehat{\mathbf{U}}^\top\right)\mathbf{V}_j\mathbf{V}_j^\top\right\|_* \\
&\leq \sigma_{\max}\gamma^2/\sigma_{\min}^2 .
\end{aligned}
$$

Next, we optimize over this choice of $j$ to get the tightest bound that does not depend on the singular values. Applying a similar bound as the above series of inequalities, we get

$$
\begin{aligned}
\sum_{i\in[k]}\left\|\left(\mathbf{I} - \widehat{\mathbf{U}}\widehat{\mathbf{U}}^\top\right)\mathbf{x}_i\right\|_2^2 &= \sum_{i\in[k]}\left\|\left(\mathbf{I} - \widehat{\mathbf{U}}\widehat{\mathbf{U}}^\top\right)\mathbf{V}_j\mathbf{V}_j^\top\mathbf{x}_i\right\|_2^2 + \sum_{i\in[k]}\left\|\left(\mathbf{I} - \widehat{\mathbf{U}}\widehat{\mathbf{U}}^\top\right)(\mathbf{I} - \mathbf{V}_j\mathbf{V}_j^\top)\mathbf{x}_i\right\|_2^2 \\
&\leq \left(\sigma_{\max}\gamma^2/\sigma_j^2\right) + (k-j)\sigma_{j+1} ,
\end{aligned}
$$

where we used the fact that

$$
\begin{aligned}
\sum_{i\in[k]}\left\|\left(\mathbf{I} - \widehat{\mathbf{U}}\widehat{\mathbf{U}}^\top\right)(\mathbf{I} - \mathbf{V}_j\mathbf{V}_j^\top)\mathbf{x}_i\right\|_2^2 &\leq \mathrm{Tr}\left[(\mathbf{I} - \mathbf{V}_j\mathbf{V}_j^\top)\left(\sum_{i\in[k]}\mathbf{x}_i\mathbf{x}_i^\top\right)(\mathbf{I} - \mathbf{V}_j\mathbf{V}_j^\top)\right] \\
&\leq (k-j)\sigma_{j+1} .
\end{aligned}
$$

A good choice of $j$ to approximately minimizes the upper bound is for the two terms to be of similar orders. Precisely, we choose $j$ to be the largest index such that $\sigma_j \geq \gamma^{2/3}\sigma_{\max}^{1/3}(k-j)^{-1/3}$ (we take $j = 0$ if $\sigma_1 \leq \gamma^{2/3}\sigma_{\max}^{1/3}k^{-1/3}$). This gives an upper bound of $2\gamma^{2/3}\sigma_{\max}^{1/3}k^{2/3}$.

The second upper bound in (29) follows from a similar argument, and is a direct corollary of [42, Lemma A.11].

# E   Proof sketch of the adaption to exponential tail setting

In this setting, we give a sketch of how to adapt the proof of Algorithm 2 to the setting where the distribution has exponential like tail.

**Closing the gap in Outlier-Robust PCA (ORPCA).** [75, 28, 76, 19] study robust PCA under the assumption that each sample $\mathbf{z}_i = \mathbf{A}\mathbf{x}_i + \mathbf{v}_i$ where $\mathbf{x}_i, \mathbf{v}_i$ are drawn from isotropic Gaussian distribution, and the goal is to learn the top-$k$ eigenspace of $\mathbf{A}\mathbf{A}^\top$. When $n$ samples are observed, $\alpha$ fraction of which are corrupted by an adversary, [75] introduces a filtering algorithm to find a subspace $\widehat{\mathbf{U}}$ achieving:

$$\left\| \widehat{\mathbf{U}}^\top \mathbf{A}\mathbf{A}^\top \widehat{\mathbf{U}} \right\|_* \geq (1 - c\sqrt{\alpha}\log(1/\alpha)) \left\| \mathbf{A}\mathbf{A}^\top \right\|_* ,$$

for some $c > 0$ when $n/k(\log k)^5 \to \infty$. For the reasons explained in §2.2, this is sub-optimal in $\alpha$ in comparisons to an information theoretic lower bound with a multiplicative factor of $(1 - c\alpha)$. Applying the proposed Algorithm 2, it is possible to generalize Proposition 2.6 to this Gaussian setting and achieve an optimal upper bound. To get some intuition, notice that with our assumption on the second moment of the projected variance $\mathrm{Tr}\left[\mathbf{U}^\top \mathbf{X}_i \mathbf{U}\right]$, our current proof proceeds by focusing on that $1 - \alpha$ probability mass, which falls in the interval $[-\Theta(\sqrt{1/\alpha}), \Theta(\sqrt{1/\alpha})]$. This is tight with only the second moment assumption. However, if we assume exponential tail on $\mathrm{Tr}\left[\mathbf{U}^\top \mathbf{X}_i \mathbf{U}\right]$, namely $\mathbb{P}\left[\mathrm{Tr}\left[\mathbf{U}^\top \mathbf{X}_i \mathbf{U}\right] > t\right] \leq \exp(-t^p)$, for some $p > 0$, we can instead focus on the probability mass in $[-\Theta(\log^{1/p}(1/\alpha)), \Theta(\log^{1/p}(1/\alpha))]$ which is also at least $1 - \alpha$. The would give an error of $\alpha \log^{1/p}(1/\alpha)$. We provide a sketch of how to adapt the proof of our algorithm to the exponential tail setting as follows.

**Lemma E.1** (Main Lemma for Algorithm 2, adaption from Lemma D.4). *Let $\mathcal{P}$ be a distribution over $d \times d$ PSD matrices with the property that,*

$$\mathbb{E}_{\mathbf{X}\sim\mathcal{P}}[\mathbf{X}] = \mathbf{M} , \quad \|\mathbf{X} - \mathbf{M}\|_2 \leq B ,$$

*and* $\quad \max_{\|\mathbf{A}\|_\mathrm{F}\leq 1,\mathrm{rank}(\mathbf{A})\leq k} \mathbb{P}_{\mathbf{X}\sim\mathcal{P}}[|\mathrm{Tr}[\mathbf{A}(\mathbf{X} - \mathbf{M})]| \geq \nu(k)t] \leq \exp(-t^p) ,$

*for some $p > 0$. Let a set of $n$ random matrices $G = \left\{\mathbf{X}_i \in \mathbb{R}^{d\times d}\right\}_{i\in[n]}$ where each $\mathbf{X}_i$ is independently drawn from $\mathcal{P}$, and the at most $\alpha$ fraction is corrupted by an adversary such that the input dataset $S = (G \setminus L) \cup E$ with $|E| = |L| \leq \alpha n$, $L \subset G$. There exists a numerical constant $c > 0$ such that for any $0 < \alpha < c$, if $n = \widetilde{\Omega}((dk^2 + (B/\nu)\sqrt{k}\alpha)/\alpha^2)$, Algorithm 2 outputs a dataset $S' \subseteq S$ satisfying the following for $\widehat{\mathbf{M}} = \frac{1}{|S'|}\sum_{\mathbf{X}_i\in S'} \mathbf{X}_i$:*

1. *for the top-$k$ singular vectors $\widehat{\mathbf{U}} \in \mathbb{R}^{d\times k}$ of $\widehat{\mathbf{M}}$,*

$$\mathrm{Tr}\left[\widehat{\mathbf{U}}^\top \left(\widehat{\mathbf{M}} - \mathbf{M}\right)\widehat{\mathbf{U}}\right] \leq \mathcal{O}\left(\alpha\,\mathrm{Tr}\left[\widehat{\mathbf{U}}^\top \mathbf{M}\widehat{\mathbf{U}}\right] + \nu\sqrt{k}\alpha\log(1/\alpha)^{1/p}\right) .$$

2. *for all rank-$k$ semi-orthogonal matrices $\mathbf{V} \in \mathbb{R}^{d\times k}$, we have*

$$\mathrm{Tr}\left[\mathbf{V}^\top \left(\widehat{\mathbf{M}} - \mathbf{M}\right)\mathbf{V}\right] \geq -\mathcal{O}\left(\alpha\,\mathrm{Tr}\left[\mathbf{V}^\top \mathbf{M}\mathbf{V}\right] + \nu\sqrt{k}\alpha\log(1/\alpha)^{1/p}\right) .$$

Notice that the probability mass beyond $\left|\mathrm{Tr}\left[\sum_{\mathbf{X}_i\in G_\mathbf{V}} \left(\mathbf{V}^\top (\mathbf{X}_i - \mathbf{M})\mathbf{V}\right)\right]\right| \geq \nu\sqrt{k}\log(1/\epsilon)^{1/p}$ is less than $\epsilon$ by the exponential tail bound. Hence similar to Lemma D.5, by letting $G_\mathbf{V} \subset G$ to contain all the points in $G$ such that $\left|\mathrm{Tr}\left[\sum_{\mathbf{X}_i\in G_\mathbf{V}} \left(\mathbf{V}^\top (\mathbf{X}_i - \mathbf{M})\mathbf{V}\right)\right]\right| \leq \nu\sqrt{k}\log(1/\epsilon)^{1/p}$, we have part 1 of the following lemma. Part 2 of the following lemma follows from matrix Bernstein inequality, and part 3 can be shown with the same argument of Lemma D.5.

**Lemma E.2** (Adaption from Lemma D.5). *Under the hypotheses of Lemma E.1, when $n = \widetilde{\Omega}((dk^2 + \frac{B}{\nu}\sqrt{k}\epsilon)/\epsilon^2)$ with probability $1 - \widetilde{\delta}$, the following events happen for all semi-orthogonal matrices $\mathbf{V} \in \mathbb{R}^{d\times k} s.t. \mathbf{V}^\top \mathbf{V} = \mathbf{I}_k$,*

1. *There exists $G_\mathbf{V} \subset G$ such that*

*(a)* $|G_{\mathbf{V}}| \geq (1 - \epsilon)n$,

*(b)* $\left| \frac{1}{|G_{\mathbf{V}}|} \operatorname{Tr}\left[ \sum_{\mathbf{X}_i \in G_{\mathbf{V}}} \left( \mathbf{V}^\top (\mathbf{X}_i - \mathbf{M})\mathbf{V} \right) \right] \right| \leq \nu \sqrt{k} \epsilon \log(1/\epsilon)^{1/p}$, *and*

*(c)* $\left| \operatorname{Tr}\left[ \sum_{\mathbf{X}_i \in G_{\mathbf{V}}} \left( \mathbf{V}^\top (\mathbf{X}_i - \mathbf{M})\mathbf{V} \right) \right] \right| \leq \nu \sqrt{k} \log(1/\epsilon)^{1/p}$,

2. $\left| \frac{1}{n} \sum_{\mathbf{X}_i \in G} \operatorname{Tr}\left[ \mathbf{V}^\top (\mathbf{X}_i - \mathbf{M})\mathbf{V} \right] \right| \leq \nu \sqrt{k} \epsilon \log(1/\epsilon)^{1/p}$,

3. *All subset $T \subset G$ such that $|T| \leq \epsilon n$ satisfies*

$$\sum_{\mathbf{X}_i \in T} \operatorname{Tr}\left[ \mathbf{V}^\top (\mathbf{X}_i - \mathbf{M})\mathbf{V} \right] \leq 7n\nu \sqrt{k} \epsilon \log(1/\epsilon)^{1/p} + n\epsilon \operatorname{Tr}\left[ \mathbf{V}^\top \mathbf{M} \mathbf{V} \right].$$

Thus by changing line 5 of Algorithm 3 to $48(\alpha \mu^{S_G} + \nu \sqrt{k}\alpha \log(1/\alpha)^{1/p})$, the statement in Proposition D.6 can be changed to that either the output of Algorithm 3 satisfies

$$\operatorname{Tr}\left[ \widehat{\mathbf{U}}^\top \left( \frac{1}{|S'|} \sum_{\mathbf{X}_i \in S'} \mathbf{X}_i - \mathbf{M} \right) \widehat{\mathbf{U}} \right] \leq \mathcal{O}\left( \alpha \operatorname{Tr}\left[ \widehat{\mathbf{U}}^\top \mathbf{M} \widehat{\mathbf{U}} \right] + \nu \sqrt{k}\alpha \log(1/\alpha)^{1/p} \right) \qquad (67)$$

or the algorithm removes more corrupted data points than uncorrupted data points in expectation. Finally, similar to Proposition 2.6, we get

$$\operatorname{Tr}[\mathcal{P}_k(\boldsymbol{\Sigma})] - \operatorname{Tr}\left[ \widehat{\mathbf{U}}^\top \boldsymbol{\Sigma} \widehat{\mathbf{U}} \right] = \mathcal{O}\left( \alpha \operatorname{Tr}[\mathcal{P}_k(\boldsymbol{\Sigma})] + \nu \sqrt{k}\alpha \log(1/\alpha)^{1/p} \right),$$

and $\quad \left\| \boldsymbol{\Sigma} - \widehat{\mathbf{U}}\widehat{\mathbf{U}}^\top \boldsymbol{\Sigma} \widehat{\mathbf{U}}\widehat{\mathbf{U}}^\top \right\|_* \leq \| \boldsymbol{\Sigma} - \mathcal{P}_k(\boldsymbol{\Sigma}) \|_* + \mathcal{O}\left( \alpha \| \mathcal{P}_k(\boldsymbol{\Sigma}) \|_* + \nu \sqrt{k}\alpha \log(1/\alpha)^{1/p} \right).$

## F Lower bound for robust PCA, proof of Proposition 2.7

In this section we show that under the setting of Proposition 2.6, it is information theoretically impossible to learn subspace $\widehat{\mathbf{U}}$ such that $\| \boldsymbol{\Sigma} - \widehat{\mathbf{U}}\widehat{\mathbf{U}}^\top \boldsymbol{\Sigma} \widehat{\mathbf{U}}\widehat{\mathbf{U}}^\top \| = o\left( \nu(k)\sqrt{k\alpha} \right)$.

**Definition F.1.** *Given a subset $I \subset [d], |I| = k$, define distribution $\mathcal{P}_I$ as follows: Suppose random variable $\mathbf{x} \sim \mathcal{P}_I$, and each coordinate $x_i, i \in I$ is sampled independently such that*

$$x_i = \begin{cases} \sqrt{\nu(k)} & \text{with probability } (1 - \alpha/k)/2 \\ -\sqrt{\nu(k)} & \text{with probability } (1 - \alpha/k)/2 \\ (\nu(k)^2 k/\alpha)^{1/4} & \text{with probability } \alpha/2k \\ -(\nu(k)^2 k/\alpha)^{1/4} & \text{with probability } \alpha/2k \end{cases}.$$

*The other coordinates $x_i, i \notin I$ is sampled independently such that*

$$x_i = \begin{cases} \sqrt{\nu(k)} & \text{with probability } 1/2 \\ -\sqrt{\nu(k)} & \text{with probability } 1/2 \end{cases}.$$

The second moment matrix $\boldsymbol{\Sigma}^I := \mathbb{E}_{\mathbf{x} \sim \mathcal{P}_I}\left[ \mathbf{x}\mathbf{x}^\top \right]$ of $\mathcal{P}_I$ satisfies

$$\Sigma^I_{i,j} = \begin{cases} \nu(k)(1 - \alpha/k + \sqrt{\alpha/k}) & i = j, i \in I \\ \nu(k) & i = j, i \notin I \\ 0 & i \neq j \end{cases}.$$

It is clear that with probability 1, $\left\| \mathbf{x}\mathbf{x}^\top - \boldsymbol{\Sigma}^I \right\|_2 \leq \| \mathbf{x} \|_2^2 + \left\| \boldsymbol{\Sigma}^I \right\|_2 \leq \nu(k)\left( d + 2k^{3/2}/\sqrt{\alpha} \right) \leq 2\nu(k)d = B$. Next we verify the fourth moment condition in Proposition 2.6. WLOG, let us assume $A_{i,j} = 0$ for any $j < i$, hence $\mathbf{A}$ is a upper triangular.

$$\mathbb{E}\left[ \operatorname{Tr}\left[ \mathbf{A}\left( \mathbf{x}\mathbf{x}^\top - \boldsymbol{\Sigma}^I \right) \right]^2 \right] = \mathbb{E}\left[ \left( \sum_{i,j} A_{i,j}\left( x_i x_j - \Sigma^I_{i,j} \right) \right)^2 \right]$$

$$= \mathbb{E}\left[ \sum_{i,j} \sum_{i',j'} A_{i,j}\left( x_i x_j - \Sigma^I_{i,j} \right) A_{i',j'}\left( x_{i'} x_{j'} - \Sigma^I_{i',j'} \right) \right]$$

Based on the number of distinct values $i, j, i', j'$ take, the terms inside the summation can be classified into 4 difference cases

1. $i, j, i', j'$ takes 4 difference values. In this case,
$$\mathbb{E}\big[A_{i,j}\big(x_i x_j - \Sigma_{i,j}^I\big) A_{i',j'}\big(x_{i'} x_{j'} - \Sigma_{i',j'}^I\big)\big] = 0$$

2. $i, j, i', j'$ takes 3 difference values. In this case,
$$\mathbb{E}\big[A_{i,j}\big(x_i x_j - \Sigma_{i,j}^I\big) A_{i',j'}\big(x_{i'} x_{j'} - \Sigma_{i',j'}^I\big)\big] = 0$$

3. $i, j, i', j'$ takes 2 difference values. In this case, if $i = j$, $i' = j'$ and $i \neq i'$,
$$\mathbb{E}\big[A_{i,i}\big(x_i^2 - \Sigma_{i,i}^I\big) A_{i',i'}\big(x_{i'}^2 - \Sigma_{i',i'}^I\big)\big] = 0.$$
If $i = i'$, $j = j'$ and $i \neq j$,
$$\mathbb{E}\Big[A_{i,j}^2\big(x_i x_j - \Sigma_{i,j}^I\big)^2\Big] = \mathbb{E}\big[A_{i,j}^2(x_i x_j)^2\big] = \mathbb{E}\big[A_{i,j}^2 \Sigma_{i,i}^I \Sigma_{j,j}^I\big].$$
If $i = j'$, $j = i'$ and $i \neq j$, $A_{i,j}$ or $A_{j,i}$ must be 0, hence the expectation is 0.

4. $i, j, i', j'$ takes 1 value. In this case
$$\mathbb{E}\Big[A_{i,i}^2\big(x_i^2 - \Sigma_{i,i}^I\big)^2\Big] = \begin{cases} 0 & i \notin I \\ A_{i,i}^2 \nu(k)^2(2 - \alpha/k) & i \in I \end{cases}$$

Taking summation over the above cases yields the following bound
$$\sum_{i<j} A_{i,j}^2 \Sigma_{i,i}^I \Sigma_{j,j}^I + \sum_{i \in I} A_{i,i}^2 (2 - \alpha/k) \leq 2\|\mathbf{A}\|_{\mathrm{F}}^2 \nu(k)^2 \leq 2\nu(k)^2.$$

Here we have shown that each distribution $\mathcal{P}_I$ satisfies
$$\max_{\|\mathbf{A}\|_{\mathrm{F}} \leq 1} \mathbb{E}_{\mathbf{x} \sim \mathcal{P}_I}\Big[\big(\langle \mathbf{A}, \mathbf{x}\mathbf{x}^\top - \mathbb{E}[\mathbf{x}\mathbf{x}^\top]\rangle\big)^2\Big] \leq 2\nu(k)^2.$$

Then we define the base case distribution $\mathcal{P}_\emptyset$ as follows:

**Definition F.2.** *Suppose random variable $\mathbf{x} \sim \mathcal{P}_\emptyset$, and each coordinate $x_i, i \in I$ is sampled independently*
$$x_i = \begin{cases} \sqrt{\nu(k)} & \text{with probability } 1/2 \\ -\sqrt{\nu(k)} & \text{with probability } 1/2 \end{cases}.$$

It is clear that $D_{\mathrm{TV}}(\mathcal{P}_I, \mathcal{P}_\emptyset) = \alpha$. Thus we have shown that each pair $(\mathcal{P}_\emptyset, \mathcal{P}_I) \in \Theta_{\nu(k), \alpha}$. Now let us fix an estimator $\widehat{\mathbf{U}}$, and let $\widehat{\mathbf{U}}_\emptyset = \widehat{\mathbf{U}}(\{\mathbf{x}_i\}_{i=1}^n)$, $\{\mathbf{x}_i\}_{i=1}^n \sim \mathcal{P}_\emptyset^n$ denote the random subspace when the datapoints are drawn from $\mathcal{P}_\emptyset$. WLOG, let us assume $d$ is a multiple of $k$, and let $I_1, \ldots, I_{d/k}$ be a partition of $[n]$. Notice that
$$\mathbb{E}\left[\sum_{i=1}^{d/k} \mathrm{Tr}\Big[\widehat{\mathbf{U}}_\emptyset^\top \Sigma_{I_i} \widehat{\mathbf{U}}_\emptyset\Big]\right] = \nu(k) \cdot k \cdot \Big(d/k + 1 - \alpha/k + \sqrt{\alpha/k}\Big),$$
and hence there exists a $i^*$ such that
$$\mathbb{E}\Big[\mathrm{Tr}\Big[\widehat{\mathbf{U}}_\emptyset^\top \Sigma_{I_{i^*}} \widehat{\mathbf{U}}_\emptyset\Big]\Big] \leq \nu(k) \cdot k \cdot \Big(d/k + 1 - \alpha/k + \sqrt{\alpha/k}\Big) \cdot (d/k)^{-1}$$
$$\leq \nu(k) \cdot k \cdot \left(1 + \frac{1 + \sqrt{\alpha/k}}{d/k}\right).$$

The sub-optimality can be expressed as
$$\mathbb{E}\Big[\Big\|\mathbf{\Sigma}_{I_{i^*}} - \widehat{\mathbf{U}}_\emptyset \widehat{\mathbf{U}}_\emptyset^\top \mathbf{\Sigma}_{I_{i^*}} \widehat{\mathbf{U}}_\emptyset \widehat{\mathbf{U}}_\emptyset^\top\Big\|_* - \|\mathbf{\Sigma}_{I_{i^*}} - \mathcal{P}_k(\mathbf{\Sigma}_{I_{i^*}})\|_*\Big]$$
$$= \nu(k)k\Big(1 - \alpha/k + \sqrt{\alpha/k}\Big) - \mathbb{E}\Big[\mathrm{Tr}\Big[\widehat{\mathbf{U}}_\emptyset^\top \Sigma_{I_{i^*}} \widehat{\mathbf{U}}_\emptyset\Big]\Big]$$
$$\geq \nu(k)\left(\sqrt{\alpha k} - \alpha - \frac{k^2(1 + \sqrt{\alpha/k})}{d}\right) \quad \text{(Using } k \geq 16, d \geq k^2/\alpha\text{)}$$
$$\geq \frac{1}{16}\nu(k)\sqrt{\alpha k}$$

This implies that for any subspace estimator $\widehat{\mathbf{U}}$, we can find distribution $\mathcal{D} = \mathcal{P}_\emptyset, \mathcal{D}' = \mathcal{P}_{I_{i*}}$ such that

1. $\displaystyle\mathop{\mathbb{E}}_{\{\mathbf{x}_i\}_{i=1}^n \sim \mathcal{D}^n} \left[ \left\| \mathbf{\Sigma}_{I_{i*}} - \widehat{\mathbf{U}}\widehat{\mathbf{U}}^\top \mathbf{\Sigma}_{I_{i*}} \widehat{\mathbf{U}}\widehat{\mathbf{U}}^\top \right\|_* - \left\| \mathbf{\Sigma}_{I_{i*}} - \mathcal{P}_k(\mathbf{\Sigma}_{I_{i*}}) \right\|_* \right] \geq \frac{1}{16} \nu(k)\sqrt{k\alpha}$,

2. $D_{\mathrm{TV}}(\mathcal{D}, \mathcal{D}') \leq \alpha$,

3. $\displaystyle\max_{\|\mathbf{A}\|_{\mathrm{F}} \leq 1} \mathop{\mathbb{E}}_{\mathbf{x} \sim \mathcal{D}'} \left[ \left( \langle \mathbf{A}, \mathbf{x}\mathbf{x}^\top - \mathbb{E}\left[ \mathbf{x}\mathbf{x}^\top \right] \rangle \right)^2 \right] \leq 2\nu(k)^2$.

The proof is complete.

# G  Robust clustering algorithm

**Definition G.1.** *Pseudo-distributions are generalizations of probability distributions except for the fact that they need not be non-negative. A level-$2m$ pseudo-distribution $\xi$, for $m \in \mathbb{N} \cup \{\infty\}$, is a measurable function that must satisfy*

$$\int_{\mathbb{R}^d} q(\mathbf{x})^2 d\xi(\mathbf{x}) \geq 0 \qquad \text{for all polynomials } q \text{ of degree at-most } m, \text{ and} \tag{68}$$

$$\int_{\mathbb{R}^d} d\xi(\mathbf{x}) = 1. \tag{69}$$

*A straightforward polynomial interpolation argument shows that every level-$\infty$ pseudo-distribution $\xi$ is non-negative, and thus are actual probability distributions.*

**Definition G.2.** *A pseudo-expectation $\widetilde{\mathbb{E}}_\xi[f(\mathbf{x})]$ of a function $f$ on $\mathbb{R}^d$ with respect to a pseudo-expectation $\xi$, just like the usual expectation, is denoted as*

$$\widetilde{\mathbb{E}}_\xi[f(\mathbf{x})] = \int_{\mathbb{R}^d} f(\mathbf{x}) d\xi(\mathbf{x}).$$

**Definition G.3.** *The SOS ordering $\preceq_{SOS}$ between two finite dimensional tensors $\mathcal{T}_1$, and $\mathcal{T}_2$, i.e., $\mathcal{T}_1 \preceq_{SOS} \mathcal{T}_2$, means $\langle \mathcal{T}_1, \mathbf{v}^{\otimes 2m} \rangle \preceq_{2m} \langle \mathcal{T}_2, \mathbf{v}^{\otimes 2m} \rangle$ as polynomial in $\mathbf{v}$.*

Given $\left\{ \widehat{\beta}_i \right\}_{i=1}^n$, we want to find $\{\gamma_i\}_{i=1}^n$ such that $\frac{1}{n}\sum_{i=1}^n \widetilde{\mathbb{E}}_\xi \left[ \left\langle \widehat{\beta}_i - \gamma_i, \mathbf{v} \right\rangle^{2m} \right]$ is small for all pseudo-distributions $\xi$ of $\mathbf{v}$ over the sphere. Since $\gamma_i = \widehat{\beta}_i \ \forall \ i \in [n]$ would be an over-fit, therefore to avoid it, it turns out that the natural way is to introduce the term $\sum_{i=1}^n \langle \gamma_i - \beta_i, \gamma_i \rangle^{2m}$ which must be small at the same time. On the other hand, we know that if $\sum_{i \in \mathcal{G}} \widetilde{\mathbb{E}}_\xi \left[ \left\langle \widehat{\beta}_i - \gamma_i, \mathbf{v} \right\rangle^{2m} \right]$, and $\sum_{i \in \mathcal{G}} \widetilde{\mathbb{E}}_\xi \left[ \left\langle \widehat{\beta}_i - \beta_i, \mathbf{v} \right\rangle^{2m} \right]$ (from the SOS proof) are small, then from the Minkowski's inequality, $\sum_{i \in \mathcal{G}} \widetilde{\mathbb{E}}_\xi \left[ \langle \gamma_i - \beta_i, \mathbf{v} \rangle^{2m} \right]$ will also be small. To make this hold, it is sufficient to impose that whenever $\{\mathbf{z}_i\}_{i=1}^n$ are such that $\sum_{i=1}^n \widetilde{\mathbb{E}}_\xi \left[ \langle \mathbf{z}_i, \mathbf{v} \rangle^{2m} \right] \leq 1$ for all pseudo-distributions $\xi$ over the unit sphere, then $\sum_{i=1}^n \langle \mathbf{z}_i, \gamma_i \rangle^{2m}$ is also small. This, however, is not efficiently imposable, but there is a standard SOS way of relaxing this, which is to require $\sum_{i=1}^n \widetilde{\mathbb{E}}_{\zeta_i} \left[ \langle \mathbf{z}_i, \gamma_i \rangle^{2m} \right]$ to be small whenever $\sum_{i=1}^n \widetilde{\mathbb{E}}_{\zeta_i} \left[ \mathbf{z}_i^{\otimes 2m} \right] \preceq_{SOS} \mathcal{I}$ for all pseudo-distributions $\{\zeta_i(\mathbf{z}_i)\}_{i=1}^n$, where $\mathcal{I}$ is the identity tensor of the appropriate dimension.

Therefore, we need to find $\{\gamma_i\}_{i=1}^n$ such that $\sum_{i=1}^n \widetilde{\mathbb{E}}_\xi \left[ \left\langle \widehat{\beta}_i - \gamma_i, \mathbf{v} \right\rangle^{2m} \right]$, and $\sum_{i=1}^n \widetilde{\mathbb{E}}_{\zeta_i} \left[ \langle \mathbf{z}_i, \gamma_i \rangle^{2m} \right]$ are small whenever $\sum_{i=1}^m \widetilde{\mathbb{E}}_{\zeta_i} \left[ \mathbf{z}_i^{\otimes 2m} \right] \preceq_{SOS} \mathcal{I}$ for all pseudo-distributions $\xi$ over the unit sphere, and $\{\zeta_i\}_{i=1}^n$.

---

**Algorithm 5** Basic clustering relaxation [45, Adaptation of Algorithm 1]

---

1: **Input:** $\{\widehat{\beta}_i\}_{i\in[n]}$, $\{c_i \in [0,1]\}_{i=1}^n$, $m \in \mathbb{N}$, multiplier $\lambda \geq 0$, threshold $\Gamma \geq 0$.

2: Define $\tau_i(\gamma_i, \xi, \zeta_i) := \widetilde{\mathbb{E}}_{\mathbf{v}\sim\xi}\left[\left\langle \widehat{\beta}_i - \gamma_i, \mathbf{v}\right\rangle^{2m}\right] + \lambda\widetilde{\mathbb{E}}_{\mathbf{z}_i\sim\zeta_i}\left[\langle\gamma_i, \mathbf{z}_i\rangle^{2m}\right]$.

3: Find $\{\gamma_i^*\}_{i=1}^n$ such that $\sum_{i=1}^n c_i\tau_i(\gamma_i^*, \xi, \zeta_i) \leq 2\Gamma$, for all $\xi$ over unit sphere, and

for all $\{\zeta_i\}_{i=1}^n$ satisfying $\sum_{i=1}^n \widetilde{\mathbb{E}}_{\zeta_i}\left[\mathbf{z}_i^{\otimes 2m}\right] \preceq_{\text{SOS}} \mathcal{I}$.

4: Or else, find $\xi^*$, and $\{\zeta_i^*\}_{i=1}^n$ such that $\forall \{\gamma_i\}_{i=1}^n$, $\sum_{i=1}^n c_i\tau_i(\gamma_i, \xi^*, \zeta_i^*) \geq \Gamma$.

5: **Output:** $\{\gamma_i^*\}_{i=1}^n$, or $\{\xi^*, \{\zeta_i^*\}_{i=1}^n\}$.

---

If there is no solution that makes the desired quantities small, then from duality there must exist pseudo-distributions $\xi^*$, and $\{\zeta_i^*\}_{i=1}^n$ such that the objective cannot be small for any choice of $\{\gamma_i\}_{i=1}^n$. Since elements in $\{\gamma_i\}_{i=1}^n$ are independent of each other in the objective for a fixed $\xi$ and $\{\zeta_i\}_{i=1}^n$, and the objective can made small on the good set $\mathcal{G}$, therefore we can look at $\min_\gamma \widetilde{\mathbb{E}}_{\xi^*}\left[\left\langle\widehat{\beta}_i - \gamma, \mathbf{v}\right\rangle^{2m}\right]$ or $\min_\gamma \widetilde{\mathbb{E}}_{\zeta_i^*}\left[\langle\mathbf{z}_i, \gamma\rangle^{2m}\right]$ if they are large for any $i \in [n]$. Such tasks can be removed and the process can be repeated. It is shown in [45] that the procedure after a finite number of iterations can remove all the outliers and eventually the sum of the desired quantities can be made small.

---

**Algorithm 6** Outlier Removal Algorithm [45, Adaptation of Algorithm 2]

---

1: **Input:** $\left\{\widehat{\beta}_i\right\}_{i\in[n]}$, $B \geq 0$, $m \in \mathbb{N}$, $p_{\min}$, and $\rho \geq 0$.

2: Initialize $\mathbf{c} = \mathbf{1}_n$, and set $\lambda = p_{\min}n(B/\rho)^{2m}$.

3: **while** true **do**

4:     Run Algorithm 5 with $\{\widehat{\beta}_i\}_{i\in[n]}$, $\mathbf{c}$, $\lambda$, and threshold $\Gamma = 4(nB^{2m} + \lambda\rho^{2m}/p_{\min})$ to obtain $\{\gamma_i^*\}_{i=1}^n$, or $\{\xi^*, \{\zeta_i^*\}_{i=1}^n\}$.

5:     **if** $\{\gamma_i^*\}_{i=1}^n$ are obtained **then**

6:         **Output:** $\{\gamma_i^*\}_{i=1}^n$, $\mathbf{c}$

7:     **else if** $\{\xi^*, \{\zeta_i^*\}_{i=1}^n\}$ is obtained **then**

8:         $\tau_i^* \leftarrow \min_\gamma \tau_i(\gamma, \xi^*, \zeta_i^*) \quad \forall i \in [n]$, as defined in Algorithm 5

9:         $c_i \leftarrow c_i(1 - \tau_i^*/\max_{j\in[n]}\tau_j^*) \quad \forall i \in [n]$

10:    **end if**

11: **end while**

---

Algorithm 6 uses a down-weighting way of reducing the weight of possible outlier tasks, and [45] show that the re-weighting step down-weights the outlier tasks more than the tasks in the $\ell$-th set. They also show that the returned $\{\gamma_i^*\}_{i=1}^n$ constitute a good clustering such that one of the clusters is centered close to some true mean $\mathbf{w}_j$ for some $j \in [k]$.

Algorithm 7 repeatedly uses Algorithm 6 on re-centered data to find the individual clusters. The set $S_j$ obtained in Algorithm 7 is almost entirely a subset of one of the true good sets. After obtaining $M$ centers from Algorithm 7 we can re-consolidate the sets into $k$ new sets $\{\mathcal{C}_\ell\}_{\ell=1}^k$ (by merging together all $S_j$ whose means are within distance $B\widetilde{p}_{\min}^{-1/m}/4$.), and can shown to obey the desired guarantee using [45, Theorem 5.4.].

# H   Proof of Lemma B.4 for robust clustering

We use [45, Theorem 1.2], to analyze Algorithm 7 with input $\left\{\widehat{\mathbf{U}}^\top\widehat{\beta}_i = \frac{1}{t}\sum_{j=1}^t y_{i,j}\widehat{\mathbf{U}}^\top\mathbf{x}_{i,j}\right\}_{i\in[n]}$ for $t = t_H$ and $n = n_H$.

---

**Algorithm 7** Algorithm for re-clustering $\{\gamma_i\}_{i=1}^n$ [45, Adaptation of Algorithm 3]

---

1: **Input:** $\mathcal{D}_H = \{\{(\mathbf{x}_{i,j}, y_{i,j})\}_{j \in [t_H]}\}_{i \in [n_H]}, \widehat{\mathbf{U}}, B \geq 0, m, k \in \mathbb{N}, p_{\min}$, and $\rho \geq 0$.
2: Initialize $R = \rho$, set $W = \{\mathbf{0}\}$, and $\widehat{\beta}_i \leftarrow (1/t_H) \sum_{\ell=1}^{t_H} \widehat{\mathbf{U}}^\top \mathbf{x}_{i,j} y_{i,j}$ for all $i \in [n]$
3: $\rho_{\text{final}} \leftarrow \Theta\left(B p_{\min}^{-1/m}\right), M = \lceil 4/p_{\min} \rceil$
4: **while** $R \geq \rho_{\text{final}}$ **do**
5:    $b \leftarrow 1$
6:    **for** $\mathbf{w}' \in W$ **do**
7:       Let $\{\gamma_i^{(b)}\}_{i=1}^n, \mathbf{c}^{(b)}$ be the output of Algorithm 6 with $\{\widehat{\beta}_i - \mathbf{w}'\}_{i=1}^n, B, p_{\min}, R$ as input.
8:       $b \leftarrow b + 1$
9:    **end for**
10:   Let $\{\mathcal{C}_j\}_{j=1}^M$ be the maximal covering derived from $\{\gamma_i^{(j)}\}_{i=1,j=1}^{n,M}, \{\mathbf{c}^{(j)}\}_{j=1}^M$
11:   $W \leftarrow \{\mathbf{w}_j'\}_{j=1}^M$ where $\mathbf{w}_j'$ is the mean of points in $\mathcal{C}_j \; \forall \; j \in [M]$
12:   $R \leftarrow C'\left(\sqrt{RB p_{\min}^{-1/m}} + B p_{\min}^{-1/m}\right)$
13: **end while**
14: **Output:** $W, \{\mathcal{C}_\ell\}_{\ell=1}^M$

---

---

**Algorithm 8** Estimating $\{r_\ell^2\}_{\ell=1}^k$

---

1: **Input:** $\mathcal{D}_H = \{\{(\mathbf{x}_{i,j}, y_{i,j})\}_{j \in [t_H]}\}_{i \in [n_H]}, \{\widetilde{\mathbf{w}}_\ell\}_{\ell=1}^k, \alpha \geq 0, \delta \in (0,1)$
2: **for** $\ell \in [k]$ **do**
3:    $r_{\ell,i}^2 \leftarrow t_H^{-1} \sum_{j \in [t_H]} (y_{i,j} - \mathbf{x}_{i,j}^\top \widetilde{\mathbf{w}}_\ell)^2$ for all $i \in \mathcal{C}_\ell$
4:    **if** $\alpha > 0$ **then**
5:       $\widetilde{r}_\ell^2 \leftarrow$ Univariate_Mean_Estimator$\left(\left\{r_{\ell,i}^2\right\}_{i \in \mathcal{C}_\ell}, \alpha, \delta\right)$         [ [50]]
6:    **else**
7:       $\widetilde{r}_\ell^2 \leftarrow \frac{1}{|\mathcal{C}_\ell|} \sum_{i \in \mathcal{C}_\ell} r_{\ell,i}^2$
8:    **end if**
9: **end for**
10: **Output:** $\{\widetilde{r}_\ell^2\}_{\ell=1}^k$

---

**Theorem H.1** ([45, Theorem 1.2]). *Suppose* $\left\{\widehat{\mathbf{U}}^\top \widehat{\beta}_i \in \mathbb{R}^k\right\}_{i \in [n]}$ *can be partitioned into sets* $\{\mathcal{G}_\ell\}_{\ell=1}^k \cup \mathcal{H}$, *where* $\mathcal{H}$ *is the set of outliers, of size* $\alpha n$. *Suppose* $|\mathcal{G}_\ell| = n\widetilde{p}_\ell$, *and has mean* $\mathbf{w}_\ell$, *that its* $2m$-*th moment* $M_{2m}(\mathcal{G}_\ell) \preceq_{2m} B$. *Also suppose that* $\alpha \leq \widetilde{p}_{\min}/8$. *Finally suppose the separation* $\Delta \geq C_{\text{sep}} B/\widetilde{p}_{\min}^{1/m}$ *with* $C_{\text{sep}} \geq C_0$ *for a universal constant* $C_0$. *Then Algorithm 7 runs in time* $\mathcal{O}((nk)^{\mathcal{O}(m)})$ *and outputs estimates* $\{\widetilde{\mathbf{w}}_\ell'\}_{\ell=1}^k$ *such that* $\left\|\widetilde{\mathbf{w}}_\ell' - \widehat{\mathbf{U}}^\top \mathbf{w}_\ell\right\|_2 \leq \mathcal{O}\left(B \cdot \left(\alpha/\widetilde{p}_{\min} + C_{\text{sep}}^{-2m}\right)^{1-1/2m}\right)$.

It shows how the accuracy depends on the choice of $m$ and the SOS proof of an upper bound $B$. We will show that if $B = \rho\sqrt{2mC/t}$ then the SOS proof holds, in which case the condition $\Delta \geq C_{\text{sep}} B/\widetilde{p}_{\min}^{1/m}$ in Theorem H.1 translates into

$$t \geq \frac{2mC\rho^2}{\Delta^2} \cdot \frac{C_{\text{sep}}^2}{\widetilde{p}_{\min}^{2/m}}. \tag{70}$$

Further, to get $\left\|\widetilde{\mathbf{w}}_j - \widehat{\mathbf{U}}^\top \mathbf{w}_j\right\|_2 = \mathcal{O}(\Delta)$ we need

$$t \gtrsim \frac{2mC\rho^2}{\Delta^2} \cdot \max\left\{\frac{1}{C_{\text{sep}}^{4m-2}}, \left(\frac{\alpha}{\widetilde{p}_{\min}}\right)^{2-\frac{1}{m}}\right\}. \tag{71}$$

Combining the two conditions, we finally get

$$t \gtrsim \frac{2mC\rho^2}{\Delta^2} \cdot \max\left\{\frac{C_{\text{sep}}^2}{\widetilde{p}_{\text{min}}^{2/m}}, \frac{1}{C_{\text{sep}}^{4m-2}}, \left(\frac{\alpha}{\widetilde{p}_{\text{min}}}\right)^{2-\frac{1}{m}}\right\}. \tag{72}$$

We are left to show that SOS proof exists for the choice of $B = \rho\sqrt{2mC/t}$. The following lemma, whose proof is in §H.1, gives

$$\frac{1}{|\mathcal{G}_\ell|}\sum_{i\in\mathcal{G}_\ell}\left\langle\widehat{\mathbf{U}}^\top\left(\widehat{\beta}_i - \beta_i\right), \mathbf{v}\right\rangle^{2m} \preceq_{2m} \rho^{2m}\|\mathbf{v}\|_2^{2m}(2m)^m\frac{C^m}{t^m} \leq (2m)^m\frac{C^m}{t^m} \tag{73}$$

for all $\|\mathbf{v}\|_2 \leq 1$ with probability at least $7/8$ for all $\ell \in [k]$.

**Lemma H.2** (SOS proof exists with high probability). *Given $t \geq 2m$, for $m, t \in \mathbb{N}$, there exists a constant $C > 0$ such that for any $\ell \in [k]$, and $np_\ell \geq (km)^{\Theta(m)}/\delta$, with probability at least $1 - \delta$, it holds that*

$$\frac{1}{n\widetilde{p}_\ell}\sum_{i\in\mathcal{G}_\ell}\left\langle\widehat{\mathbf{U}}^\top(\widehat{\beta}_i - \beta_i), \mathbf{v}\right\rangle^{2m} \preceq_{2m} \rho^{2m}\|\mathbf{v}\|_2^{2m}(2m)^m\frac{C^m}{t^m},$$

*for all $\mathbf{v} \in \mathbb{R}^k$.*

From [42, Proposition D.7.] we have that if $n = \Omega\left(\frac{\log k}{p_{\text{min}}}\right)$, then $\widetilde{p}_{\text{min}} \geq p_{\text{min}}/2$ with probability at least $7/8$. To further simplify the conditions, note that $\alpha < \widetilde{p}_{\text{min}}/8$, and fix $C_{\text{sep}} = \Theta(1)$, then we simply need

$$t \gtrsim \frac{m\rho^2}{p_{\text{min}}^{2/m}\Delta^2}. \tag{74}$$

Using a median of means algorithm from [49, Proposition 1] and [33, 18], by repeatedly and independently estimating $M = \Omega\left(\log\frac{1}{\delta}\right)$ number of estimates, $\left\{\left\{\widetilde{\mathbf{w}}_j'^{(\ell)}\right\}_{j=1}^k\right\}_{\ell=1}^M$ we can compute the improved estimates $\left\{\widetilde{\mathbf{w}}_j \in \mathbb{R}^d\right\}_{j=1}^k$ by applying back $\widehat{\mathbf{U}}$ that satisfy

$$\left\|\widetilde{\mathbf{w}}_j - \widehat{\mathbf{U}}\widehat{\mathbf{U}}^\top\mathbf{w}_j\right\|_2 \lesssim \Delta \quad \forall j \in [k] \tag{75}$$

With the assumption that

$$\left\|\widehat{\mathbf{U}}\widehat{\mathbf{U}}^\top\mathbf{w}_j - \mathbf{w}_j\right\|_2 \lesssim \Delta$$

and Equation (75), we have

$$\|\widetilde{\mathbf{w}}_j - \mathbf{w}_j\|_2 \leq \left\|\widetilde{\mathbf{w}}_j - \widehat{\mathbf{U}}\widehat{\mathbf{U}}^\top\mathbf{w}_j\right\|_2 + \left\|\widehat{\mathbf{U}}\widehat{\mathbf{U}}^\top\mathbf{w}_j - \mathbf{w}_j\right\|_2 \lesssim \Delta. \tag{76}$$

We next show a bound on the error in estimating $r_\ell$.

**Proposition H.3** (Estimating $r_\ell$). *If $n_H = \widetilde{\Omega}\left(\frac{\rho^4}{\Delta^4 t_H p_{\text{min}}}\right)$, we can estimate $r_\ell^2$ as $\widetilde{r}_\ell^2$ satisfying*

$$\left|\widetilde{r}_\ell^2 - r_\ell^2\right| \leq r_\ell^2\frac{\Delta^2}{50\rho^2} \tag{77}$$

*with probability at least $1 - \delta$, for all $\ell \in [k]$ using Algorithm 8, where $r_\ell^2 := \|\widetilde{\mathbf{w}}_\ell - \mathbf{w}_\ell\|_2^2 + s_\ell^2$, if $\alpha_H = \mathcal{O}\left(\frac{\Delta^2\sqrt{t_H}p_{\text{min}}}{\rho^2\log\left(\frac{\rho^2}{\Delta^2 t_H}\right)}\right).$*

*Proof.* Define $r_{\ell,i}^2 = t_H^{-1}\sum_{j=1}^{t_H}\left(y_{i,j} - \widetilde{\mathbf{w}}_\ell^\top\mathbf{x}_{i,j}\right)^2 \sim \frac{r_\ell^2}{t_H}\chi^2(t_H)$ for all $i \in \mathcal{C}_\ell, \ell \in [k]$. Since we compute $r_\ell^2$ for each cluster $\mathcal{C}_\ell$ independently, the maximum corruption in the $\ell$-th cluster is

bounded by $\alpha_H/p_\ell$. Using Corollary J.3, we can compute an estimator using Algorithm 8, that given $\widetilde{\alpha}_\ell = \alpha_H/p_\ell$ corrupted samples, returns $\widetilde{r}_\ell^2$ for all $\ell \in [k]$ satisfying

$$\left|r_\ell^2 - \widetilde{r}_\ell^2\right| = \mathcal{O}\left(r_\ell^2 \cdot \widetilde{\alpha}_\ell \cdot \max\left\{\frac{\log(1/\widetilde{\alpha}_\ell)}{t_H}, \sqrt{\frac{\log(1/\widetilde{\alpha}_\ell)}{t_H}}\right\}\right)$$

$$= \mathcal{O}\left(r_\ell^2 \cdot \frac{\alpha_H}{p_\ell} \cdot \max\left\{\frac{\log(p_\ell/\alpha_H)}{t_H}, \sqrt{\frac{\log(p_\ell/\alpha_H)}{t_H}}\right\}\right)$$

$$= \mathcal{O}\left(r_\ell^2 \cdot \frac{\alpha_H}{p_\ell} \cdot \frac{\log(p_\ell/\alpha_H)}{\sqrt{t_H}}\right)$$

when $n_H p_\ell = \widetilde{\Omega}\left(\frac{1}{\widetilde{\alpha}_\ell^2}\right) = \widetilde{\Omega}\left(\frac{p_\ell^2}{\alpha_H^2}\right)$ for all $\ell \in [k]$. Using the fact that

$$\frac{\beta}{\log\frac{1}{\beta}} \geq \frac{e}{e-1}\alpha \implies \alpha\log\frac{1}{\alpha} \leq \beta$$

for $\alpha, \beta \in (0,1)$, we have that for $\alpha_H \leq C\frac{\Delta^2\sqrt{t_H}p_{\min}}{\rho^2\log\left(\frac{\rho^2}{\Delta^2 t_H}\right)}$ for some $C > 0$,

$$\left|r_\ell^2 - \widetilde{r}_\ell^2\right| = r_\ell^2 \cdot \frac{\Delta^2}{50\rho^2} \tag{78}$$

with probability at-least $1 - \delta$, when $n_H = \widetilde{\Omega}\left(\frac{\rho^4}{\Delta^4 t_H p_{\min}}\right)$. $\qquad\square$

## H.1 Proof of Lemma H.2 for Sum-of-Squares proof $y\mathbf{x}$

We combine the following Proposition H.4 and Lemma H.8 to yield the desired SOS proof.

**Proposition H.4** (see the proof of Lemma 4.1 in [35]). *Let* $Z_i = \frac{1}{t}\sum_{j=1}^t y_{i,j}\mathbf{x}_{i,j} - \beta_i$ *for* $i \in \mathcal{G}_\ell$. *If* $np_\ell \geq (km)^{\Theta(m)}/\delta \; \forall\, \ell \in [k]$, *then with probability at least* $1 - \delta$,

$$\sum_{i=1}^n \langle Z_i, \mathbf{v}\rangle^{2m} - \mathbb{E}\left[\langle Z_i, \mathbf{v}\rangle^{2m}\right] \preceq_{2m} \frac{1}{4}\|\mathbf{v}\|_2^{2m}\frac{C^m}{t^m}$$

*Proof.* We show in Lemma H.8 that the distribution of $\frac{\sqrt{t}}{\sqrt{C}}Z_i$ is $2m$-explicitly bounded with variance proxy 1. In the proof of Lemma 4.1 in [35], it is shown that for a $2m$-explicitly bounded distribution, given $n \geq (km)^{\Theta(m)}/\delta$ samples, with probability at least $1 - \delta$ (Fact 7.6 in [35]),

$$\frac{t^m}{C^m}\left(\sum_{i=1}^n \langle Z_i, \mathbf{v}\rangle^{2m} - \mathbb{E}\left[\langle Z_i, \mathbf{v}\rangle^{2m}\right]\right) \preceq_{2m} \frac{1}{4}\|\mathbf{v}\|_2^{2m},$$

which implies the propositions. $\qquad\square$

**Fact H.5** (Claim A.9. in [26]). *Let* $\sigma_y^2 = \|\beta\|_2^2 + \sigma^2$. *For any* $v \in \mathbb{R}^d$, *we have that*

$$(\mathbf{v}^\top\mathbf{x})y = \left(\frac{\mathbf{v}^\top\beta + \|\mathbf{v}\|_2\sigma_y}{2}\right)Z_1^2 + \left(\frac{\mathbf{v}^\top\beta - \|\mathbf{v}\|_2\sigma_y}{2}\right)Z_2^2,$$

*where* $Z_1, Z_2 \sim \mathcal{N}(0,1)$ *and* $Z_1, Z_2$ *are independent.*

We say that polynomial $p(v) \succeq q(v)$ if $p(x) - q(x)$ can be written as a sum of squares of polynomials. We write this as $p(v) \succeq_{2m} q(v)$ if we want to emphasize that the proof only involves polynomials of degree at most $2m$.

**Fact H.6** (Basic facts about SOS proofs).

- $\left(\mathbf{v}^\top\beta\right)^2 \preceq_2 \|\mathbf{v}\|_2^2\|\beta\|_2^2$ *(Cauchy-Schwarz),*

- $p_1 \succeq_{m_1} p_2 \succeq_{m_1} 0$ , *and* $q_1 \succeq_{m_2} q_2 \succeq_{m_2} 0 \implies p_1 p_2 \succeq_{m_1+m_2} q_1 q_2$.

**Definition H.7.** *A distribution $\mathcal{D}$ over $\mathbb{R}^d$ with mean $\mu \in \mathbb{R}^d$ is 2m-explicitly bounded for $m \in \mathbb{N}$, if $\forall\, i \le 2m$, we have*

$$\mathop{\mathbb{E}}_{X \sim \mathcal{D}}\left[\langle X - \mu, \mathbf{v}\rangle^i\right] \preceq_i (\sigma i)^{i/2}\|\mathbf{v}\|_2^i$$

*for variance proxy $\sigma^2 \in \mathbb{R}_+$.*

**Lemma H.8.** *Given that $t \ge 2m$, for $m, t \in \mathbb{N}$, there exists a constant $C > 0$ such that*

$$\mathbb{E}\left[\left\langle \frac{1}{t}\sum_{i=1}^{t} y_i \mathbf{x}_i - \beta, \mathbf{v}\right\rangle^{2m}\right] \preceq_{2m} \rho^{2m}\|\mathbf{v}\|_2^{2m}(2m)^m \frac{C^m}{t^m}.$$

*Proof.*

$$\mathbb{E}\left[\left\langle \frac{1}{t}\sum_{i=1}^{t} y_i \mathbf{x}_i - \beta, \mathbf{v}\right\rangle^{2m}\right]$$
$$= \mathbb{E}\left[\left(\left(\frac{\mathbf{v}^\top \beta + \|\mathbf{v}\|_2 \sigma_y}{2}\right)(S_1 - 1) + \left(\frac{\mathbf{v}^\top \beta - \|\mathbf{v}\|_2 \sigma_y}{2}\right)(S_2 - 1)\right)^{2m}\right]$$
$$= \sum_{i=0}^{2m}\binom{2m}{i}\left(\frac{\mathbf{v}^\top \beta + \|\mathbf{v}\|_2 \sigma_y}{2}\right)^i \left(\frac{\mathbf{v}^\top \beta - \|\mathbf{v}\|_2 \sigma_y}{2}\right)^{2m-i} M_i M_{2m-i}$$
$$= \sum_{i=0}^{m}\binom{2m}{i}\left(\left(\frac{\mathbf{v}^\top \beta + \|\mathbf{v}\|_2 \sigma_y}{2}\right)^{2m-2i} + \left(\frac{\mathbf{v}^\top \beta - \|\mathbf{v}\|_2 \sigma_y}{2}\right)^{2m-2i}\right)\left(\frac{(\mathbf{v}^\top \beta)^2 - \|\mathbf{v}\|_2^2 \sigma_y^2}{4}\right)^i M_i M_{2m-i}$$

where $S_1, S_2 \sim \frac{\chi^2(t)}{t}$, and $M_i = \mathbb{E}_{Z_j \sim \mathcal{N}(0,1)}\left[\left(\frac{1}{t}\sum_{j=1}^{t}(Z_j^2 - 1)\right)^i\right]$.

First, notice that

$$\left(\frac{\mathbf{v}^\top \beta + \|\mathbf{v}\|_2 \sigma_y}{2}\right)^{2m-2i} + \left(\frac{\mathbf{v}^\top \beta - \|\mathbf{v}\|_2 \sigma_y}{2}\right)^{2m-2i}$$
$$= 2^{2i-2m} \cdot 2 \cdot \sum_{j=0}^{m-i}\binom{2m-2i}{2j}(\mathbf{v}^\top \beta)^{2j}(\|\mathbf{v}\|_2 \sigma_y)^{2m-2i-2j}$$
$$\preceq_{2m-2i} 2^{2i-2m} 2^{2m-2i}\|\mathbf{v}\|_2^{2m-2i}\rho^{2m-2i}$$
$$= \|\mathbf{v}\|_2^{2m-2i}\rho^{2m-2i}. \tag{79}$$

The SOS order hold by the fact that $\left(\mathbf{v}^\top \beta\right)^2 \preceq_2 \|\beta\|_2^2\|\mathbf{v}\|_2^2 \preceq_2 \sigma_y^2\|\mathbf{v}\|_2^2$ and Fact H.6. Secondly, notice that

$$\left(\frac{(\mathbf{v}^\top \beta)^2 - \|\mathbf{v}\|_2^2 \sigma_y^2}{4}\right)^i = 2^{-2i}\sum_{j=0}^{i}\binom{i}{j}(-1)^{i-j}(\mathbf{v}^\top \beta)^{2j}(\|\mathbf{v}\|_2 \sigma_y)^{2i-2j}$$
$$\preceq_{2i} 2^{-i}\|\mathbf{v}\|_2^{2i}\rho^{2i}$$
$$\preceq_{2i} \|\mathbf{v}\|_2^{2i}\rho^{2i} \tag{80}$$

Combining Equation (79) and Equation (80) yields

$$\mathbb{E}\left[\left\langle \frac{1}{t}\sum_{i=1}^{t} y_i \mathbf{x}_i - \beta, \mathbf{v}\right\rangle^{2m}\right] \preceq_{2m} \rho^{2m}\|\mathbf{v}\|_2^{2m}\sum_{i=0}^{m}\binom{2m}{i}M_i M_{2m-i}$$

**Fact H.9.** *Given that $t \ge 2m$, it holds that*

$$M_i \le 2e^{2i}i^{i/2}/t^{i/2} \ ,$$

*Proof.* By Bernstein inequality, $\sum_{j=1}^{t}(Z_j^2 - 1)$ is a combination of sub-Gaussian with norm $\Theta(\sqrt{t})$ and sub-exponential with norm $\Theta(1)$ with disjoint supports. Hence

$$\frac{1}{t^i}\mathbb{E}\left[\left(\sum_{j=1}^{t}(Z_j^2 - 1)\right)^i\right] \lesssim \frac{e^i(ti)^{i/2} + e^{2i}i^i}{t^i}$$

$$= e^i\frac{i^{i/2}}{t^i}(t^{i/2} + e^i i^{i/2}),$$

$$= e^i\frac{i^{i/2}}{t^{i/2}}\left(1 + e^i\frac{i^{i/2}}{t^{i/2}}\right)$$

$$\leq 2e^{2i}\frac{i^{i/2}}{t^{i/2}}. \qquad \square$$

Applying Fact H.9 gives

$$\mathbb{E}\left[\left\langle\frac{1}{t}\sum_{i=1}^{t}y_i\mathbf{x}_i - \beta, \mathbf{v}\right\rangle^{2m}\right] \preceq_{2m} \rho^{2m}\|\mathbf{v}\|_2^{2m}(2m)^m\frac{C^m}{t^m}$$

for some $0 < C < e^6$. This concludes the proof. $\qquad \square$

## H.2 The distribution of $y\mathbf{x}$ is $\Omega(d)$-Poincaré

We show that even for the simplest parameter setting where the regression vector $\beta = 0$, and the noise has variance 1, the distribution of $y\mathbf{x}$ is Poincaré with parameter $\Omega(d)$, and thus applying the result on Poincaré distribution from [45] naively would only yield trivial guarantee.

**Remark H.10.** *Suppose* $\mathbf{x} \sim \mathcal{N}(0, \mathbf{I})$, $y \sim \mathcal{N}(0, 1)$, *and function* $f(\mathbf{z}) = \|\mathbf{z}\|_2^2$. *Then* $\mathrm{Var}[f(\mathbf{z})] = (2d + 6)\mathbb{E}[\|\nabla f(\mathbf{z})\|_2^2]$.

*Proof.*

$$\mathrm{Var}[f(\mathbf{z})] = \mathbb{E}[y^4\|\mathbf{x}\|_2^4] - \mathbb{E}[y^2\|\mathbf{x}\|_2^2]^2$$

$$= 3d(d + 2) - d^2$$

$$= 2d^2 + 6d.$$

Since $\nabla f(\mathbf{z}) = 2\mathbf{z}$, we have

$$\mathbb{E}[\|\nabla f(\mathbf{z})\|_2^2] = \mathbb{E}[4y^2\|\mathbf{x}\|_2^2] = 4d.$$

Hence

$$\mathrm{Var}[f(\mathbf{z})] = \frac{1}{2}(d + 3)\mathbb{E}[\|\nabla f(\mathbf{z})\|_2^2]. \qquad \square$$

**Remark H.11.** *The choice of $m$ can be made from $t$ appropriately by considering the analytical inverse map. For any $c > 0$, if $y = \frac{t\Delta^2}{2c\rho^2\log(1/p_{\min})}$, and $x = \frac{2\log(1/p_{\min})}{m}$, then it is clear that the condition $t \geq c \cdot \frac{m}{p_{\min}^{2/m}} \cdot \frac{\rho^2}{\Delta^2}$, can be written as $y \geq e^x/x$, or $-1/y \geq (-x)e^{-x}$. Therefore, this condition is satisfied when*

$$\frac{2\log(1/p_{\min})}{-W_{-1}\left(-\frac{2c\rho^2\log(1/p_{\min})}{t\Delta^2}\right)} \leq m \leq \frac{2\log(1/p_{\min})}{-W_0\left(-\frac{2c\rho^2\log(1/p_{\min})}{t\Delta^2}\right)} \tag{81}$$

*if $t \geq 2ec \cdot \frac{\rho^2}{\Delta^2}\log\frac{1}{p_{\min}}$, where $W_0$ and $W_{-1}$ are the Lambert W functions.*

**Algorithm 9** Classification and robust estimation
___
1: **Input:** data $\mathcal{D}_{L2} = \{(\mathbf{x}_{i,j}, y_{i,j})\}_{i \in [n_{L2}], j \in [t_{L2}]}$, $\{\mathcal{C}_\ell, \widetilde{\mathbf{w}}_\ell, \widetilde{r}_\ell^2\}_{\ell \in [k]}$, $\alpha > 0, \delta \in (0,1)$.
2: **compute** for all $i \in [n_{L2}]$

$$h_i \leftarrow \arg\min_{\ell \in [k]} \frac{1}{2\widetilde{r}_\ell^2} \sum_{j \in [t_{L2}]} \left(y_{i,j} - \mathbf{x}_{i,j}^\top \widetilde{\mathbf{w}}_\ell\right)^2 + t_{L2} \log \widetilde{r}_\ell$$

3:      $\mathcal{C}_{h_i} \leftarrow \mathcal{C}_{h_i} \cup \left\{(\mathbf{x}_{i,j}, y_{i,j})\right\}_{j=1}^{t_{L2}}$
4: **compute** for all $\ell \in [k]$,
5:      $\widehat{\mathbf{w}}_\ell \leftarrow \text{Robust\_Least\_Squares}(\mathcal{C}_\ell)$                                 [ [26, Algorithm 2]]
6:      $r_{\ell,i}^2 \leftarrow t_{L2}^{-1} \sum_{j \in [t_{L2}]} \left(y_{i,j} - \mathbf{x}_{i,j}^\top \widehat{\mathbf{w}}_\ell\right)^2$ for all $i \in \mathcal{C}_\ell$
7:      $\widehat{s}_\ell^2 \leftarrow \text{Univariate\_Mean\_Estimator}\left(\left\{r_{\ell,i}^2\right\}_{i \in \mathcal{C}_\ell}, \alpha, \delta\right)$               [ [50]]
8:      $\widehat{p}_\ell \leftarrow |\mathcal{C}_\ell| / n_{L2}$
9: **Output:** $\left\{\mathcal{C}_\ell, \widehat{\mathbf{w}}_\ell, \widehat{s}_\ell^2, \widehat{p}_\ell\right\}_{\ell=1}^k$
___

# I    Proof of Lemma B.6, Classification and robust estimation

**Lemma I.1** (Lemma A.15 in [42]). *Given estimated parameters satisfying* $\|\widetilde{\mathbf{w}}_i - \mathbf{w}_i\|_2 \leq \Delta/10$, $(1 - \Delta^2/50)\widetilde{r}_i^2 \leq s_i^2 + \|\widetilde{\mathbf{w}}_i - \mathbf{w}_i\|_2^2 \leq (1 + \Delta^2/50)\widetilde{r}_i^2$ *for all* $i \in [k]$, *and a new task with* $t_{\text{out}} \geq \Theta\left(\log(k/\delta)/\Delta^4\right)$ *samples whose true regression vector is* $\beta = \mathbf{w}_h$, *Algorithm 9 predicts* $h$ *correctly with probability* $1 - \delta$.

Since the set $G$ contains $n_{L2}$ i.i.d. samples from our data generation model, by the assumption that $n_{L2} = \widetilde{\Omega}\left(\frac{d}{p_{\min} \epsilon^2 t_{L2}}\right) = \Omega\left(\frac{\log(k/\delta)}{p_{\min}}\right)$ and from Proposition J.6, it holds that the number of tasks such that $\beta = \mathbf{w}_i$ is $n_{L2} \widehat{p}_i \geq \frac{1}{2} n_{L2} p_i$ with probability at least $1 - \delta$. Hence, with this probability, there exists at least $n_{L2} p_i/2$ i.i.d. examples in $G$ for estimating $\mathbf{w}_i$ and $s_i^2$. Lemma B.6 guarantee that our algorithm correctly classified all the tasks in $G$, which implies that there are at least $(p_i/2 - \alpha_{L2})n_{L2}$ uncorrupted tasks, and at most $\alpha_{L2} n_{L2}$ corrupted tasks, and hence the corruption level is at most $\frac{\alpha_{L2} n}{(p_i/2 - \alpha_{L2})n} \leq \frac{4\alpha_{L2}}{p_i}$ since $\alpha_{L2} \leq p_{\min}/4$. We can apply robust linear regression algorithm to each cluster separately, and the error of the algorithm is bounded by the following lemma.

**Lemma I.2** (Robust Linear Regression, Lemma 1.3 in [26]). *Let* $S'$ *be an* $\alpha$-*corrupted set of labeled samples of size* $\Omega((d/\alpha^2) \operatorname{poly} \log(d/(\alpha\tau)))$. *There exists an efficient algorithm that on input* $S'$ *and* $\alpha > 0$, *returns a candidate vector* $\widehat{\beta}$ *such that with probability at least* $1 - \tau$ *it holds* $\left\|\widehat{\beta} - \beta\right\|_2 = \mathcal{O}(\sigma\alpha \log(1/\alpha))$.

For a single regressor $i \in [k]$, Lemma I.2 implies that for any $\epsilon \geq \Omega\left(\frac{\alpha_{L2}}{p_i} \log(p_i/\alpha_{L2})\right)$, given that $n_{L2} t_{L2} p_i \geq \widetilde{\Omega}(d/\epsilon^2)$, it holds that with probability $1 - \delta$, our estimation satisfies

$$\|\widehat{\mathbf{w}}_i - \mathbf{w}_i\|_2 \leq \epsilon s_i.$$

Using Corollary J.3, we have that our robust variance estimator guarantees that

$$\left|\widehat{s}_i^2 - \left(s_i^2 + \|\widehat{\mathbf{w}}_i - \mathbf{w}_i\|_2^2\right)\right| \leq \frac{\epsilon}{\sqrt{t_{L2}}}\left(s_i^2 + \|\widehat{\mathbf{w}}_i - \mathbf{w}_i\|_2^2\right)$$

$$\leq \frac{\epsilon}{\sqrt{t_{L2}}} s_i^2,$$

with probability $1 - \delta$. Taking a union bound over $k$ regressors, we have that for any $\epsilon \geq \Omega\left(\frac{\alpha}{p_{\min}} \log(p_{\min}/\alpha)\right)$, given that

$$n_{L2} \geq \widetilde{\Omega}\left(\frac{d}{t_{L2} p_{\min} \epsilon^2}\right),$$

for all $i \in [k]$, our estimators satisfy

$$\|\widehat{\mathbf{w}}_i - \mathbf{w}_i\|_2^2 \leq \epsilon s_i, \quad \text{and}$$

$$\left|\widehat{s}_i^2 - s_i^2\right| \leq \frac{\epsilon}{\sqrt{t_{L2}}} s_i^2. \text{ (Applying Corollary J.3)}$$

By Proposition J.6, it holds that

$$|\widehat{p}_i - p_i| \leq \sqrt{\frac{3\log(k/\delta)}{n_{L2}}} p_i + \alpha_{L2}$$

$$\leq \min\left\{p_{\min}/10, \epsilon p_i \sqrt{t_{L2}/d}\right\} + \alpha_{L2}.$$

The condition on $\epsilon$ can be converted in to a condition on $\alpha_H$ by the fact that $\frac{\beta}{\log\frac{1}{\beta}} \geq \frac{e}{e-1}\alpha \implies \alpha\log\frac{1}{\alpha} \leq \beta$ for $\alpha, \beta \in (0,1)$. This completes the proof.

# J  Proofs of technical lemmas and remarks

## J.1  Auxiliary Lemmas

**Fact J.1** ($\epsilon$-tail bound for distributions with bounded second moment). *Suppose random variable $z$ with probability density function $p(\cdot)$, satisfies $\mathbb{E}[z^2] \leq \sigma^2$, then for any event $\mathcal{E}$ with $\mathbb{P}[\mathcal{E}] \geq 1 - \epsilon$, it holds that*

$$|\mathbb{E}[z] - \mathbb{P}[\mathcal{E}]\,\mathbb{E}[z|\mathcal{E}]| \leq \sqrt{\epsilon}\sigma.$$

*Proof.* Notice that

$$
\begin{aligned}
&|\mathbb{E}[z] - \mathbb{P}[\mathcal{E}]\,\mathbb{E}[z|\mathcal{E}]|\\
=&\left|\mathbb{P}[\bar{\mathcal{E}}]\,\mathbb{E}[z|\bar{\mathcal{E}}]\right|\\
=&\left|\int_{-\infty}^{\infty} \mathbb{1}\left\{z \in \bar{\mathcal{E}}\right\} zp(z)dz\right|\\
\leq&\sqrt{\int_{-\infty}^{\infty} \mathbb{1}\left\{z \in \bar{\mathcal{E}}\right\} p(z)dz \cdot \int_{-\infty}^{\infty} z^2 p(z)dz} \quad \text{(Using Cauchy–Schwarz)}\\
\leq&\sqrt{\epsilon}\sigma. \hspace{10cm}\square
\end{aligned}
$$

**Proposition J.2** (Matrix Bernstein inequality, Theorem 1.6.2 in [70]). *Let $\mathbf{S}_1, \ldots, \mathbf{S}_n$ be independent, centered random matrices with common dimension $d_1 \times d_2$, and assume that each one is uniformly bounded $\mathbb{E}[\mathbf{S}_k] = \mathbf{0}$ and $\|\mathbf{S}_k\|_2 \leq L \; \forall\, k = 1, \ldots, n$.*

*Introduce the sum*

$$\mathbf{Z} := \sum_{k=1}^{n} \mathbf{S}_k$$

*and let $v(\mathbf{Z})$ denote the matrix variance statistic of the sum:*

$$v(\mathbf{Z}) := \max\left\{\left\|\mathbb{E}\left[\mathbf{Z}\mathbf{Z}^\top\right]\right\|_2, \left\|\mathbb{E}\left[\mathbf{Z}^\top\mathbf{Z}\right]\right\|_2\right\}$$

*Then*

$$\mathbb{P}[\|\mathbf{Z}\|_2 \geq t] \leq (d_1 + d_2)\exp\left\{\frac{-t^2/2}{v(\mathbf{Z}) + Lt/3}\right\}$$

*for all $t \geq 0$.*

**Corollary J.3** (Robust mean estimation of chi-square distribution). *Let $G = \{x_i\}_{i=1}^n$ where each $x_i$ is drawn independently from a scaled chi-square distribution $\frac{\sigma^2}{t}\chi^2(t)$ for $t \in \mathbb{N}$. Suppose $S = (G \setminus L) \cup E$ where $|L| \leq \epsilon n$ and $|E| \leq \epsilon n$. Assuming that $n = \widetilde{\Omega}(\frac{1}{\epsilon^2})$, the trimmed mean estimator define in [50] takes $S$ as input and output an estimate $\widehat{\sigma}^2$, with probability $1 - \delta$, that satisfies*

$$\left|\widehat{\sigma}^2 - \sigma^2\right| = \mathcal{O}\left(\sigma^2 \epsilon \max\left\{\frac{\log(1/\epsilon)}{t}, \sqrt{\frac{\log(1/\epsilon)}{t}}\right\}\right).$$

*Proof.* We show the corollary by applying Theorem 1 in [50] to the chi-square setting. First we bound the $\mathcal{E}(4\epsilon, X)$ term in [50, Theorem 1]. For a zero mean random variable $X$, define quantile

$$Q_p(X) = \sup\{M \in \mathbb{R} : \mathbb{P}[X \geq M] \geq 1 - p\},$$

and $\mathcal{E}(\epsilon, X)$ is defined as

$$\mathcal{E}(\epsilon, X) := \max\{\mathbb{E}[|X|\mathbb{1}\{X \leq Q_{\epsilon/2}\}], \mathbb{E}[|X|\mathbb{1}\{X \geq Q_{1-\epsilon/2}\}]\}.$$

where we denote $Q_p(X)$ by $Q_p$ simply. Let $X = \frac{x_i}{\sigma^2} - 1$. Note that under chi-square distribution for $x_i$, for small $\epsilon$, we can assume $Q_{\epsilon/2} \leq 0$ and $Q_{1-\epsilon/2} \geq 0$. By Bernstein's inequality, we have that for any $u \geq 0$,

$$\mathbb{P}\left[\left|\frac{x_i}{\sigma^2} - 1\right| \geq u\right] \leq 2\exp\left(-ct\min\{u^2, u\}\right).$$

If $\log(2/\epsilon) \leq ct$, let $u_\epsilon = \sqrt{\log(2/\epsilon)/ct}$ such that $2\exp\left(-ct\min\{u_\epsilon^2, u_\epsilon\}\right) = \epsilon$. We have

$$\mathbb{E}[|X|\mathbb{1}\{X \geq Q_{1-\epsilon/2}\}] = \int_{Q_{1-\epsilon}}^\infty \mathbb{P}[X \geq u]du$$

$$= \int_{Q_{1-\epsilon}}^{u_\epsilon} \mathbb{P}[X \geq u]du + \int_{u_\epsilon}^\infty \mathbb{P}[X \geq z]dz$$

$$\leq \epsilon \cdot u_\epsilon + \frac{\epsilon}{ct}\sqrt{\log\frac{2}{\epsilon}}$$

$$= \mathcal{O}\left(\frac{\epsilon\sqrt{\log(1/\epsilon)}}{\sqrt{t}}\right).$$

Otherwise, let $u_\epsilon = \log(2/\epsilon)/ct$ such that $2\exp(-ct\min(u_\epsilon^2, u_\epsilon)) = \epsilon$. Then,

$$\mathbb{E}[|X|\mathbb{1}\{X \geq Q_{1-\epsilon}\}] = \int_{Q_{1-\epsilon}}^\infty \mathbb{P}[X \geq u]du$$

$$= \int_{Q_{1-\epsilon}}^{u_\epsilon} \mathbb{P}[X \geq u]du + \int_{u_\epsilon}^\infty \mathbb{P}[X \geq u]du$$

$$\leq \epsilon \cdot u_\epsilon + \frac{2}{ct}\exp(-ctu_\epsilon)$$

$$= \mathcal{O}\left(\frac{\epsilon\log(1/\epsilon)}{t}\right).$$

Combing the two term we get

$$Q_{1-\epsilon/2}(X) = \mathcal{O}\left(\epsilon\max\left\{\frac{\log(1/\epsilon)}{t}, \sqrt{\frac{\log(1/\epsilon)}{t}}\right\}\right),$$

which implies

$$\mathcal{E}(4\epsilon, X) = Q_{1-\epsilon/2}(X) = \mathcal{O}\left(\epsilon\max\left\{\frac{\log(1/\epsilon)}{t}, \sqrt{\frac{\log(1/\epsilon)}{t}}\right\}\right).$$

The variance of $X$ is bounded as

$$\sigma_X^2 = \mathrm{Var}\left[\frac{x_i}{\sigma^2} - 1\right] = \mathcal{O}\left(\frac{1}{t}\right).$$

Hence, with the assumption that $n = \widetilde{\Omega}(1/\epsilon^2)$, Theorem 1 in [50] guarantee to estimate $\sigma^2$ with error

$$\frac{|\widehat{\sigma}^2 - \sigma^2|}{\sigma^2} = \mathcal{O}\left(\epsilon\max\left\{\frac{\log(1/\epsilon)}{t}, \sqrt{\frac{\log(1/\epsilon)}{t}}\right\}\right). \qquad \square$$

**Proposition J.4** (Trimmed mean estimator for distributions with bounded variances (see, e.g. Proposition 2.2 in [65])). *Suppose a multi-set $S = \{x_i\}_{i=1}^n$, $0 < \epsilon \leq 1/8$, satisfies $S = (G \setminus L) \cup E$, $|E| \leq \epsilon|G|$, $|L| \leq \epsilon|G|$, and set $G$ satisfies*

$$\left| \frac{1}{|G|} \sum_{x_i \in G} x_i - \mu \right| \leq \sqrt{\epsilon}$$

$$\frac{1}{|G|} \sum_{x_i \in G} (x_i - \mu)^2 \leq 1$$

*Let $R$ be the set containing the lower and upper $2\epsilon$ quantiles from $S$, then set $S' = S \setminus R = (G \setminus L') \cup E'$ satisfies*

$$\left| \frac{1}{|S'|} \sum_{x_i \in S'} x_i - \mu \right| \leq 18\sqrt{\epsilon},$$

*Proof.* The result is well-known. We provide a proof here for completeness. First note that all the datapoints in $E$ that exceed the $\epsilon$-quantile of $G$ must lie in $R$. By Chebyshev's inequality, $\frac{1}{|G|} \sum_{x_i \in G} \mathbb{1}\left\{ |x_i - \mu| \geq \sqrt{\frac{1}{\epsilon}} + \sqrt{\epsilon} \right\} \leq \epsilon$. Therefore $|x_i - \mu| \leq \sqrt{\frac{1}{\epsilon}} + \sqrt{\epsilon}$ for all $x_i \in E'$. Second, since $|L'| \leq 5\epsilon$, by Fact J.1, the mean of $G \setminus L'$ lies within $\sqrt{\frac{10\epsilon}{1-5\epsilon}}$ of $\mu$. Finally, the difference between $\mu$ and the mean of $(G \setminus L') \cup E'$ is bounded by

$$\left| \frac{1}{|S'|} \sum_{x_i \in S'} x_i - \mu \right| \leq \frac{1}{|S'|} \left( |E'|(\sqrt{\frac{1}{\epsilon}} + \sqrt{\epsilon}) + |G \setminus L'|\sqrt{\frac{10\epsilon}{1-5\epsilon}} \right)$$

$$\leq 18\sqrt{\epsilon} \text{ (Assuming that } \epsilon \leq 1/8 \text{)}. \qquad \square$$

**Proposition J.5** ($\ell_1$ deviation bound of multinomial distributions [74]). *Let $\mathbf{p} = \{p_1, \ldots, p_k\}$ be a vector of probabilities (i.e. $p_i \geq 0$ for all $i \in [k]$ and $\sum_{i=1}^k p_i = 1$). Let $\mathbf{x} \sim \text{multinomial}(n, \mathbf{p})$ follow a multinomial distribution with $n$ trials and probability $\mathbf{p}$. Then given $n \geq 2k \log(2/\delta)/\alpha^2$ with probability $1 - \delta$,*

$$\left\| \frac{1}{n}\mathbf{x} - \mathbf{p} \right\|_1 \leq \alpha,$$

**Proposition J.6** ($\ell_\infty$ deviation bound of multinomial distributions, Proposition D.7 in [42]). *Let $\mathbf{p} = \{p_1, \ldots, p_k\}$ be a vector of probabilities (i.e. $p_i \geq 0$ for all $i \in [k]$ and $\sum_{i=1}^k p_i = 1$). Let $\mathbf{x} \sim \text{multinomial}(n, \mathbf{p})$ follow a multinomial distribution with $n$ trials and probability $\mathbf{p}$. Then with probability $1 - \delta$, for all $i \in [k]$,*

$$\left| \frac{1}{n}x_i - p_i \right| \leq \sqrt{\frac{3\log(k/\delta)}{n} p_i},$$

*which implies*

$$\left\| \frac{1}{n}\mathbf{x} - \mathbf{p} \right\|_\infty \leq \sqrt{\frac{3\log(k/\delta)}{n}}.$$

*for all $i \in [k]$.*

**Fact J.7** (Gaussian 4-th moment conditions). *Let $\mathbf{v}$, $\mathbf{u}$, and $\mathbf{w}$ denote three fixed vectors, we have*

1. $\mathbb{E}_{\mathbf{x} \sim \mathcal{N}(\mathbf{0}, \mathbf{I})}\left[ (\mathbf{v}^\top \mathbf{x})^2 (\mathbf{u}^\top \mathbf{x})^2 \right] = \|\mathbf{u}\|_2^2 \cdot \|\mathbf{v}\|_2^2 + 2\langle \mathbf{u}, \mathbf{v} \rangle^2$

2. $\mathbb{E}_{\mathbf{x} \sim \mathcal{N}(\mathbf{0}, \mathbf{I})}\left[ (\mathbf{v}^\top \mathbf{x})^3 (\mathbf{u}^\top \mathbf{x}) \right] = 3\|\mathbf{v}\|_2^2 \cdot \langle \mathbf{v}, \mathbf{u} \rangle$

3. $\mathbb{E}_{\mathbf{x} \sim \mathcal{N}(\mathbf{0}, \mathbf{I})}\left[ (\mathbf{u}^\top \mathbf{x})(\mathbf{v}^\top \mathbf{x})(\mathbf{w}^\top \mathbf{x})^2 \right] = \|\mathbf{w}\|_2^2 \langle \mathbf{u}, \mathbf{v} \rangle + 2\langle \mathbf{u}, \mathbf{w} \rangle \langle \mathbf{v}, \mathbf{w} \rangle$

## K    Outlier robust principal component analysis

We provide comparisons of Algorithm 2 to state-of-the-art baselines, in both theory and numerical experiments.

### K.1    Theoretical comparisons

**Comparisons with [75].** Outlier-Robust Principal Component Analysis (ORPCA) [75, 28, 76] studies a similar problem under a Gaussian model. For comparison, we can modify the best known ORPCA estimator from [75] to our setting in Proposition 2.6, to get a semi-orthogonal $\widehat{\mathbf{U}}$ achieving

$$\left\| \mathbf{\Sigma} - \widehat{\mathbf{U}}\widehat{\mathbf{U}}^\top \mathbf{\Sigma} \widehat{\mathbf{U}}\widehat{\mathbf{U}}^\top \right\|_* = \left\| \mathbf{\Sigma} - \mathcal{P}_k(\mathbf{\Sigma}) \right\|_* + \mathcal{O}\left( \alpha^{1/2} \| \mathcal{P}_k(\mathbf{\Sigma}) \|_* + \nu k \alpha^{1/4} \right) . \tag{82}$$

We significantly improve in the dominant third term (see Eq. (5)).

This is due to our double filtering in Algorithm 3, which guarantees that we remove more corrupted examples than uncorrupted examples. On the other hand, ORPCA estimator in [75] uses a single filter that removes a single example per iteration. The removed example is a corrupted one with probability at least $\gamma \in (0, 1)$. This filter runs for $\Theta(n\alpha/\gamma)$ iterations to ensure that the corrupted examples are sufficiently removed (remaining corrupted examples only contribute to $\gamma$-fraction of the second moment). However, this filter also remove roughly a $\Theta(\alpha/\gamma)$-fraction of good examples. Under our setting, this causes a $\Theta(\alpha/\gamma)$ multiplicative error and $\Theta(k\sqrt{\alpha/\gamma})$ additive deviation by Lemma D.5, part 3. This achieves

$$\mathrm{Tr}\left[\mathbf{U}^\top \mathbf{\Sigma} \mathbf{U}\right] \ \geq \ (1 - \Theta(\gamma + \alpha/\gamma)) \, \mathrm{Tr}[\mathcal{P}_k(\mathbf{\Sigma})] - \mathcal{O}\left(\nu k \sqrt{\alpha/\gamma}\right) ,$$

for any $\gamma > 0$. Setting $\gamma = \sqrt{\alpha}$ gives Eq. (82), following a similar line of analysis as in §D.1.

**Comparisons with filters based on the second moment of $z_i$'s.** Popular recent results on robust estimation are based on filters that rely on the second moment [24, 66]. One might wonder if it is possible to apply these filters to $z_i$'s to remove the corrupted samples. Such approaches fail when $n = \mathcal{O}(d)$, even when $\mathbf{x}_i$ are standard Gaussian; this is immediate from the fact that the empirical second moment of $z_i$'s does not concentrate until we have $n = \mathcal{O}(d^2)$ uncorrupted samples.

**Comparisons with robust mean estimation [17].** Another approach is to use the existing off-the-shelf robust mean estimators, such as [17], directly to estimate the second moment matrix $\mathbf{\Sigma} \in \mathbb{R}^{d \times d}$, as remarked in [64]. However, the application of [17, Theorem 1.3] does not take advantage of the spiked low-rank structure of $\mathbf{\Sigma}$. This results in a larger sample complexity scaling as $n = \widetilde{\Omega}(d^2/\alpha)$ to achieve the following bound similar to Eq. (5).

$$\left\| \mathbf{\Sigma} - \widehat{\mathbf{U}}\widehat{\mathbf{U}}^\top \mathbf{\Sigma} \widehat{\mathbf{U}}\widehat{\mathbf{U}}^\top \right\|_* = \left\| \mathbf{\Sigma} - \mathcal{P}_k(\mathbf{\Sigma}) \right\|_* + \mathcal{O}(\nu \, k \sqrt{\alpha} \, \|\mathbf{\Sigma}\|_2) \tag{83}$$

**Remark K.1.** *Given $\alpha \in (0, 1/3)$ fraction corrupted tasks, there exists an algorithm [17] that can robustly estimate the matrix $\mathbf{M} = \sum_{j=1}^{k} p_j \mathbf{w}_j \mathbf{w}_j^\top$ with $n = \Omega\left( \frac{d^2}{\alpha} \log \frac{d}{\delta} \right)$, and time $\widetilde{\mathcal{O}}\left(nd^2/\mathrm{poly}(\alpha)\right)$. The algorithm returns $\bar{\mathbf{M}}$ that satisfies*

$$\left\| \bar{\mathbf{M}} - \mathbf{M} \right\|_2 \lesssim \rho^2 \sqrt{\alpha}$$

*with probability at least $1 - \delta$ for $\delta \in (0, 1)$.*

#### K.1.1    Proof of robust mean estimation for the second moment in Remark K.1

From the definition of $\widehat{\mathbf{M}}$,

$$\widehat{\mathbf{M}} = (2n)^{-1} \sum_{i=1}^{n} \left( \widehat{\beta}_i^{(1)} \widehat{\beta}_i^{(2)\top} + \widehat{\beta}_i^{(2)} \widehat{\beta}_i^{(1)\top} \right), \tag{84}$$

which is the empirical mean of the matrices $\widehat{\beta}_i^{(1)} \widehat{\beta}_i^{(2)\top} + \widehat{\beta}_i^{(2)} \widehat{\beta}_i^{(1)\top}$. Let us consider the $d^2$-length vectors $\widehat{\mathbf{m}}_i$, and $\mathbf{m}_i$ constructed by unrolling the matrix $\widehat{\beta}_i^{(1)} \widehat{\beta}_i^{(2)\top}$, and $\beta_i \beta_i^\top$ respectively. Then

$(2n)^{-1}\mathbf{B}\sum_{i=1}^{n}\widehat{\mathbf{m}}_i$ will be the unrolled vector corresponding to the matrix $\widehat{\mathbf{M}}$ where $\mathbf{B} := \mathbf{I}_d + \mathbf{P}_d$, and $\mathbf{P}_d \in \mathbb{R}^{d^2 \times d^2}$ is the permutation matrix corresponding to the transposition of $d \times d$ matrices. The covariance matrix of the samples $\mathbf{B}\widehat{\mathbf{m}}$ is therefore

$$\mathbf{B}\,\mathbb{E}\big[\widehat{\mathbf{m}}\widehat{\mathbf{m}}^\top\big]\mathbf{B} - \mathbf{B}\mathbf{m}\mathbf{m}^\top\mathbf{B}, \tag{85}$$

and its norm is bounded by

$$\big\|\mathbb{E}\big[\widehat{\mathbf{m}}\widehat{\mathbf{m}}^\top\big] - \mathbf{m}\mathbf{m}^\top\big\|_2. \tag{86}$$

Returning back to the matrix notation, we therefore essentially need to bound the operator norm of the covariance tensor of the samples $\widehat{\beta}_i^{(1)}\widehat{\beta}_i^{(2)\top}$, where $\beta_i = \mathbf{w}_j$ with probability $p_j$. This can be bounded as

$$\begin{aligned}
&\sup_{\|\mathbf{A}\|_{\mathrm{F}}=1}\,\mathbb{E}_{i\sim\mathbf{p},\mathbf{x},y}\left[\mathrm{Tr}\Big[\mathbf{A}\Big(\widehat{\beta}_i^{(1)}\widehat{\beta}_i^{(2)\top} - \mathbf{M}\Big)\Big]^2\right]\\
&= \sup_{\|\mathbf{A}\|_{\mathrm{F}}=1}\,\mathbb{E}_{i\sim\mathbf{p},\mathbf{x},y}\left[\Big(\widehat{\beta}_i^{(2)\top}\mathbf{A}\widehat{\beta}_i^{(1)} - \mathrm{Tr}[\mathbf{A}\mathbf{M}]\Big)^2\right]\\
&= \sup_{\|\mathbf{A}\|_{\mathrm{F}}=1}\,\mathbb{E}_{i\sim\mathbf{p},\mathbf{x},y}\left[\Big(\widehat{\beta}_i^{(2)\top}\mathbf{A}\widehat{\beta}_i^{(1)}\Big)^2 - (\mathrm{Tr}[\mathbf{A}\mathbf{M}])^2\right]\\
&= \sup_{\|\mathbf{A}\|_{\mathrm{F}}=1}\,\mathbb{E}_{i\sim\mathbf{p},\mathbf{x},y}\left[\Big(\widehat{\beta}_i^{(2)\top}\mathbf{A}\widehat{\beta}_i^{(1)}\Big)^2 - \big(\mathrm{Tr}\big[\mathbf{A}\beta_i\beta_i^\top\big]\big)^2 + \big(\mathrm{Tr}\big[\mathbf{A}\beta_i\beta_i^\top\big]\big)^2 - (\mathrm{Tr}[\mathbf{A}\mathbf{M}])^2\right]\\
&\leq \sup_{\|\mathbf{A}\|_{\mathrm{F}}=1}\,\mathbb{E}_{i\sim\mathbf{p},\mathbf{x},y}\left[\Big(\widehat{\beta}_i^{(2)\top}\mathbf{A}\widehat{\beta}_i^{(1)}\Big)^2 - \big(\beta_i^\top\mathbf{A}\beta_i\big)^2\right] + \sup_{\|\mathbf{A}\|_{\mathrm{F}}=1}\,\mathbb{E}_{i\sim\mathbf{p}}\left[\big(\mathrm{Tr}\big[\mathbf{A}\beta_i\beta_i^\top\big]\big)^2 - (\mathrm{Tr}[\mathbf{A}\mathbf{M}])^2\right]\\
&\leq \mathbb{E}_{i\sim\mathbf{p}}\left[\sup_{\|\mathbf{A}\|_{\mathrm{F}}=1}\,\mathbb{E}_{\mathbf{x},y}\left[\Big(\widehat{\beta}_i^{(2)\top}\mathbf{A}\widehat{\beta}_i^{(1)}\Big)^2 - \big(\beta_i^\top\mathbf{A}\beta_i\big)^2\,\Big|\,i\right]\right]\\
&\qquad\qquad + \sup_{\|\mathbf{A}\|_{\mathrm{F}}=1}\,\mathbb{E}_{i\sim\mathbf{p}}\left[\big(\mathrm{Tr}\big[\mathbf{A}\beta_i\beta_i^\top\big]\big)^2 - (\mathrm{Tr}[\mathbf{A}\mathbf{M}])^2\right]
\end{aligned} \tag{87}$$

Considering the first term of Equation (87), where for a fixed $i$, we compute the inner expectation

$$\begin{aligned}
&\mathbb{E}_{\mathbf{x},y}\left[\Big(\widehat{\beta}_i^{(2)\top}\mathbf{A}\widehat{\beta}_i^{(1)}\Big)^2 - \big(\beta_i^\top\mathbf{A}\beta_i\big)^2\right]\\
&= \mathbb{E}_{\mathbf{x},y}\left[\widehat{\beta}_i^{(2)\top}\mathbf{A}\widehat{\beta}_i^{(1)}\widehat{\beta}_i^{(1)\top}\mathbf{A}^\top\widehat{\beta}_i^{(2)} - \beta_i^\top\mathbf{A}\beta_i\beta_i^\top\mathbf{A}^\top\beta_i\right]\\
&= \mathbb{E}_{\widehat{\beta}^{(1)}}\left[\mathbb{E}_{\mathbf{x},y}\left[\widehat{\beta}_i^{(2)\top}\mathbf{A}\widehat{\beta}_i^{(1)}\widehat{\beta}_i^{(1)\top}\mathbf{A}^\top\widehat{\beta}_i^{(2)} - \beta_i^\top\mathbf{A}\beta_i\beta_i^\top\mathbf{A}^\top\beta_i\,\Big|\,\widehat{\beta}_i^{(1)}\right]\right]
\end{aligned} \tag{88}$$

Define the PSD matrix $\mathbf{B} := \mathbf{A}\widehat{\beta}_i^{(1)}\widehat{\beta}_i^{(1)\top}\mathbf{A}^\top$, and note that its expectation is

$$\begin{aligned}
\mathbb{E}\left[\mathbf{A}\widehat{\beta}_i^{(1)}\widehat{\beta}_i^{(1)\top}\mathbf{A}^\top\right] &\leq \mathbf{A}\left(\Big(1+\frac{1}{t}\Big)\beta_i\beta_i^\top + \frac{\rho^2}{t}\mathbf{I}\right)\mathbf{A}^\top\\
&= \Big(1+\frac{1}{t}\Big)\mathbf{A}\beta_i\beta_i^\top\mathbf{A}^\top + \frac{\rho^2}{t}\mathbf{A}\mathbf{A}^\top
\end{aligned} \tag{89}$$

from which we get

$$\mathop{\mathbb{E}}_{\mathbf{x},y}\left[\left(\widehat{\beta}_i^{(2)\top}\mathbf{A}\widehat{\beta}_i^{(1)}\right)^2 - \left(\beta_i^\top\mathbf{A}\beta_i\right)^2\right]$$

$$= \mathop{\mathbb{E}}_{\widehat{\beta}_i^{(1)}}\left[\mathop{\mathbb{E}}_{\mathbf{x},y}\left[\widehat{\beta}_i^{(2)\top}\mathbf{B}\widehat{\beta}_i^{(2)} - \beta_i^\top\mathbf{A}\beta_i\beta_i^\top\mathbf{A}^\top\beta_i \;\Big|\; \widehat{\beta}_i^{(1)}\right]\right]$$

$$= \mathop{\mathbb{E}}_{\widehat{\beta}_i^{(1)}}\left[\operatorname{Tr}\left[\mathbf{B}\mathop{\mathbb{E}}_{\mathbf{x},y}\left[\widehat{\beta}_i^{(2)}\widehat{\beta}_i^{(2)\top}\right]\right]\right] - \beta_i^\top\mathbf{A}\beta_i\beta_i^\top\mathbf{A}^\top\beta_i$$

$$\leq \mathop{\mathbb{E}}_{\widehat{\beta}_i^{(1)}}\left[\operatorname{Tr}\left[\mathbf{B}\left(1+\frac{1}{t}\right)\beta_i\beta_i^\top + \frac{\rho^2}{t}\mathbf{B}\right]\right] - \beta_i^\top\mathbf{A}\beta_i\beta_i^\top\mathbf{A}^\top\beta_i$$

$$= \mathop{\mathbb{E}}_{\widehat{\beta}_i^{(1)}}\left[\left(1+\frac{1}{t}\right)\beta_i^\top\mathbf{B}\beta_i + \frac{\rho^2}{t}\operatorname{Tr}[\mathbf{B}]\right] - \beta_i^\top\mathbf{A}\beta_i\beta_i^\top\mathbf{A}^\top\beta_i$$

$$= \left(1+\frac{1}{t}\right)\beta_i^\top\left(\left(1+\frac{1}{t}\right)\mathbf{A}\beta\beta^\top\mathbf{A}^\top + \frac{\rho^2}{t}\mathbf{A}\mathbf{A}^\top\right)\beta_i$$

$$\quad + \frac{\rho^2}{t}\operatorname{Tr}\left[\left(1+\frac{1}{t}\right)\mathbf{A}\beta_i\beta_i^\top\mathbf{A}^\top + \frac{\rho^2}{t}\mathbf{A}\mathbf{A}^\top\right] - \beta_i^\top\mathbf{A}\beta_i\beta_i^\top\mathbf{A}^\top\beta_i$$

$$= \left[\left(1+\frac{1}{t}\right)^2 - 1\right]\left(\beta_i^\top\mathbf{A}\beta_i\right)^2 + \frac{\rho^2}{t}\left(1+\frac{1}{t}\right)\beta_i^\top\left(\mathbf{A}\mathbf{A}^\top + \mathbf{A}^\top\mathbf{A}\right)\beta_i + \frac{\rho^4}{t^2}\|\mathbf{A}\|_\mathrm{F}^2$$

$$= \left(\frac{2}{t}+\frac{1}{t^2}\right)\left(\operatorname{Tr}\left[\mathbf{A}\beta_i\beta_i^\top\right]\right)^2 + \frac{\rho^2}{t}\left(1+\frac{1}{t}\right)\left(\operatorname{Tr}\left[\beta_i^\top\mathbf{A}\mathbf{A}^\top\beta_i\right] + \operatorname{Tr}\left[\beta_i^\top\mathbf{A}^\top\mathbf{A}\beta_i\right]\right) + \frac{\rho^4}{t^2}\|\mathbf{A}\|_\mathrm{F}^2$$

$$\leq \left(\frac{2}{t}+\frac{1}{t^2}\right)d\|\mathbf{A}\|_\mathrm{F}^2\|\beta_i\|_2^4 + \frac{2\rho^2}{t}\left(1+\frac{1}{t}\right)\|\mathbf{A}\|_\mathrm{F}^2\|\beta_i\|_2^2 + \frac{\rho^4}{t^2}\|\mathbf{A}\|_\mathrm{F}^2$$

$$\leq \frac{3\rho^4}{t} + \frac{4\rho^4}{t} + \frac{\rho^4}{t^2}$$

$$= \frac{8\rho^4}{t}. \tag{90}$$

Plugging back Equation (90) in Equation (87) we get

$$\sup_{\|\mathbf{A}\|_\mathrm{F}=1}\mathop{\mathbb{E}}_{i\sim\mathbf{p},\mathbf{x},y}\left[\operatorname{Tr}\left[\mathbf{A}\left(\widehat{\beta}_i^{(1)}\widehat{\beta}_i^{(2)\top} - \mathbf{M}\right)\right]^2\right] \lesssim \frac{\rho^4}{t} + \sup_{\|\mathbf{A}\|_\mathrm{F}=1}\mathop{\mathbb{E}}_{i\sim\mathbf{p}}\left[\left(\operatorname{Tr}\left[\mathbf{A}\beta_i\beta_i^\top\right]\right)^2 - \left(\operatorname{Tr}[\mathbf{A}\mathbf{M}]\right)^2\right]$$

$$\lesssim \frac{\rho^4}{t} + \rho^4$$

$$\lesssim \rho^4$$

$$\implies \left\|\mathbf{B}\mathbb{E}[\widehat{\mathbf{m}}\widehat{\mathbf{m}}^\top]\mathbf{B} - \mathbf{B}\mathbf{m}\mathbf{m}^\top\mathbf{B}\right\|_2 \lesssim \rho^4. \tag{91}$$

Using [17, Theorem 1.3.], we finally get that the robust mean estimate $\bar{\mathbf{M}}$ of $\mathbf{M}$ can be computed using $n = \Omega\left(\frac{d^2}{\alpha}\log\frac{d}{\delta}\right)$ independent tasks in time $\mathcal{O}\left(nd^2/\operatorname{poly}(\alpha)\right)$, and satisfies

$$\left\|\bar{\mathbf{M}} - \mathbf{M}\right\|_2 \leq \left\|\bar{\mathbf{M}} - \mathbf{M}\right\|_\mathrm{F} \leq \mathcal{O}\left(\rho^2\sqrt{\alpha}\right) \tag{92}$$

with probability at-least $1-\delta$ for $\delta \in (0,1)$ using [17, Algorithm 2].

**Lemma K.2.** *Let $\beta \in \mathbb{R}^d$ be the true model for a task which gets to observe $t$ samples $\mathbf{X} \in \mathbb{R}^{t\times d}$, and provides labels $\mathbf{y} \sim \mathcal{N}\left(\mathbf{X}\beta, \sigma^2\mathbf{I}_t\right)$ where $y_j \in \mathbb{R}$ is the label for $\mathbf{x}_j \sim \mathcal{N}\left(\mathbf{0}, \mathbf{I}_d\right)$. The estimator for $\beta$ would be $\widehat{\beta} := \frac{1}{t}\mathbf{X}^\top\mathbf{y}$, and will satisfy*

$$\mathbb{E}\left[\widehat{\beta}\widehat{\beta}^\top\right] - \beta\beta^\top = \frac{1}{t}\beta\beta^\top + \frac{\|\beta\|_2^2 + \sigma^2}{t}\mathbf{I}. \tag{93}$$

*Proof.*

$$\widehat{\beta} = \frac{1}{t}\mathbf{X}^\top \mathbf{y}$$

$$= \frac{1}{t}\mathbf{X}^\top \mathbf{X}\beta + \frac{1}{t}\mathbf{X}^\top \epsilon \qquad (\text{where } \epsilon \sim \mathcal{N}(\mathbf{0}, \sigma^2 \mathbf{I}_t))$$

$$\implies \widehat{\beta} - \beta = \left(\frac{1}{t}\mathbf{X}^\top \mathbf{X} - \mathbf{I}_d\right)\beta + \frac{1}{t}\mathbf{X}^\top \epsilon$$

Let $\mathbf{z} := \widehat{\beta} - \beta$, then

$$\mathbb{E}[\mathbf{z}] = \mathbf{0}, \quad \text{and}$$

$$\mathbb{E}[\mathbf{z}\mathbf{z}^\top] = \mathbb{E}\left[\left(\left(\frac{1}{t}\mathbf{X}^\top \mathbf{X} - \mathbf{I}_d\right)\beta + \frac{1}{t}\mathbf{X}^\top \epsilon\right)\left(\left(\frac{1}{t}\mathbf{X}^\top \mathbf{X} - \mathbf{I}_d\right)\beta + \frac{1}{t}\mathbf{X}^\top \epsilon\right)^\top\right]$$

$$= \frac{\sigma^2}{t}\mathbf{I}_d + \frac{1}{t^2}\mathbb{E}[\mathbf{X}^\top \mathbf{X}\beta\beta^\top \mathbf{X}^\top \mathbf{X}] - \beta\beta^\top$$

$$= \frac{1}{t}\left(\|\beta\|_2^2 + \sigma^2\right)\mathbf{I}_d + \frac{1}{t}\beta\beta^\top \qquad (\text{Using Fact J.7}) \tag{94}$$

completing the proof. $\qquad\qquad\qquad\qquad\qquad\qquad\qquad\qquad\qquad\qquad\qquad\qquad\qquad\square$

## K.2 Experimental comparisons

We demonstrate the comparison between Algorithm 2 and HRPCA by considering a distribution class. Consider the distribution of the uncorrupted samples $\mathbf{x} \in \mathbb{R}^d$ to be as follows:

- $x_1 \sim \mathcal{N}(0, 1.1)$,
- $x_2 = z \cdot x_1/\sqrt{1.1}$, where $z$ is an independent Rademacher random variable,
- $\mathbf{x}_{3:d} \sim \mathcal{N}(\mathbf{0}_{d-2}, \mathbf{I}_{d-2})$,

and $\alpha$ is the corruption level. We sample $n \geq 5d/\alpha$ points from this distribution. Note that $\mathbf{\Sigma} := \mathbb{E}[\mathbf{x}\mathbf{x}^\top] = \mathbf{I}_d + 0.1\mathbf{e}_1\mathbf{e}_2^\top$. The adversary then corrupts a point $\mathbf{x}$ with probability $\alpha$ as

$$\mathbf{x}' \leftarrow \begin{bmatrix} 0 & z' \cdot 2\alpha^{1/4} & \mathbf{x}_{3:d} \end{bmatrix},$$

where $z'$ is an independent Rademacher random variable.

We run Algorithm 2 and HRPCA, on the above described setup by choosing $d = 10$, $\alpha \in \{0.005, 0.01, \ldots, 0.025\}$, $k = 1$, and $n = 10^4$. To evaluate the performance of both the algorithms, we compute the variance captured: $\text{Tr}\left[\widehat{\mathbf{U}}^\top \mathbf{\Sigma}\widehat{\mathbf{U}}\right]$, where $\widehat{\mathbf{U}} \in \mathbb{R}^{d \times k}$ is the output of either algorithms. We also compute the best oracle solution which is $\text{Tr}\left[\mathbf{U}_n^\top \mathbf{\Sigma}\mathbf{U}_n\right]$ where $\mathbf{U}_n \in \mathbb{R}^{d \times k}$ is the top $k$ singular vector matrix of $\widehat{\mathbf{\Sigma}}_n := \frac{1}{n}\sum_{i=1}^n \mathbf{x}_i\mathbf{x}_i^\top$ where $\{\mathbf{x}_i\}_{i=1}^n$ is the original uncorrupted sample set. Notice that this estimator is the optimal subspace estimator in the absence of extra structural assumptions about the subspaces.

We demonstrate the variance captured, the number of corrupted points left, and the number of uncorrupted points removed by the algorithms in Figure 2. A random guess of the subspace will capture roughly variance 1. The oracle estimator has variance $\approx 1.0886$. The best rank-1 subspace is spanned by $\mathbf{e}_1$ whose captured variance is 1.1. We show the average performance of Algorithm 2 over 100 independent trials. The HRPCA algorithm is very slow since one need to pick the best subspace from the $n/2$ iterations while each iteration requires eigen-decomposition once. Thus we only take the average over 10 trials, which is enough to see the trend of its performance.