[Reviews · NeurIPS 2020]

Review 1

Summary and Contributions: This paper studies the meta-learning in the setting of mixture of k linear regressions. Based on an outlier-robust PCA algorithm and a SOS-based algorithm to utilize the higher order moments, it improves the sample complexity required for heavy tasks.

Strengths: The paper is sound. It improves the theoretical guarantees in terms of the sample complexity, and extends the existing work to accommodate adversarial data. The idea of using the SOS-based method to exploit the higher order statistics as well as the robust subspace estimation algorithm may find useful for other problems. And the topic is trendy in learning at the moment.

Weaknesses: It would improve the quality of the paper if more comprehensive numerical experiments can be provided, especially the performance of Algorithm 1.

Correctness: I believe them to be correct.

Clarity: The paper is well written.

Relation to Prior Work: The paper gives a nice review of and comparison with exsiting works.

Reproducibility: Yes

Additional Feedback:


Review 2

Summary and Contributions: The paper designs an outlier-robust PCA algorithm and a sum-of-squares algorithm to exploit the information from higher order moments. The proposed approach is robust to adversarial corruption and achieves a statistical trade-off.

Strengths: 1. The motivation is clear and the author provides clear and intuitive analysis. 2. The claims and conclusion are convincing supported by sufficient theoretical analysis and proof. 3. The topic and contribution are suitable for the NeurIPS community.

Weaknesses: 1. No sufficient experimental evaluation for its performance and comparison 2. The claim that “Our main result relies on making each step of Algorithm 1 robust” has no corresponding empirical verification.

Correctness: Correct [but not carefully check for each proof and equation]

Clarity: Basically, it is well written, but there should be suitable trade-offs for the main paper and supplement, theoretical analysis and experimental (if provided more)

Relation to Prior Work: Yes, the paper discusses the relation between this work and previous work, and analyses the limitation of prior work, highlights the contribution of this work.

Reproducibility: Yes

Additional Feedback: The paper design a novel meta-learning approach which is robust to data scarcity and adversarial corruption. The author provides clear and intuitive analysis for the use of light and heavy tasks. The proposed algorithms and corresponding claims are convincing supported by sufficient theoretical analysis and proof. But there is no sufficient evaluation for its performance and comparison in experiment part, the figure 2 only verifies the impact of the corrupted points. So, the actual improvement and performance of the algo. 1 are not clear; in addition, the claim that “Our main result relies on making each step of Algorithm 1 robust” has no corresponding empirical verification. So, there should be suitable trade-offs for the main paper and supplement, theoretical analysis and experimental (if provided more).


Review 3

Summary and Contributions: This paper deals with "meta-learning" for the case of mixture of linear regressions. More specifically, suppose we deal with n linear regression data sets after which we are challenged with a final learning task of linear regression, but the parameters of these “tasks” are not completely unrelated. In particular, suppose there is a prior distribution (with at most k possible outcomes) from which parameters of linear regression (i.e., the linear function and noise's variance) are sampled. The general idea here is that by learning from "different" (yet related) tasks the learner aims to do better on the final task, and the paper’s focus is on a theoretically natural setting. (One of the) paper's contribution is to address with the *robust* version of the mentioned problem. Namely, it is shown how to do the meta learning (of mixed linear regressions) when some small (say 1%) fraction of each task (out of n tasks) is corrupted adversarially. The paper also shows how to deal with "small batches" setting in which even the larger batches are << sqrt(k) in size, though then the paper's algorithm needs more batches. E.g., if *only* small tasks are given (of size log k) then the paper's algorithm needs quasi-polynomial(k)=n tasks and running time. The paper does this by looking at more moments (than just the 2nd moment, which was the approached of previous work stuck at requiring batches of size sqrt(k)). The algorithm follows the following recipe of first finding a subspace spanned by the k possible regression vectors (a form of PCA) then a clustering of that space to find the k parameters and then a "classification" step that fine tunes the found parameters. The middle step uses large batches and the other steps use small batches. The paper's contribution is to do all three steps above in a robust way (the 2nd of which uses ideas from some-of-squares methods) such that they can tolerate some fraction of arbitrary points added to it. At the same time, the paper also shows how to manage small batches of way smaller than what previous work could handle (going from sqrt(k) to log(k)). Post rebuttal: Thanks for the clarifications. They answered all of my questions.

Strengths: Addressing an important basic problem in the theory of meta learning. Extension to the robust regime is very interesting. The paper (at least the main body) is written quite well.

Weaknesses: The abstract and intro both start with mentioning practical scenarios (i.e., medical image processing) but I found no more discussions in the paper that points out to such applications of the proved results. I know that this is theoretical work, but still it would be better if applications of such meta learning framework were explored as well.

Correctness: I did not check the proofs in detail, but the discussions, intuition, and citations for technical tools seem solid to me.

Clarity: For someone who is not working on this area, I still found the paper accessible (partly due to formally stated setting and definitions of the problems).

Relation to Prior Work: It seems so.

Reproducibility: Yes

Additional Feedback: Assumption 2 needs clarification: Do you mean the adversary's knowledge in tampering with each of the three groups of batches is oblivious to the other two groups of batches? or could the adversary first look at *all* batches and then pick the corruptions? minor comments/typos: please define the notation \Tilde\Omega(.). I guessed that means hiding polylog(k) factors, but that is not quite as standard yet, so needs clarification. line 51: "it is critical tailor" -> to tailor? line 78: please say what epsilon is (it becomes clear later). line 89: "but focusing on a simple but canonical" -> yet canonical (repeating but) Line 96: you first describe the general goal of meta learning as learning a final task from previously related tasks. But then the recipe for doing so is split into two steps: first approximating the meta parameter and then doing the final learning. This seems like a very natural approach, but could there be any other more *direct* approach that combines the two steps? E.g, one can define a maximum likelihood formulation that directly aims to maximize the chance of a label y for a given query x for the final task given ALL the data so far. I guess this would be hard to do algorithmically, but just commenting that the way these paragraphs are written it seems like doing as discussed is the only way to do it, because you say in line 105 that "the meta-learning problem refers to solving the following:...". line 107-8: Our goal is solve this -> is to solve (or, is solving) page 4: "As k << d in typical settings" Why? also, maybe good to bring this up earlier as helps getting the right intuition. In algorithm 2: Define k_SVD (k first singular values?) line 250: the sentence needs to be rewritten.


Review 4

Summary and Contributions: I have read the author feedback. This paper looks at learning mixed linear regressions which were observed across a collection of data sets ranging from small to larger. Additionally, the datasets are allowed to be corrupted by adversarial noise of some predetermined size. The authors present an algorithm to robustly estimate the model parameters by (1) estimating the subspace within which the linear regression parameters live (2) grouping "large" regression tasks in this subspace then learning a regression coefficient for each; and finally (3) assigning "small" regression tasks to these clusters and using their information to refine the parameter estimates. The main result of the paper is Theorem 1 which gives estimation bounds on the various model parameters as a function of the amount adversarial noise and other inputs. Some very small experiments are provided.

Strengths: The main contribution are the theoretic guarantees on the estimation accuracy of a meta-learning algorithm in terms of adversarial corruption rates.

Weaknesses: As the length of the appendix indicates this is clearly a substantial topic with great depth. However, overall, I found the paper very hard to read. For a start, the motivation for the adversarial setup is weak. The authors should make a more convincing case why these types of extensions are necessary in practice. E.g., why are the existing methods they mention brittle and why is this the right adversarial model? In what way does this improve over other method's performance in practice? The first part of the paper seems to be aimed at building early intuition. However, I found it somewhat unhelpful as it presents results before we've even seen what the model in question is. For example, the adversarial model is only explained in Assumption 2 in Section 2, i.e., 5 pages into the paper. The generative model for the data is only introduced on page 3, making it hard to interpret the various corollaries that come before. Additionally, Section 2 mentions much prior context in passing which isn't very well explained (e.g., SOS, Poincar\'e distributions). I would encourage the authors to further streamline the presentation and to clarify context. The only part of the paper that is experimentally verified is the subspace estimation method, which forms only one small part of Algorithm 1 (and the description of the experimental setup is in the Appendix). I would encourage the authors to provide a more comprehensive suite of experiments, which contrasts to existing outlier methods to demonstrate the effectiveness of this approach.

Correctness: I was not able to verify all theoretical details. Most of Algorithm 1 is not empirically verified.

Clarity: As mentioned above, I found the paper hard to read. Instead of giving key intuitions, the authors focus a lot the space on discussing sample rates and complexities. There are almost no experiments given to underline key conclusions.

Relation to Prior Work: The authors discuss related work, which is helpful, though I would have liked to see more justification why this particular adversarial model is the "right" one. Much more context should be given on SOS methods.

Reproducibility: Yes

Additional Feedback: Corollary 1.1: I am a bit surprised that there is no dependence shown on the constant hidden in the O(1) notation. How does this impact the required sample/time complexity? L74: What is the dependence on k in the O(d^2) notation? Thm. 1: Why is the only dependence on delta in t_L2? Shouldn't the clustering step also have a bearing on overall quality of the estimation?

[Author Response · NeurIPS 2020]

We thank the reviewers for their constructive feedback, and first address **adding more experiments**, common to
Reviewers 1, 2 and 4. Our main *algorithmic* innovation is in step 1 (robust subspace estimation), and our contribution
in steps 2 and 3 (robust clustering and robust estimation) are *theoretical*, as we borrow existing algorithms. Hence, it
was natural for us to focus on experimentally verifying step 1. However, we agree that more experiments will help
solidify the theoretical guarantees, and will verify the following experimentally: $(a)$ an experiment showing that the
SOS approach (step 2) is robust under the linear regression probabilistic model which is not Poincaré; and $(b)$ add an
experiment showing that robust parameter estimation succeeds when applied with the classification together (step 3).

**Detailed response to Reviewer 3:**

• *Q: Could the adversary first look at \*all\* batches and then pick the corruptions?* A: Yes, the adversary can take a
look at all three groups of batches and add corruption. We have revised Assumption 2 accordingly.

• *Q: more \*direct\* approach ...* A: [31] defines "meta-learning" as Eq. (2). However, we agree there are many ways
to meta-learn, some of which are more direct but less understood. We will survey those approaches in Section 3.

• In practice, we oftentimes have problems with a large ambient data dimension, but a simple structure among the
meta-training tasks (captured by small $k$ in our setting). Our approach is tailored for such settings with $k \ll d$.

• We will address all the comments and typos in the final version of the paper.

**Detailed response to Reviewer 4:**

• *Q: ...why these types of extensions are necessary in practice...why are the existing methods they mention brittle...*
A: Existing methods can completely break down with a single corrupted user. We will add the following remark and
references: " [41] builds upon principal component analysis and linear regression, both of which are known to be
brittle to outliers [39,19]. For example, a single corrupted user can result in an arbitrarily bad subspace estimation in
the first step of [41]. This causes the meta-learning algorithm to learn nothing about the true regression parameters,
resulting in an completely random prediction in the subsequent step."
In particular, under the setting of Corollary 1.3, with a tiny fraction of adversary with $\alpha = 1/n$, our approach
guarantees error $\mathcal{O}(k/n)$ that decreases with $n$ whereas [41] will have error $\Omega(\rho/\Delta)$ that does not decrease with $n$.
Further, for general $\alpha \in (0, 1/2)$, naive pruning techniques, for example removing the datapoints with extremely
large magnitude, is not enough to resolve the issue in a high dimension.

• *Q: Why is this the right adversarial model?* A: This is the right model for security, in the sense that it is the strongest
adversarial model (among those that can corrupt the same number of samples), and more importantly, we can still
make the algorithm robust. We will add a remark that "Following robust learning literature [44, 25], we assume a
general adversary who can adaptively corrupt any $\alpha$ fraction of the tasks, formally defined in Assumption 2. This
parameter $\alpha \in [0, 1]$ captures how powerful an adversary is. Among all adversaries that can corrupt an $\alpha$ fraction
of the dataset, we assume the strongest possible one that can *adaptively* select which samples to corrupt and replace
them with *arbitrary* data points.". This is also a realistic adversarial model, in settings like federated learning where
an $\alpha$ fraction of devices can be compromised.

• *Q: ...unhelpful as it presents results before we've even seen what the model in question is...* A: We moved the
generative model earlier than Corollary 1.1. We moved the adversarial model earlier than Corollary 1.3.

• *Q: Much more context should be given into SOS methods.* A: Due to the space limitation, we had to be selective. In
the revision, we will add a subsection in Section 3 with preliminary on SOS methods applied to robust estimation.

• *Q: Corollary 1.1: I am a bit surprised that there is no dependence...* A: Since Corollary 1.1 is an informal version,
we restrict our focus on $d$ and $k$ and assumed that the error $\epsilon$ is a positive constant. A more formal version of
Corollary 1.1 is Corollary 1.3 and Theorem 1, where the dependencies on the final accuracy are highlighted in
adversarial tolerance, sample and running time complexity. Below we re-write Corollary 1.1 with dependency in $\epsilon$:

**Corollary.** *For any $\epsilon > 0$, given a collection of $n$ tasks each associated with $t = \tilde{\Omega}(1)$ labeled examples, if the*
*effective sample size $nt = \tilde{\Omega}(dk^2 + k^{\Theta(\log k)} + dk/\epsilon^2)$, then Algorithm 4 estimates the meta-parameters up to the*
*accuracy of $\epsilon$ w.h.p. in time $\mathrm{poly}(d, k^{(\log k)^2}, 1/\epsilon)$, under certain assumptions on the meta-parameters.*

• *Q: L74: What is the dependence on $k$ in the $\tilde{\mathcal{O}}(d^2)$?* A: $\tilde{\mathcal{O}}(d^2)$ sample suffices for estimating the covariance matrix
itself accurately under Frobenius norm, which implies accurate estimation of top-$k$ subspaces for any $k$. Therefore,
there is no dependence of $k$ in $\tilde{\mathcal{O}}(d^2)$, as we explicitly write in Remark K.1 in the supplementary material.

• *Q: Thm. 1: Why is the only dependence on $\delta$ in $t_{L2}$?* A: The target guarantee is parametrized by the failure
probability $\delta$ and the accuracy $\epsilon$. For subspace estimation and clustering, we apply concentration of measure on the
whole dataset, and hence $n_{L1}$ and $n_H$ depends on $\log(1/\delta)$, which is hidden in the $\tilde{\Omega}(\cdot)$ notation. For classification,
we apply concentration to each task, and hence $t_{L2}$ depends on $\log(1/\delta)$. As for the accuracy $\epsilon$, (as we explain in
L159 of the submission), the subspace estimation and clustering steps succeed with high probability as soon as they
achieve accuracy of $\mathcal{O}(\Delta/\rho)$, regardless of the final target accuracy $\epsilon$. The refinement with classification is solely
in charge of achieving the target $\epsilon$ accuracy, and hence $1/\epsilon^2$ dependence only shows up in $n_{L2}$.

• We will modify the presentation of the setting and priors for better readability in the final submission as suggested.

[Meta-Review · NeurIPS 2020]

The reviewers uniformly felt that this is an interesting paper and a good contribution to the community.